# STaRFormer: Semi-Supervised Task-Informed Representation Learning via Dynamic Attention-Based Regional Masking for Sequential Data

**Maximilian Forstenhäusler**[1,2,∗]   **Daniel Külzer**[1]   **Christos Anagnostopoulos**[2]
**Shameem Puthiya Parambath**[2]   **Natascha Weber**[1]

[1]BMW Group     [2]University of Glasgow

Project Page: https://star-former.github.io

## Abstract

Understanding user intent is essential for situational and context-aware decision-making. Motivated by a real-world scenario, this work addresses intent predictions of smart device users in the vicinity of vehicles by modeling sequential spatiotemporal data. However, in real-world scenarios, environmental factors and sensor limitations can result in non-stationary and irregularly sampled data, posing significant challenges. To address these issues, we propose STaRFormer, a Transformer-based approach that can serve as a universal framework for sequential modeling. STaRFormer utilizes a new dynamic attention-based regional masking scheme combined with a novel semi-supervised contrastive learning paradigm to enhance task-specific latent representations. Comprehensive experiments on 56 datasets varying in types (including non-stationary and irregularly sampled), tasks, domains, sequence lengths, training samples, and applications demonstrate the efficacy of STaRFormer, achieving notable improvements over state-of-the-art approaches.

## 1   Introduction

Advancements in machine learning architectures, such as LSTM [1] and Transformer [2], have enhanced the ability to model sequential data. However, these algorithms typically assume that the data is fully observed, stationary, and sampled at regular intervals [3]. In reality, sensor technology and external conditions often influence data collection, leading to non-stationary and irregularly sampled time series. For instance, in the automotive industry, manufacturers have recently integrated Ultra-Wideband (UWB) and Bluetooth Low-Energy (BLE) technologies to enhance the Digital Key (DK) [4–8]. This integration ensures precise and secure vehicle access along with applications for connected vehicles. Precise localization is achieved by performing time-of-flight calculations between each UWB anchor in a vehicle and a smart device, leveraging UWB's $2ns$ pulse duration [9]. Nonetheless, the measuring algorithm for UWB ranging may yield irregularly recorded time-of-flight calculations, resulting in irregularly sampled time series. Additionally, when recording real-world data using UWB-capable ranging devices, external factors such as signal interference and device positioning can introduce non-stationarity. These conditions may ultimately affect the overall performance of Machine Learning (ML) algorithms. In the real-world Digital Key Trajectories (DKT) dataset provided by the BMW Group (Appendix A and C.1.1), we confirmed, by Kwiatkowski–Phillips–Schmidt–Shin (KPSS) and augmented Dickey-Fuller (ADF) tests, that approximately $79\%$ of the sequences are non-stationary. Based on the real-world trajectories generated from the DK, we focus on predicting the smart device user's intent, formulated as a specific classification task.

---

∗Email: maximilian.forstenhaeusler@bmw.de, m.forstenhaeusler.1@research.glasgow.ac.uk

39th Conference on Neural Information Processing Systems (NeurIPS 2025).

Generally, trajectories involve variables such as latitude, longitude, altitude, and speed, which are often irregular. Similarly, weather conditions, geographical barriers, sensor availability, and device malfunctions [10] can result in non-stationary characteristics, aligning with the properties found in the DKT dataset. Although several solutions exist to address these issues, they require substantial prior knowledge and effort in model selection [11–20]. To address these challenges, we propose a versatile framework, STaRFormer, designed to effectively model time series with the aforementioned characteristics while maintaining applicability to standard time series data. STaRFormer proposes dynamic regional masking to manipulate key task-specific regions within an input sequence, introducing synthetic variations in statistical properties, such as mean, variance, and sampling frequency. By incorporating this masking layer during the learning process of a downstream task, STaRFormer generates masked and unmasked latent representations of the same input sequence. Building on prior work, which highlighted that the task-specific importance of elements within a sequence can vary in their influence on downstream tasks [21, 22], we extend this approach by coupling representation learning with a downstream objective. This coupling allows to incorporate context-specific information that may be overlooked in decoupled self-supervised frameworks [23–27]. Through a novel combination of self-supervised and supervised contrastive learning (CL), STaRFormer creates robust task-informed latent embeddings by maximizing agreement between class-wise and batch-wise similarities of the masked and unmasked latent representation. This technique is designed to enhance the model's robustness to irregularities in time series while serving as an augmentation method to improve performance for various time series types and tasks. In summary, our main **contributions** are:

- We propose **STaRFormer**, a highly effective and robust approach boosting the performance of downstream tasks for diverse types of time series and tasks.

- We develop a novel **semi-supervised CL** approach for time series analysis, leveraging batch-wise and class-wise similarities by reconstructing latent representations from masked inputs.

- We design a novel **Dynamic Attention-based Regional Masking (DAReM)** scheme that identifies task-specific important regions of a sequence, allowing to embed task-specific knowledge.

- We assess STaRFormer using **56 public and non-public datasets** to validate its effectiveness compared to state-of-the-art methods, highlighting its versatility for various types of time series.

## 2 Related work

**Regular time series modeling for classification.**   Time series modeling for classification seeks to analyze and identify patterns in sequential data collected over consistent time intervals, with the goal of assigning labels to entire sequences or per elements within the sequence. It is generally assumed that the sequential data is stationary and uniformly sampled. Common ML baselines include dimension-dependent dynamic time warping (DTWD) [28, 29] and WEASEL-MUSE [30]. Deep Learning (DL) has proven powerful for time series classification by automatically extracting complex features. Unlike traditional methods that rely on handcrafted features, DL models such as RNN [31, 32], LSTM [1] and GRU [14] learn hierarchical representations directly from the data. However, these models often struggle with capturing long-term dependencies and spatiotemporal patterns. ROCKET [33] and MiniROCKET [34], CNNs that have been effective in capturing local dependencies, have achieved impressive results by learning features through diverse random convolutional kernels. Transformer-based approaches have recently gained attention due to their ability to capture long-range dependencies in sequential data. Various Transformer-based models have been proposed for forecasting, classification, and anomaly detection [3, 21, 23, 35–40]. Initial approaches utilized a full encoder-decoder Transformer architecture for univariate time series forecasting [41], while TST [23] generalized unsupervised representation learning for Transformers and time series, similarly to BERT's Masked Language Modeling (MLM) [42]. TARNet [21] addresses the issue of decoupling unsupervised pretraining from downstream tasks using dynamic masking and reconstruction. We address the challenge of time series classification by pairing a novel semi-supervised CL approach with a proposed generalization of the dynamic masking approach from TARNet. In doing so, we extend the proposition of coupling representation learning while learning a downstream task.

**Non-stationary and irregularly sampled time series modeling.**   Non-stationary time series modeling addresses the variability in statistical properties over time, i.e., changing means and covariances [43, 44]. Traditional models often fail to capture these dynamics. While most research has focused

on forecasting, some efforts have been directed towards non-stationary time series classification. Recent advancements include adaptive RNNs [45, 46], normalization-based approaches [47, 35], and non-stationary Transformers, which incorporate non-stationary factors to improve accuracy while addressing distribution shifts [48]. Irregularly sampled time series modeling addresses sequences with varying time intervals between observations. A standard solution is converting continuous time observations into fixed intervals [13, 15]. Several models have been proposed to capture dynamics between observations such as GRU-D [16] and multi-directional RNN [49]. Attention-based models, including Transformers, [2, 23] and ATTAIN [50], incorporate attention mechanisms to handle time irregularity. Raindrop [20] uses graph neural networks to model irregular time series as graphs. Meanwhile, TrajFormer [51] introduces a Transformer architecture that generates continuous point embeddings to deal with irregularities of trajectories. Recently, ViTST [3] focused on time series in the visual modality by transforming sequences into visualized line graphs, leveraging pretrained Vision-Transformer backbones. To handle non-stationarity and sampling irregularity, we introduce a dynamic regional masking strategy that perturbs input sequences by modifying their statistical and sampling properties. Coupled with our CL scheme, this representation learning approach promotes robustness to distributional shifts and irregular sampling, enhancing the latent space rather than relying solely on input reconstruction.

**Time series contrastive learning.** CL has proven effective in extracting high-quality, discriminative features [52]. CL operates as a self-supervised learning paradigm, learning representations by contrasting positive and negative pairs. The goal is to bring similar (positive) pairs closer and push dissimilar (negative) pairs apart, typically using contrastive losses like NT-Xent [52], InfoNCE [53], or triplet loss [54]. For sequential data, self-supervised CL aims to extract invariant representations from augmented views of unlabeled data through carefully designed pretext tasks. Methods such as TCL [55], and TNC [56] use subsequence-, neighborhood-based sampling assuming distant segments as negative pairs and neighboring segments as positive pairs. InfoTS [26] emphasizes appropriate augmentation selection using meta-learning, and TS2Vec [24] learns contextual representations across semantic levels. CoST [25] uses model inductive biases to separate seasonal and trend patterns, introducing a frequency-domain contrastive loss. However, these methods often suffer from flawed augmentations, weak negative samples, and limited information use [27]. TimesURL [27] proposes a self-supervised framework that combines CL, time reconstruction, and a frequency-temporal augmentation with hard negative sampling to learn universal time series representations for diverse downstream tasks. While prior work applies self-supervised CL to learn universal time series representations, we propose a task-coupled approach that jointly optimizes representation learning with the downstream objective, embedding task-specific information into the representations.

## 3 Approach

STaRFormer adopts a Siamese network architecture [57] consisting of two 'towers' of $N$ encoder-only Transformer blocks, $f$, that share a common set of model parameters. STaRFormer is illustrated in Fig. 1. Without loss of generality, we consider classification, anomaly detection and regression as downstream tasks. For sequence-level classification tasks, a special token is utilized to facilitate the downstream predictions. The other downstream tasks utilize appropriate variations, such as pooling operations or element-wise predictions, to facilitate the computation of the desired task predictions. For detailed information, output head formulations, and related remarks, see Appendix B.

**Notation.** Let $\mathcal{D} = \{(\mathbf{S}^{(i)}, y^{(i)}) \mid i = 1, \ldots, M\}$ denote a time series dataset containing $M$ samples. Each sequence, $\mathbf{S}^{(i)} \in \mathbb{R}^N$ has $N$ elements and is assigned to a label $y^{(i)} \in \{1, \ldots, C\}$, where $C$ is the number of classes. Each data point in the sequence can have an associated timestamp. Thus, the $j$-th data point in $\mathbf{S}^{(i)}$ can be represented as $\mathbf{s}_j^{(i)} = (x_j^{(i)}, t_j^{(i)}) \in \mathbb{R}^2$. Therefore, $\mathbf{S}^{(i)} = \{\mathbf{s}_j^{(i)} \mid j = 1, \ldots, N\} \in \mathbb{R}^{N \times 2}$ is formed by concatenating all $N$ elements. For multivariate time series, the dimensionality is not fixed to two; thus $\mathbf{S}^{(i)} \in \mathbb{R}^{N \times D}$, where $N \in \mathbb{N}_{\neq 0}$ and $D \in \mathbb{N}_{\geq 2}$. A mini-batch of size $B$, where $B \ll M$, is defined as $\mathbf{X} \subset \mathcal{D}$, where $\mathbf{X} \in \mathbb{R}^{N \times B \times D}$.

**Problem 1 - Classification.** Given a dataset $\mathcal{D} = \{(\mathbf{S}^{(i)}, y^{(i)}) \mid i = 1, \ldots, M\}$ where $\mathbf{S}^{(i)} \in \mathbb{R}^{N \times D}$ can be multivariate, predict the class $y^{(i)} \in \{1, ..., C\}$, for each sequence $\mathbf{S}^{(i)}$ in $\mathcal{D}$.

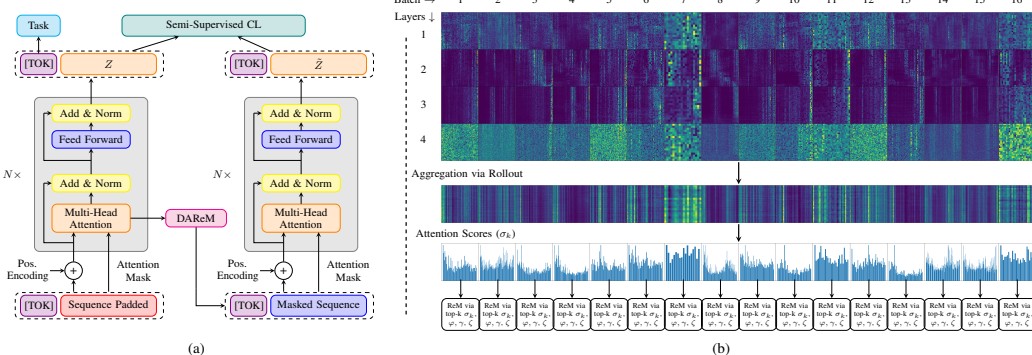

Figure 1: Architecture of STaRFormer; (a) High level Siamese network architecture - the left tower performs the downstream task while the right tower performs the reconstruction of the masked sequence. (b) The DAReM scheme exemplified by a single batch from the DKT dataset with batch size 16 for an encoder with $N = 4$ layers. ReM abbreviates regional mask.

**Problem 2 - Anomaly detection.** Given a dataset $\mathcal{D} = \{(\mathbf{S}^{(i)}, \mathbf{y}^{(i)}) \mid i = 1, \ldots, M\}$ where $\mathbf{S}^{(i)} \in \mathbb{R}^{N \times D}$ can be multivariate and $\mathbf{y}^{(i)} \in \mathbb{R}^N$, predict for each element of sequence $\mathbf{S}^{(i)}$ in $\mathcal{D}$ whether $y_{j \in N}^{(i)} = 0$ (normal observation) or $y_{j \in N}^{(i)} \in \{1, \ldots, C\}$ (anomalous observation).

**Problem 3 - Regression.** Given a dataset $\mathcal{D} = \{(\mathbf{S}^{(i)}, \mathbf{y}^{(i)}) \mid i = 1, \ldots, M\}$, where $\mathbf{S}^{(i)} \in \mathbb{R}^{N \times D}$ can be multivariate, predict the continuous target value $y^{(i)} \in \mathbb{R}$ for each sequence $\mathbf{S}^{(i)}$ in $\mathcal{D}$.

### 3.1 Semi-supervised task informed representation learning

This section presents STaRFormer's components facilitating task-informed representation learning.

#### 3.1.1 Dynamic attention-based regional masking (DAReM)

Prior work has shown that the task-specific importance of elements within a sequence varies w.r.t. their impact on downstream tasks [21, 22]. STaRFormer adopts this characteristic by dynamically masking regions around the features the model deems important. These masks force the model to learn changes in statistical properties and irregular sampling induced by the masking. Our rationale is that reconstructing key sequential regions amplifies non-stationary and irregular sampling characteristics. This enables the model to generate more effective latent representations for the downstream task. This masking scheme, termed DAReM, can be seen as a generalization of the masking scheme proposed in [21]. During training of a downstream task, STaRFormer dynamically gathers attention weights $\mathbf{A} = \mathrm{softmax}\left(\frac{QK^T}{\sqrt{d_k}}\right)$, $\mathbf{A} \in \mathbb{R}^{L \times B \times N \times N}$ (left tower, Fig. 1), where $L, B, N$ represent the number of attention layers, the mini-batch size, and the number of elements in the sequences, respectively. The attention weights, denoted as $\mathbf{A}$, essentially indicate the importance of each sequential element with respect to each other. The collected attention weights are then aggregated via attention rollout [58], refer to Eq. (39), resulting in $\tilde{\mathbf{A}} \in \mathbb{R}^{B \times N \times N}$. In order to determine the 'global' importance of specific elements within a sequence, we compute the attention scores, $\sigma_{i,k'}$, refer to Eq. (40), as in [21], where greater $\sigma_{i,k'}$ values indicate a higher importance of a sequential element and vice versa. The resulting attention scores, $\sigma_{i,k'} \in \mathbb{R}^{B \times N}$, allow a distinct masking scheme for each element in $\mathbf{X}$, resulting in $B$ masks per $\mathbf{X}$. The creation of the regional mask, $g : \mathcal{D} \to \mathcal{R}$, requires three hyperparameters: $\varphi$, determines the maximum amount of elements that are masked; $\zeta$, determines the number of sequential elements that are masked based on the attention scores $\sigma$ (see Eq. (40)); and $\gamma$, which determines the bounds of the region to be masked. Further details of DAReM, including the implementation, are provided in Appendix B.3.

#### 3.1.2 Semi-supervised contrastive learning

Previous work has focused on pretraining techniques aimed at creating generalizable time series representations applicable to a wide range of downstream tasks, as well as on learning sequence reconstructions both during pretraining and during training of a downstream task [24, 27, 42, 23, 21].

Instead, STaRFormer aims to enhance the latent space representation utilized by the model to perform a downstream task. While training for a downstream task, DAReM allows the creation of two correlated latent representations, i.e., masked ($\tilde{\mathbf{Z}}^{(i)}$) and unmasked ($\mathbf{Z}^{(i)}$). It is well know that CL can extract high-quality, discriminative features [52–54, 59, 55, 60]. Thus, with STaRFormer, we aim to facilitate CL in optimizing the trade-off between these representations (Appendix B.3, Fig. 8) leveraging unmasked and masked embeddings of the same input sequence as **batch-wise** and of the same class as **class-wise positive pairs**. This aims to: strengthen the model's robustness to perturbations, enhance generalization, reduce overfitting, and improve resilience to challenges like non-stationarity and irregular sampling. Based on these positive pairs, STaRFormer fuses two types of CL tasks: (i) **self-supervised** using batch-wise, and (ii) **supervised** using class-wise similarities. We propose three formulations: the first requiring a class label per sequence; the second requiring a label for every sequential element; and the third requiring a scalar target value per sequence.

**Formulation 1 - Sequence-level prediction tasks.** During training, the latent spaces $\mathbf{Z}, \tilde{\mathbf{Z}} \in \mathbb{R}^{N \times B \times F}$ become three-dimensional tensor representations, where $\mathbf{Z} = f(\mathbf{X}), \tilde{\mathbf{Z}} = f(g(\varphi, \gamma, \zeta, \mathbf{X}))$ and $F$ is the latent embedding dimension. To extract the similarity scores, by computing the inter-sequence cosine similarity ($\text{sim}(\mathbf{u}, \mathbf{v}) = \mathbf{u}^T \mathbf{v} / \|\mathbf{u}\| \|\mathbf{v}\|$) between the sequences in a batch, we average the latent representations along their first dimension, i.e., $\hat{\mathbf{Z}}_{i,j} = \frac{1}{N} \sum_{n=1}^{N} \mathbf{Z}_{n,i,j} \mid \in \mathbb{R}^{B \times F}$, reducing each sequence to a single vector representation. This allows us to formulate the NT-Xent [52] inspired batch-wise contrastive loss for a single positive batch-wise sample as:

$$l_{\text{bw}}^{(i)} = -\log \frac{\exp\left(\text{sim}\left(\hat{\mathbf{z}}^{(i)}, \hat{\tilde{\mathbf{z}}}^{(i)}\right) / \tau\right)}{\sum_{k=1}^{B} \mathbb{I}_{[k \neq i]} \exp\left(\text{sim}\left(\hat{\mathbf{z}}^{(i)}, \hat{\tilde{\mathbf{z}}}^{(k)}\right) / \tau\right)} \tag{1}$$

and the class-wise contrastive loss for a single positive class-wise sample as:

$$l_{\text{cw}}^{(i)} = -\log \frac{\sum_{j=1}^{B} \mathbb{I}_{[\mathcal{C}_j = \mathcal{C}_i]} \exp\left(\text{sim}\left(\hat{\mathbf{z}}^{(i)}, \hat{\tilde{\mathbf{z}}}^{(j)}\right) / \tau\right)}{\sum_{k=1}^{B} \mathbb{I}_{[\mathcal{C}_k \neq \mathcal{C}_i]} \exp\left(\text{sim}\left(\hat{\mathbf{z}}^{(i)}, \hat{\tilde{\mathbf{z}}}^{(k)}\right) / \tau\right)}. \tag{2}$$

The indicator function differs in the two cases: $\mathbb{I}_{[k \neq i]}$ for batch-wise, which is 1 iff $k \neq i$, $\mathbb{I}_{[\mathcal{C}_k \neq \mathcal{C}_i]}$ for class-wise, which is 1 if the class of $i$ is different from the class of $k$, and vice versa for $\mathbb{I}_{[\mathcal{C}_k = \mathcal{C}_i]}$. The complete loss is the sum over all sequences in a batch, where the batch-wise and class-wise components are defined as $\mathfrak{L}_{\text{bw}} = \frac{1}{B} \sum_{i=1}^{B} l_{\text{bw}}^{(i)}$ and $\mathfrak{L}_{\text{cw}} = \frac{1}{B} \sum_{i=1}^{B} l_{\text{cw}}^{(i)}$ respectively.

**Formulation 2 - Sequence element-level prediction tasks.** The previous formulation is insufficient for element-wise prediction tasks. To address this, we introduce modifications that enable the application of our contrastive loss compositions in such settings. To create element-wise positive pairs per batch element, the first two dimensions of $\mathbf{Z}$ are collapsed to form $\mathbf{Z}_{\text{flat}}, \tilde{\mathbf{Z}}_{\text{flat}} \in \mathbb{R}^{N*B \times F}$. Thus, at each position where $i = j$, the element originates from the same sequential input element. Consequently, the element-wise contrastive loss for a single sequential element becomes:

$$l_{\text{bw}}^{(i)} = -\log \frac{\exp\left(\text{sim}\left(\mathbf{z}_{\text{flat}}^{(i)}, \tilde{\mathbf{z}}_{\text{flat}}^{(i)}\right) / \tau\right)}{\sum_{k=1}^{N*B} \mathbb{I}_{[k \neq i]} \exp\left(\text{sim}\left(\mathbf{z}_{\text{flat}}^{(i)}, \tilde{\mathbf{z}}_{\text{flat}}^{(k)}\right) / \tau\right)}. \tag{3}$$

In the element-wise formulation, the class-wise positive pairs allow *intra-* and *inter-class* formulations, whereas, in Formulation 1, only *inter-class* formulations are possible. To compute the positive pairs, we need to define a left, $\mathbf{Y}_\text{l} \in \mathbb{R}^{B*N \times 1}$, and a right, $\mathbf{Y}_\text{r} \in \mathbb{R}^{1 \times B*N}$, label tensor as well as a sequence indicator tensor, $\mathfrak{S} = \lfloor \frac{i}{N*B} \rfloor$, where $i = \{0, 1, ..., N * (B - 1)\}$. Thus, the *inter-class* element-wise contrastive loss for a single sequential element becomes:

$$l_{\text{cw-inter}}^{(i)} = -\log \frac{\sum_{j=1}^{N*B} \mathbb{I}_{\text{inter}, \left[\mathbf{Y}_\text{l}^{(i,j)} = \mathbf{Y}_\text{r}^{(i,j)}\right]}^{(i,j)} \exp\left(\text{sim}\left(\mathbf{z}_{\text{flat}}^{(i)}, \tilde{\mathbf{z}}_{\text{flat}}^{(j)}\right) / \tau\right)}{\sum_{k=1}^{N*B} \mathbb{I}_{\text{inter}, \left[\mathbf{Y}_\text{l}^{(i,k)} \neq \mathbf{Y}_\text{r}^{(i,k)}\right]}^{(i,k)} \exp\left(\text{sim}\left(\mathbf{z}_{\text{flat}}^{(i)}, \tilde{\mathbf{z}}_{\text{flat}}^{(k)}\right) / \tau\right)} \tag{4}$$

where $\mathbb{I}_{\text{inter}, \left[\mathbf{Y}_\text{l}^{(i,j)} = \mathbf{Y}_\text{r}^{(i,j)}\right]}^{(i,j)}$ is 1 iff $\mathfrak{S}_i \neq \mathfrak{S}_j \wedge \mathbf{Y}_\text{l}^{(i,j)} = \mathbf{Y}_\text{r}^{(i,j)} \wedge \mathbf{Y}_\text{l}^{(i,j)} > -1 \wedge \mathbf{Y}_\text{r}^{(i,j)} > -1$, refer to Eq. (41). The *intra-class* formulation requires the cosine similarity computation between

each element of a sequence, $\text{sim}_{\text{intra}}$. We use a batch-wise matrix multiplication operator $\bigotimes_{\text{bmm}}$ : $\mathbb{R}^{B \times N \times M} \times \mathbb{R}^{B \times M \times P} \to \mathbb{R}^{B \times N \times P}$ to compute the three-dimensional similarity matrix (Eq. (42)). $\mathbf{Z}_{\text{perm}}$ and $\tilde{\mathbf{Z}}_{\text{perm}}$ are permuted equivalents of $\mathbf{Z}$ and $\tilde{\mathbf{Z}}$ fitted to the required shapes for $\bigotimes_{\text{bmm}}$. Thus, the *intra-class* element-wise contrastive loss for a single sequential element becomes:

$$l_{\text{cw-intra}}^{(i,j)} = -\log \frac{\mathbb{I}_{\text{intra},\left[\mathbf{Y}_{\text{l}}^{(i,j)}=\mathbf{Y}_{\text{r}}^{(i,j)}\right]}^{(i,j)} \exp\left(\text{sim}_{\text{intra}}\left(\mathbf{z}_{\text{perm}}^{(i)}, \tilde{\mathbf{z}}_{\text{perm}}^{(j)}\right)/\tau\right)}{\sum_{k=1}^{N} \mathbb{I}_{\text{intra},\left[\mathbf{Y}_{\text{l}}^{(i,k)}\neq\mathbf{Y}_{\text{r}}^{(i,k)}\right]}^{(i,k)} \exp\left(\text{sim}_{\text{intra}}\left(\mathbf{z}_{\text{perm}}^{(i)}, \tilde{\mathbf{z}}_{\text{perm}}^{(k)}\right)/\tau\right)} \tag{5}$$

where $\mathbb{I}_{\text{intra},\left[\mathbf{Y}_{\text{l}}^{(i,j)}=\mathbf{Y}_{\text{r}}^{(i,j)}\right]}^{(i,j)}$ is 1 iff $i \neq j \wedge \mathbf{Y}_{\text{l}}^{(i,j)} = \mathbf{Y}_{\text{r}}^{(i,j)} \wedge \mathbf{Y}_{\text{l}}^{(i,j)} > -1 \wedge \mathbf{Y}_{\text{r}}^{(i,j)} > -1$ (Eq. (43)).

For the element-wise formulation, the total batch-wise loss is $\mathfrak{L}_{\text{bw}} = \frac{1}{N*B}\sum_{i=1}^{N*B} l_{\text{bw}}^{(i)}$, whereas the total class-wise loss is $\mathfrak{L}_{\text{cw}} = \frac{1}{N*B}\sum_{i=1}^{N*B} l_{\text{cw-inter}}^{(i)} + \frac{1}{B}\sum_{i=1}^{B} \frac{1}{N}\sum_{j=1}^{N} l_{\text{cw-intra}}^{(i,j)}$.

**Formulation 3 - Sequence-level regression tasks.** This section outlines the formulation for the regression task, which necessitates scalar predictions rather than categorical classes. Consequently, only the self-supervised component, specifically the batch-wise formulation presented in the sequence-level prediction task (Eq. (1)), can be computed directly. To incorporate the supervised CL component, we generate pseudo labels by clustering the predictive target values into $K$ clusters. The parameter $k$ is a hyperparameter requiring optimization. Once the targets are clustered, each target within a cluster $k \in K$ is assigned the same pseudo-label $k$, which is subsequently employed as supervision in Eq. (2). The clustering of target values into $k$ clusters is achieved using the k-means algorithm [61].

Independent of the formulation used, we define the fused contrastive loss as the weighted sum of the batch-wise and class-wise contrastive losses:

$$\mathfrak{L}_{\text{STaR-CL}} = \lambda_{\text{fuse-CL}}\mathfrak{L}_{\text{bw}} + (1-\lambda_{\text{fuse-CL}})\mathfrak{L}_{\text{cw}}. \tag{6}$$

Finally, STaRFormer's loss is defined as the weighted sum of $\mathfrak{L}_{\text{Task}}$ and the fused contrastive loss, $\mathcal{L}_{\text{STaR-CL}}$:

$$\mathfrak{L}_{\text{STaRFormer}} = \mathfrak{L}_{\text{Task}} + \lambda_{\text{CL}}\mathfrak{L}_{\text{STaR-CL}}, \tag{7}$$

where $\lambda_{\text{CL}}$ is a tunable hyperparameter. In our experiments, we set $\lambda_{\text{fuse-CL}} = 0.5$ to equally weigh batch and class-wise similarities. For further insights, see Appendix B.3 and Figures 10 and 11.

## 4 Experiments

This work is motivated by the challenge of predicting user intent (a classification task) from non-stationary, spatiotemporal, and irregularly sampled time series. This problem can present significant difficulties for conventional modeling techniques. Our main focus is to evaluate model performance under these conditions. To ensure a robust and comprehensive assessment, we additionally employ an irregular sampled and a regular sampled time series benchmark. To demonstrate broader applicability, we extend the evaluation to additional downstream tasks, i.e., anomaly detection and regression. We compare against state-of-the-art methods to evaluate STaRFormer's effectiveness and perform exhaustive ablation studies to verify the performance gains. In [62], we present a comprehensive large-scale evaluation conducted within a federated environment.

### 4.1 Classification results

This section reports the classification results obtained across various time series domains.

#### 4.1.1 Non-stationary and spatiotemporal time series

First, we evaluate the performance on non-stationary spatiotemporal data using the DKT and Geolife (GL) [63] datasets. The DKT dataset consists of a mixture of non-stationary, spatiotemporal, and irregularly sampled time series, encompassing 559,709 labeled and anonymized customer trajectories. These trajectories were recorded from vehicles in the BMW Group's fleet over a three-month period. The associated task is intent prediction, which is framed as a binary classification problem. The DKT results in Table 1 are averaged over five seeds. We additionally use a public dataset (GL) similar to the

Table 1: Results for spatiotemporal, non-stationary time series.

| | DKT | | GL |
|---|---|---|---|
| | Accuracy ↑ | $F_{0.5}$ ↑ | Accuracy ↑ |
| RNN | 0.754 ± 0.010 | 0.754 ± 0.010 | 0.643[++] |
| TrajFormer[++] | - | - | 0.855 |
| SVM** | - | - | 0.861 |
| LSTM | 0.844 ± 0.003 | 0.843 ± 0.002 | 0.884** |
| GRU | 0.840 ± 0.003 | 0.840 ± 0.003 | 0.898** |
| ST-GRU** | - | - | 0.913 |
| Transformer | 0.849 ±0.002 | 0.849 ±0.002 | 0.881 |
| TARNet | 0.781 ±0.011 | 0.782 ±0.012 | 0.880 |
| TimesURL | 0.724 ±0.003 | - | 0.751 |
| **STaRFormer** | **0.852 ± 0.003** | **0.852 ± 0.003** | **0.932** |

Table 2: Results for irregular sampled time series (in %).

| | P19 | | P12 | | PAM | | | |
|---|---|---|---|---|---|---|---|---|
| | AUROC ↑ | AUPRC ↑ | AUROC ↑ | AUPRC ↑ | Accuracy ↑ | Precision ↑ | Recall ↑ | $F_1$-Score ↑ |
| Transformer[†] | 80.7 ± 3.8 | 42.7 ± 7.7 | 83.3 ± 0.7 | 47.9 ± 3.6 | 83.5 ± 1.5 | 84.8 ± 1.5 | 86.0 ± 1.2 | 85.0 ± 1.3 |
| Trans-mean[†] | 83.7 ± 1.8 | 45.8 ± 3.2 | 82.6 ± 2.0 | 46.3 ± 4.0 | 83.7 ± 2.3 | 84.9 ± 2.6 | 86.4 ± 2.1 | 85.1 ± 2.4 |
| GRU-D[†] | 83.9 ± 1.7 | 46.9 ± 2.1 | 81.9 ± 2.1 | 46.1 ± 4.7 | 83.3 ± 1.6 | 84.6 ± 1.2 | 85.2 ± 1.6 | 84.8 ± 1.2 |
| SeFT[†] | 81.2 ± 2.3 | 41.9 ± 3.1 | 73.9 ± 2.5 | 31.1 ± 4.1 | 67.1 ± 2.2 | 70.0 ± 2.4 | 68.2 ± 1.5 | 68.5 ± 1.8 |
| mTAND[†] | 84.4 ± 1.3 | 50.6 ± 2.0 | 84.2 ± 0.8 | 48.2 ± 3.4 | 74.6 ± 4.3 | 74.3 ± 4.0 | 79.5 ± 2.8 | 76.8 ± 3.4 |
| IP-Net[†] | 84.6 ± 1.3 | 38.1 ± 3.7 | 82.6 ± 1.4 | 47.6 ± 3.1 | 74.3 ± 3.8 | 75.6 ± 2.1 | 77.9 ± 2.2 | 76.6 ± 2.8 |
| DGM$^2$-O[†] | 86.7 ± 3.4 | 44.7 ± 11.7 | 84.4 ± 1.6 | 47.3 ± 3.6 | 82.4 ± 2.3 | 85.2 ± 1.2 | 83.9 ± 2.3 | 84.3 ± 1.8 |
| MTGNN[†] | 81.9 ± 6.2 | 39.9 ± 8.9 | 74.4 ± 6.7 | 35.5 ± 6.0 | 83.4 ± 1.9 | 85.2 ± 1.7 | 86.1 ± 1.9 | 85.9 ± 2.4 |
| Raindrop[†] | 87.0 ± 2.3 | 51.8 ± 5.5 | 82.8 ± 1.7 | 44.0 ± 3.0 | 88.5 ± 1.5 | 89.9 ± 1.5 | 89.9 ± 0.6 | 89.8 ± 1.0 |
| ViTST[†] | 89.2 ± 2.0 | 53.1 ± 3.4 | 85.1 ± 0.8 | 51.1 ± 4.1 | 95.8 ±1.3 | 96.2 ±1.3 | 96.1 ±1.1 | 96.5 ±1.2 |
| **STaRFormer** | **89.4 ±1.3** | **61.3 ±3.4** | **85.3 ±1.2** | **52.0 ±1.7** | **97.6 ±0.9** | **97.3 ±0.4** | **97.6 ±0.3** | **97.4 ±0.3** |

The model results marked with ** are taken from [17], [++] from [51] and [†] from [3].

DKT dataset to evaluate STaRFormer. Due to the environmental influences while recording GPS data, we expected some degree of non-stationary in GL [10]. KPSS and ADF tests [64, 65] confirmed that 93% of the data used for training and validation is non-stationary. Across both datasets, STaRFormer consistently outperforms state-of-the-art approaches, including TimesURL and other Transformer-based methods such as TARNet. The results are documented in Table 1. Additionally, we perform a robustness analysis with baseline models that achieve very similar performance to STaRFormer on the DKT dataset. To investigate the sensitivity of the predictions to potential sensor noise, we add noise to the last 10 and 30 elements of longer sequences in the test set of DKT and evaluate the coefficient of variation (CV). The analysis reveals that all models exhibit reduced robustness as noise increases. However, STaRFormer demonstrates superior robustness by maintaining the lowest CV and moderate MAE values, indicating minimal sensitivity to noise. In contrast, the Transformer model exhibits the most significant performance degradation, underscoring the effectiveness of our approach in learning robust latent representations that enhance downstream task performance. Refer to Appendix D.2 and Table 14 for an extended analysis.

### 4.1.2 Irregularly sampled time series

We compare STaRFormer against state-of-the-art methods designed for irregularly sampled time series on the PhysioNet Sepsis Early Prediction Challenge 2019 (P19) [66], the PhysioNet Mortality Prediction Challenge 2012 (P12) [67], and the Physical Activity Monitoring (PAM) [68] datasets. In real-world applications, particularly in healthcare, the times series data is often accompanied with static attributes. Following prior baseline methods, such as ViTST, we convert these static attributes into sentences and encode them using RoBERTa [69]. The resulting embeddings are concatenated with the latent embeddings from STaRFormer before being passed to the output head. For consistency, static features are also used in all baseline models. The results are averaged over five data splits. Across all models, STaRFormer consistently and significantly outperforms state-of-the-art baseline models on all datasets. Furthermore, STaRFormer yields predictions with significantly smaller standard deviations across all metrics than other methods, indicating greater consistency, reliability, and reduced performance variability. The results validate that our approach works particularly well for irregularly sampled time series. These findings are summarized in Table 2.

### 4.1.3 Regular time series

We evaluate STaRFormer using the UEA benchmark [70] to assess its performance on regular time series. The datasets covers a variety of domains, sensor types, sampling frequencies, number of samples, time series lengths, feature counts, and target classes for comprehensive evaluation. For the evaluation in Table 3, as not all models have reported results on the complete benchmark, we consider

Table 3: Classification results on the multivariate time series UEA benchmark (30 datasets) [70].

| | ViTST[†] | DTWD* | Weasel-Muse* | TST (TimesURL)+ | T-Loss+ | TS-TCC+ | TNC+ | TS2Vec+ | InfoTS[++] | Rocket* | Mini-Rocket* | TST (TARNet)* | InfoTS$_s$[++] | TimesURL+ | TARNet* | **STaR-Former** |
|---|---|---|---|---|---|---|---|---|---|---|---|---|---|---|---|---|
| Avg. Accuracy ↑ | 0.790 | 0.608 | 0.691 | 0.617 | 0.658 | 0.668 | 0.670 | 0.704 | 0.714 | 0.715 | 0.719 | 0.729 | 0.730 | 0.752 | 0.755 | **0.795** |
| Rank ↓ | - | - | - | 13 | 12 | 11 | 10 | 9 | 8 | 7 | 6 | 5 | 4 | 3 | 2 | **1** |
| Avg. Rank ↓ | - | - | - | 10.6 | 8.6 | 9.2 | 9.9 | 7.4 | 6.8 | 5.5 | 5.7 | 6.5 | 5.3 | 3.9 | 4.9 | **2.8** |
| Top Scores ↑ | 1 | 0 | 5 | 1 | 1 | 1 | 0 | 1 | 1 | 5 | 4 | 6 | 3 | 4 | 7 | **9** |
| 1-v-1 ↑ | 8 | 28 | 20 | 29 | 27 | 27 | 29 | 25 | 27 | 19 | 22 | 23 | 23 | 23 | 21 | - |
| DS Count | 10 | 29 | 28 | 30 | 30 | 30 | 30 | 30 | 30 | 30 | 30 | 30 | 30 | 30 | 30 | 30 |
| Accuracy 28 ↑ | - | 0.604 | 0.691 | 0.631 | 0.675 | 0.680 | 0.677 | 0.713 | 0.722 | 0.730 | 0.733 | 0.724 | 0.738 | 0.760 | 0.770 | **0.793** |
| Rank 28 ↓ | - | 15 | 10 | 14 | 13 | 11 | 12 | 9 | 8 | 6 | 5 | 7 | 4 | 3 | 2 | **1** |
| Avg. Rank 28 ↓ | - | 11.2 | 7.8 | 11.7 | 9.1 | 10.3 | 11.0 | 8.1 | 7.5 | 5.8 | 6.0 | 7.5 | 5.8 | 4.1 | 5.2 | **3.1** |
| Accuracy 9 ↑ | 0.776 | 0.702 | 0.737 | 0.674 | 0.717 | 0.708 | 0.715 | 0.734 | 0.727 | 0.756 | 0.751 | 0.771 | 0.736 | 0.770 | 0.717 | **0.793** |
| Rank 9 ↓ | 2 | 15 | 7 | 16 | 12 | 14 | 13 | 9 | 10 | 5 | 6 | 3 | 8 | 4 | 11 | **1** |
| Avg. Rank 9 ↓ | 6.4 | 11.8 | 9.0 | 12.3 | 11.1 | 11.3 | 10.8 | 9.0 | 10.0 | 6.7 | 7.4 | 3.9 | 8.4 | 5.3 | 6.3 | **3.3** |

The model results marked with * are taken from the [21], [+] from [27], [++] from [26] and [†] from [3].

three splits of the benchmark depending on the results available in literature (UEA, UEA 28 and UEA 9). The summarized scores of the complete results (Table 30) across the UEA benchmark are displayed in Table 3. STaRFormer achieves the highest accuracy in all three splits (**0.795**), improving the state-of-the-art on the complete benchmark by **4.0** percentage points; the largest number of top scores (**9**); and the best average rank (**2.8**). Furthermore, STaRFormer performs better on UEA datasets with only a few samples, achieving top scores for DDK, NT, PS, SCP2 and SWJ for example. This suggests its capability as an augmentation technique, especially for lower data regimes.

## 4.2 Anomaly detection results

In this setting, we adopt the streaming evaluation protocol proposed in [71] and utilized by [24, 27]. The model performance is evaluated on the KPI [71] and Yahoo [72] benchmark datasets and compared against several state-of-the-art approaches, such as TimesURL and TS2Vec.

Table 4: Anomaly detection results (univariate).

|  | Yahoo | | | KPI | | |
|---|---|---|---|---|---|---|
|  | $F_1$ ↑ | Precision ↑ | Recall ↑ | $F_1$ ↑ | Precision ↑ | Recall ↑ |
| SPOT | 0.338 | 0.269 | 0.454 | 0.217 | 0.786 | 0.126 |
| DSPOT | 0.316 | 0.241 | 0.458 | 0.521 | 0.623 | 0.447 |
| DONUT | 0.026 | 0.013 | 0.825 | 0.347 | 0.371 | 0.326 |
| SR | 0.563 | 0.451 | 0.747 | 0.622 | 0.647 | 0.598 |
| TS2Vec | 0.745 | 0.729 | 0.762 | 0.677 | **0.929** | 0.533 |
| TimesURL | 0.749 | 0.748 | 0.750 | 0.688 | 0.925 | 0.546 |
| **STaRFormer** | **0.789** | **0.772** | **0.807** | **0.830** | 0.852 | **0.811** |

Each time series is split chronologically, where the first half is used for training and the second for evaluation. To facilitate efficient computation, we choose to segment sequences into fixed-size windows, allowing overlap between these segments during training. STaRFormer demonstrates superior performance across both datasets in the benchmark, as shown in Table 4.

## 4.3 Time series extrinsic regression results

Table 5: Regression results on the TSR benchmark (19 datasets) [73] reported in RMSE.

|  | FPCR∗ | SVR Optimised∗ | Random Forest∗ | XG-Boost∗ | 5-NN-ED∗ | 5-NN-DTWD∗ | Rocket∗ | FCN∗ | Res-Net∗ | Inception∗ | TAR-Net | **STaR-Former** |
|---|---|---|---|---|---|---|---|---|---|---|---|---|
| Avg. Rel. Mean Difference ↓ | 0.028 | 0.208 | -0.121 | -0.132 | 0.051 | -0.034 | -0.245 | -0.160 | -0.119 | -0.220 | 0.170 | **-0.254** |
| Avg. Rel. Mean Difference Rank ↓ | 9 | 12 | 6 | 5 | 10 | 8 | 2 | 4 | 7 | 3 | 11 | **1** |
| Top Scores ↑ | 1 | 0 | 0 | 4 | 0 | 0 | 7 | 0 | 0 | 3 | 0 | **9** |

The model results marked with * are taken from the official benchmark (`http://tseregression.org/`).

The results of the Time Series Extrinsic Regression (TSR) benchmark are summarized in Table 5. The complete results can be found in Table 33. The results present the Root Mean Squared Error (RMSE) of scalar regression predictions produced by each model. In order to facilitate a comparative analysis of the models within this benchmark, we adhere to the evaluation metric established in [23], referred to as the average relative mean difference (Eq. (48)). This metric quantitatively evaluates the deviation of each model from the mean RMSE for each dataset. Therefore, superior model performance is indicated by increasingly negative values of the metric, whereas inferior performance corresponds to less negative and positive values. We implemented TARNet and utilize the model configurations provided by the authors, where available, for the respective datasets. Across the entirety of the benchmark, STaRFormer consistently achieves the greatest relative mean difference among all models, alongside the largest number of top scores.

## 4.4 What contributes to STaRFormer's performance?

This section investigates the source of STaRFormer's performance gains through empirical validation.

### 4.4.1 STaRFormer architecture ablation

To demonstrate the performance gains achieved by DAReM paired with the semi-supervised CL in STaRFormer, we train two ablations of STaRFormer: an encoder-only Transformer (Base), and STaRFormer with Random Masking (RM). We select the datasets from Sections 4.1.1, 4.1.2, and 4.2, in addition to a representative set from the UEA benchmark (see Appendix C.1.3), for a total of 19 datasets. STaRFormer outperforms STaRFormer-RM in 16 out of the 19 selected datasets, while STaRFormer-RM outperforms STaRFormer only in one dataset, verifying that DAReM significantly

Table 6: STaRFormer archi-tecture ablation results on 19 datasets.

| | Base | STaRFormer-RM | STaRFormer |
|---|---|---|---|
| Avg. Acc. | 0.824 | 0.826 | **0.841** |
| Rank | 3 | 2 | **1** |
| Avg Rank | 2.1 | 2.5 | **1.2** |
| Top Scores | 5 | 2 | **15** |
| **1-v-1** | | | |
| Base | - | 12 | 3 |
| RM | 6 | - | 1 |
| STaRFormer | 14 | 16 | - |

Table 7: Ablation results for semi-supervised CL and DAReM.

| CL Method | DKT ($\lambda_{CL} \approx 0.796$) | | GL ($\lambda_{CL} \approx 0.773$) | | PAM ($\lambda_{CL} \approx 0.567$) | | | |
|---|---|---|---|---|---|---|---|---|
| | Accuracy ↑ | $F_{0.5}$ ↑ | Accuracy ↑ | $F_{0.5}$ ↑ | Accuracy ↑ | Precision ↑ | Recall ↑ | $F_1$ ↑ |
| semi[1] | **85.2 ± 0.3** | **85.2 ± 0.3** | 90.4 ± 1.6 | 88.3 ± 1.9 | **97.6 ± 0.9** | 97.3 ± 0.4 | **97.6 ± 0.3** | 97.4 ± 0.3 |
| w/o self[1] | 84.8 ± 0.2 | 84.8 ± 0.2 | 90.0 ± 1.4 | 87.7 ± 1.5 | 96.2 ± 2.1 | 97.1 ± 0.7 | 97.0 ± 0.7 | 97.0 ± 0.7 |
| w/o sup[1] | 84.8 ± 0.1 | 84.7 ± 0.2 | 89.5 ± 1.6 | 87.7 ± 1.4 | 96.5 ± 1.6 | 97.5 ± 0.3 | 97.4 ± 0.5 | 97.4 ± 0.3 |
| semi[1]: | | | | | | | | |
| $\lambda_{CL} = 0.1$ | 84.8 ± 0.1 | 84.6 ± 0.4 | 90.0 ± 1.8 | 87.9 ± 2.1 | 96.2 ± 1.4 | 96.6 ± 0.6 | 96.8 ± 1.0 | 96.7 ± 0.8 |
| $\lambda_{CL} = 1$ | 85.1 ± 0.2 | 85.1 ± 0.2 | 90.2 ± 1.3 | 88.0 ± 1.5 | 97.2 ± 0.7 | 97.4 ± 0.3 | 97.2 ± 0.7 | 97.3 ± 0.4 |
| $\lambda_{CL} = 5$ | 84.9 ± 0.2 | 84.9 ± 0.2 | **90.8 ± 1.3** | **88.7 ± 1.6** | 96.7 ± 2.3 | 97.5 ± 1.2 | 97.0 ± 1.7 | 97.2 ± 1.5 |
| $\lambda_{CL} = 10$ | 84.6 ± 0.3 | 84.6 ± 0.3 | 90.6 ± 1.0 | 88.5 ± 1.3 | 97.0 ± 1.6 | **97.7 ± 0.7** | 97.6 ± 0.7 | **97.6 ± 0.7** |

| $\gamma$ | DKT ($\varphi \approx 0.427, \zeta = 0.2$) | | GL ($\varphi \approx 0.472, \zeta = 0.3$) | | PAM ($\varphi \approx 0.207, \zeta = 0.3$) | | | |
|---|---|---|---|---|---|---|---|---|
| | Accuracy ↑ | $F_{0.5}$ ↑ | Accuracy ↑ | $F_{0.5}$ ↑ | Accuracy ↑ | Precision ↑ | Recall ↑ | $F_1$ ↑ |
| 0.00 | 85.0 ± 0.2 | 85.0 ± 0.2 | 89.8 ± 1.9 | 87.9 ± 1.8 | 97.0 ± 0.7 | 97.4 ± 0.2 | 97.3 ± 0.6 | 97.3 ± 0.3 |
| 0.05 | 85.0 ± 0.3 | 84.8 ± 0.3 | **90.4 ± 1.6** | 88.3 ± 1.9 | 95.8 ± 1.6 | 96.8 ± 0.8 | 96.8 ± 0.7 | 96.7 ± 0.7 |
| 0.10 | 84.9 ± 0.3 | 84.9 ± 0.2 | 90.3 ± 1.2 | 88.2 ± 1.3 | **97.6 ± 0.9** | 97.3 ± 0.4 | **97.6 ± 0.3** | 97.4 ± 0.3 |
| 0.15 | 85.0 ± 0.2 | 85.0 ± 0.2 | 90.3 ± 1.5 | 88.2 ± 1.7 | 97.1 ± 1.1 | **97.5 ± 0.6** | 97.5 ± 1.0 | **97.5 ± 0.8** |
| 0.20 | 85.1 ± 0.1 | 85.1 ± 0.1 | 90.1 ± 1.1 | 87.9 ± 1.0 | 96.2 ± 0.8 | 96.7 ± 0.5 | 96.6 ± 0.6 | 96.6 ± 0.4 |
| 0.25 | **85.2 ± 0.3** | **85.2 ± 0.3** | 90.1 ± 1.6 | **88.4 ± 1.4** | 96.4 ± 1.1 | 96.9 ± 0.7 | 96.5 ± 0.6 | 96.7 ± 0.5 |
| 0.30 | 85.0 ± 0.1 | 85.0 ± 0.1 | 90.3 ± 1.3 | 88.2 ± 1.4 | 96.3 ± 0.9 | 96.7 ± 0.5 | 96.4 ± 0.5 | 96.5 ± 0.4 |

enhances the robustness of the model compared to RM, see Table 6. STaRFormer achieves the highest average accuracy (**0.841**), surpassing the two ablation variants by **1.5** and **1.7** percentage points respectively; the highest number of top scores (**15**); and best average rank (**1.2**).

#### 4.4.2 Impact of semi-supervised contrastive learning and regional masking

**Impact of semi-supervised CL.** We examine the effect of different components of the semi-supervised CL in STaRFormer by removing the respective components from the loss function. The results in Table 7 show the advantages of maximizing agreement between both batch-wise and class-wise representations in CL. The downstream task performance (accuracy) declined by **0.4** to **1.4** percentage points when these representations were not fused in the CL approach. There is no consistent trend favoring one representation over the other; e.g., in GL, supervised CL outperformed self-supervised CL, while the opposite holds for PAM. In DKT, both methods yield comparable results. Additionally, we study the impact of combining the contrastive loss $\mathcal{L}_{STaR-CL}$ and the task loss $\mathcal{L}_{Task}$ via $\lambda_{CL}$, Eq. (7). The scale difference between $\mathcal{L}_{Task}$ and $\mathcal{L}_{STaR-CL}$ is approximately a factor of 10 across all datasets. Consequently, values of $\lambda_{CL} > 0.1$ assign greater weight to $\mathcal{L}_{STaR-CL}$, thus increasing its impact on the overall loss and the model updates during backpropagation. Our results indicate that higher values of $\lambda_{CL}$ lead to improved performance, with all top scores achieved at $\lambda_{CL} > 0.1$ (in some cases, large weights of 5 and 10 yielded best results). These results indicate that emphasizing context-aware representation learning during the training of a downstream task can enhance the overall performance on this task, supporting our approach.

**Impact of regional masking.** To examine the impact and benefit of masking regions with DAReM, we perform an one-at-a-time analysis (OAT), where we iteratively change the region defining parameter, $\gamma$, while keeping all other parameters fixed. We expect better performance when using larger masking regions ($\gamma > 0$) around the top-$k$ compared to only masking the top-$k$ important sequential elements ($\gamma = 0$), which is essentially the masking approach in TARNet. However, excessively large masked regions may degrade performance by limiting the informative context for reconstruction. The observed trend indicates on a macro scale that masking larger regions enhances the performance of STaRFormer. Masking regions larger or equal to 10% of the global sequence length around the selected elements achieves top scores for 7 out of 8 metrics (Table 7 bottom section). On a micro-scale, performance peaks were observed at different optimal configurations with the best performance for DKT at $\gamma = 0.25$, whereas for GL and PAM, the performance peaks were found for smaller region masks. Further increasing or decreasing the masked regions gradually deteriorated the results, supporting our initial hypothesis.

#### 4.4.3 Latent space analysis

To evaluate the hypothesis that enhancing the latent space embedding improves prediction performance, we analyzed t-SNE visualizations [74] of four datasets (test sets only) where STaRFormer outperforms the Base ablation. Thus, we compare the latent space representation of Base and STaRFormer. As shown in Fig. 2, the t-SNE visualizations reveal that while Base achieves some degree of class separation, STaRFormer consistently produces distinct and well-separated clusters for each

---
[1]semi = semi-supervised, self = self-supervised and sup = supervised

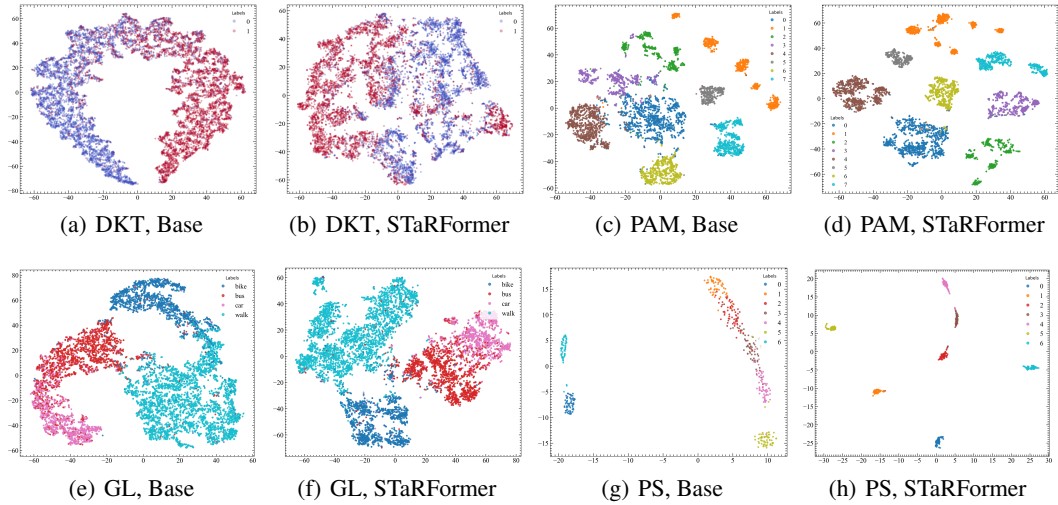

Figure 2: t-SNE visualizations (plotted with perplexity 50) of latent spaces representations for the DKT (a, b), PAM (c, d), GL (e, f), and PS (UEA) (g, h) datasets, comparing Base and STaRFormer.

class across all datasets (classes are color-coded). In DKT, the latent embeddings for both models show overlap between class clusters. However, STaRFormer can more distinctly separate clusters between the classes, whereas Base has one significant area of overlap. This trend is amplified by the observations in PAM and GL, where STaRFormer displays more distinct clusters with minimal overlap compared to Base. In GL, clusters for 'walk' and 'bike' as well as 'car' and 'bus' are distinctly separated. However, distinguishing between 'car' and 'bus' remains challenging due to similar traveling speeds and trajectories. In PS, the most pronounced difference is obtained with Base producing scattered clusters with significant overlap, while STaRFormer achieves distinctly separated clusters. These results align with the test accuracy reported in Table 31, where the accuracy difference between Base and STaRFormer is most significant on PS. In summary, the clusters from both models appear more similar for datasets with similar test accuracies. STaRFormer creates more discriminative latent representations, i.e., enhanced class separation, which, considering the improved test accuracy, leads to improved classification performance over the Base ablation. For datasets where our CL approach is very effective, e.g., PS, the improvement through our approach is clearly visualized in the t-SNE visualizations.

## 5 Limitations and conclusion

We propose a task-coupled semi-supervised CL technique that jointly optimizes representation learning with the downstream objective, embedding task-specific information into the representations. By integrating embeddings that are generated from masked (DAReM) and unmasked sequences, the semi-supervised CL exploits both batch-wise (self-supervised) and class-wise (supervised) similarities to achieve improved task-specific representations for predictions on various downstream tasks. Comprehensive experiments demonstrate that STaRFormer either surpasses or is on par with state-of-the-art techniques for various time series types. We verify this performance on 55 benchmark datasets and real-world data from the BMW Group. Notable limitations include: the computational overhead of the attention-based masking with $\mathcal{O}(N^2)$ complexity, especially for long sequences, and the additional increase in training time and complexity due to CL and DAReM, which however does not affect inference time. Additionally, the task-coupled nature of STaRFormer results in a further limitation constraining its flexibility compared to task-agnostic models such as TimesURL or InfoTS. These models aim to learn universal representations that, theoretically, can be utilized across a variety of downstream tasks without the need of training from scratch. Future work could explore more efficient attention mechanisms, such as flash attention, to enhance the scalability and efficiency of STaRFormer for long sequences and large-scale datasets. Moreover, exploring a task-agnostic implementation of STaRFormer could substantially enhance its flexibility.

## Acknowledgments

This project is supported and funded by the BMW Group and has received partial funding from the European Union's Horizon research and innovation programme (Grant Agreement No. 101159667). Any opinions, findings, conclusions or recommendations expressed herein are those of the authors. They should not be interpreted as necessarily representing the views, either expressed or implied, of the BMW Group and its affiliates or the European Commission. Neither the European Commission nor the BMW Group is responsible for any use that may be made of the information contained herein.

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

# Glossary

$\gamma$  Determines the bound of the region to be masked. 4, 9, 39

**X**  Mini-batch. 3, 4, 32, 39–41

$\sigma$  Attention scores computed from the attention weights $\tilde{\mathbf{A}}$. 4, 40

$\varphi$  Number of Elements that can be masked. 4, 39, 40

$\zeta$  Determines the number of sequential elements that are masked based on the attention scores $\sigma$. 4, 39

**ADF**  Augmented Dickey-Fuller. 1, 7, 31, 32

**AE**  Appliances Energy. 43, 53, 54, 62

**AF**  Atrial Fibrillation. 43, 53, 60

**AR**  Australia Rainfall. 43, 53, 54, 62

**AWR**  Articulary Word Recognition. 43, 53, 60

**BC**  Benzene Concentration. 43, 53, 54, 62

**BCE**  Binary Cross-Entropy. 38

**BIDMCHR**  BIDMC32HR. 43, 53, 54, 62

**BIDMCRR**  BIDMC32RR. 43, 53, 54, 62

**BIDMCSPO2**  BIDMC32SpO2. 43, 53, 54, 62

**BLE**  Bluetooth Low-Energy. 1, 31

**BM**  Basic Motions. 43, 53, 60

**BPM10**  Beijing PM10 Quality. 43, 53, 54, 62

**BPM25**  Beijing PM25 Quality. 43, 53, 54, 62

**C3M**  Covid3Month. 43, 53, 54, 62

**CCC**  Car Connectivity Consortium. 31

**CE**  Cross-Entropy. 38

**CK**  Cricket. 43, 53, 60

**CL**  Contrastive Learning. 1–6, 8–10, 38–42, 55, 57, 58

**CNN**  Convolutional Neural Network. 2, 34

**CoST**  Contrastive Learning of Disentangled Seasonal-Trend Representations for Time Series Forecasting. 3

**CT**  Character Trajectories. 43, 53, 60

**DAReM**  Dynamic Attention-based Regional Masking. 1, 2, 4, 5, 8–10, 38, 39, 42, 55, 57–60

**DDK**  Duck Duck Geese. 8, 43, 53, 60

**DK**  Digital Key. 1, 31, 43

**DKT**  Digital Key Trajectories. 1, 2, 4, 6, 7, 9, 10, 23, 26, 31, 32, 42–44, 47–51, 55, 56, 61

**DL**  Deep Learning. 2, 34, 37

**DTWD**  Dimension-dependent dynamic time warping. 2, 7, 60

**EC**  Ethanol Concentration. 43, 46, 52, 60, 61

**EP**  Epilepsy. 43, 53, 60

**ER**  ERing. 43, 53, 60

**EW**  Eigen Worms. 43, 46, 52, 60, 61

**FD**  Face Detection. 43, 46, 52, 60, 61

# Appendix - Supplementary Material

## A  Localization and Tracking via Ultra-Wideband Technology and the Digital Key

The digital key (DK) enables the use of a smart device as a 'vehicle key' [5, 7], facilitating handsfree or passive access to a vehicle through the smart device. The DK technology is standardized by the Car Connectivity Consortium (CCC), led by Apple, the BMW Group, Ford, Google, Mercedes, Xiaomi, and other global corporations [9]. In recent years, car manufacturers have started to incorporate Ultra-Wideband (UWB) and Bluetooth Low-Energy (BLE) technologies to enhance the capabilities of the DK [4–8]. This allows for precise and secure vehicle access while paving the way for the creation of additional applications for connected vehicles.

### A.1  Localization - Non-Stationary Characteristics

A Bluetooth connection is initially established between the smart device and the vehicle to detect a paired personal smart device nearby. Following the exchange of security protocols, an UWB connection is set up to enable secure ranging of the smart device. The vehicle is equipped with multiple UWB anchors. Between each UWB anchor and the smart device, time-of-flight measurements are executed, allowing precise localization due to UWB's pulse duration of $2ns$ [9]. When the localization is recorded, one is able to track the smart device around the vehicle, enabling intent predictions based on the sequentially collected localization measurements. However, various external factors can influence the localization accuracy, including materials of different vehicle models, external environments like weather conditions, interference from other signals, and the position of the smart device (e.g., in hand, front pocket, or handbag). These interferences can introduce non-stationary characteristics to the sequential data. To verify this hypothesis, we compute the Kwiatkowski–Phillips–Schmidt–Shin (KPSS) and augmented Dickey-Fuller (ADF) tests [64, 65]. We consider a time series non-stationary if both tests agree, i.e., the $p$-values for KPSS are smaller than a

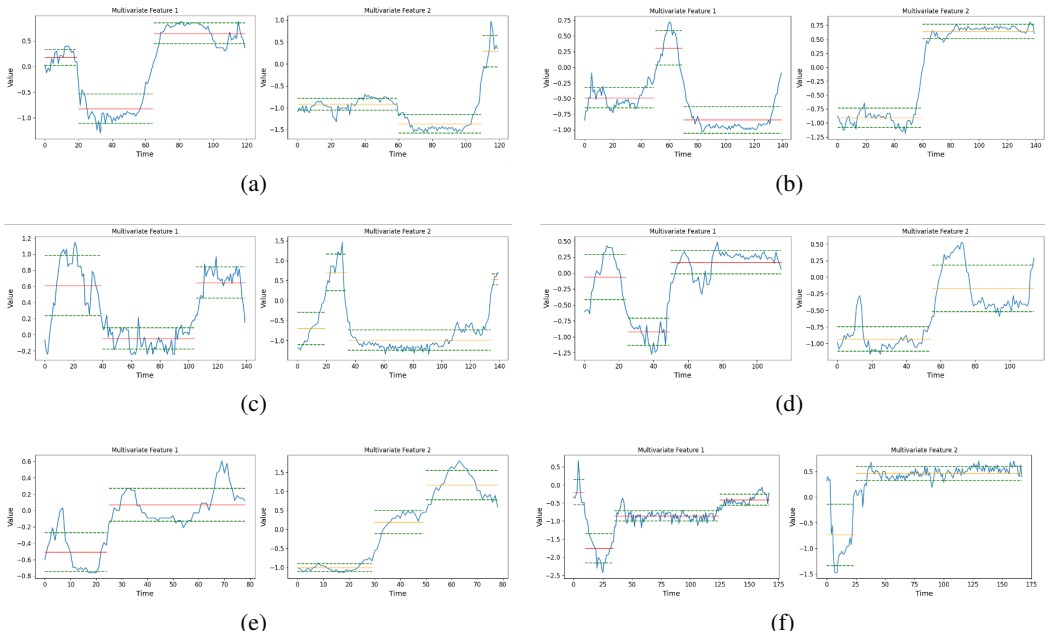

Figure 3: Example plots visualizing the non-stationary characteristics of the sequential data in the DKT dataset. The red or orange line visualizes the mean and the green dashed lines the standard deviation of a segment. Multiple mean and standard deviation lines per plot indicate changes in the underlying generative distribution of the visualized data. These plots only serve as a demonstration and visualization of the non-stationary characteristics of data samples from the DKT dataset.

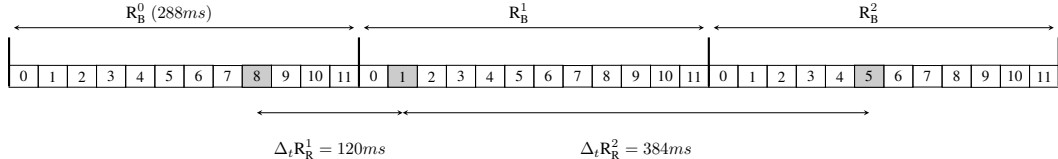

Figure 4: An illustration demonstrating the collection and utilization of UWB signal measurements to localize a smart device around the vehicle using a $\text{multiplier}_{\text{RAN}} = 3$. Due to the continuous hopping strategy for ranging, fixed ranging round indices are set before the data is collected. In this case, ranging round indices [9, 2, 6] lead to a difference in the time delta between three following $R_R$'s, e.g., $|\Delta_t R_R^1 - \Delta_t R_R^2| = 264ms$ [9].

significance level, i.e., $p_{\text{KPSS}} < 0.05$, whereas the $p$-values for ADF are larger than a significance level, i.e., $p_{\text{ADF}} > 0.05$. The results suggest that 79% of the data provided to us by the BMW Group used for training and validation is indeed non-stationary. Fig. 3 depicts six examples visually, which display non-stationary characteristics of the time series data in the DKT.

## A.2  UWB Ranging and Measuring

The measuring algorithm employed for ranging introduces another level of uncertainty. The UWB signal collection for localization occurs within a time window referred to as ranging block. A ranging block, $R_B$, is defined as $R_B = \text{multiplier}_{\text{RAN}} \times T_{\min}$, where $\text{multiplier}_{\text{RAN}} \in \{1, \ldots, 255\}$ and $T_{\min} = 96ms$ [9]. The $R_B^i$, where $i = 0, 1, \ldots, N - 1$, is divided into $M$ ranging rounds $R_R^j$, where $j = 0, 1, \ldots, M - 1$. Although these $R_R$ are fixed, the specific $R_R^j$-index used for recording varies per ranging block, to reduce the probability of interference in multi-device scenarios. Different strategies have been proposed for selecting a $R_R^j$-index; in the data provided by the BMW Group, a continuous strategy is employed such that the $R_R^j$-index for each $R_B^i$ is predetermined. However, the differences between consecutive $R_R^j$'s, $\Delta_t R_R^j = R_R^{j+1} - R_R^j$, do not necessarily match the differences in the following window, $\Delta_t R_R^{j+1} = R_R^{j+2} - R_R^{j+1}$ [9]. Fig. 4 visualizes the UWB ranging algorithm and the resulting irregular sampled sequential data conceptually.

# B  Approach

**Notation**. We extend the formulation from Section 3. In this work, the following notation is used: tensors and matrices are represented by capital letters $\mathbf{A}$ and column vectors are represented as $\mathbf{a}$. Given a matrix $\mathbf{A}$, one can access the $i$-th, $j$-th element as $\mathbf{A}_{i,j}$. To access a row of $\mathbf{A}$, one can slice the matrix $\mathbf{A}_{i,:} \equiv \mathbf{a}^{(i)}$ and vice versa $\mathbf{A}_{:,i}$ to access a column. For a 3-D tensor $\mathbf{A}$, element $(i, j, k)$ is $\mathbf{A}_{i,j,k}$. The $i$-th element of column vector $\mathbf{a}$ is $a_i$. Scalars are depicted as $a$. The dataset used to train a machine learning algorithm is noted as $\mathcal{D}$, where in general $(\mathbf{x}^{(i)}, y^{(i)})$ is the $i$-th sample-label pair of $\mathcal{D}$ in the supervised setting. $\hat{y}$ denotes the predicted labels by a function $f$. As previously defined, a mini-batch $(\mathbf{X})$, $\mathbf{X} \subset \mathcal{D}$ and $\mathbf{X} \in \mathbb{R}^{N \times B \times D}$ is a 3-D tensor where $B$ is the batch-size, $N$ the sequence length $D$ the feature dimension and the $B \ll M$. Hence, one can access the $i$-th element of the mini-batch as $\mathbf{X}_{:,i,:}$, which is the same as $\mathbf{S}^{(i)}$. A set is denoted as $\mathbb{A}$, where $\mathbb{R}$ is the set of real numbers for example. Special notations are the indicator function, $\mathbb{I}$,

$$\mathbb{I}_{[x \in \mathbb{A}]}(x) := \begin{cases} 1 \text{ if } x \in \mathbb{A} \\ 0 \text{ if } x \notin \mathbb{A} \end{cases} \tag{8}$$

and the identity matrix $\mathbf{I}_n$, a $(n \times n)$-matrix. The symbol $\odot$ is the Hadamard (elementwise) product operator, and the symbol $\otimes$ is a tensor product operator.

## B.1  Baseline-Models

In machine learning, sequential data requires specific modeling techniques because the order and context of the data points significantly influence the overall meaning and patterns, necessitating models that can effectively capture and utilize temporal dependencies and relationships. The sequences'

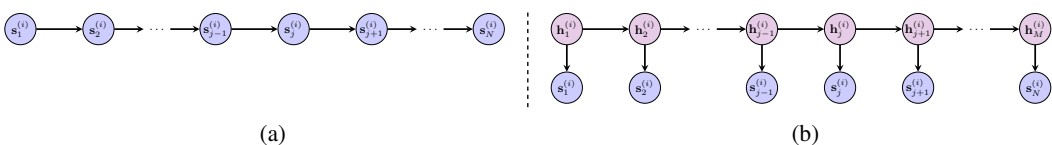

$$(a) \qquad\qquad\qquad\qquad (b)$$

Figure 5: Illustration of (a) a sequence using a first-order Markov chain and (b) a sequence using a Markov chain of latent variables.

intrinsic order is crucial for conducting analysis and making predictions. Therefore, the generally applicable independent and identically distributed (i.i.d) assumption, is not suitable [43], as a current state in a sequence depends on its preceding states. Hence, to accurately model a sequence, one would aim to compute the joint probability of all elements in the sequence, i.e.,

$$P\left(\mathbf{s}_1^{(i)}, \mathbf{s}_2^{(i)}, \dots, \mathbf{s}_N^{(i)}\right) = P(\mathbf{s}_1^{(i)}) \prod_{j=2}^{N} P\left(\mathbf{s}_j^{(i)} \mid \mathbf{s}_1^{(i)}, \mathbf{s}_2^{(i)}, \dots, \mathbf{s}_{j-1}^{(i)}\right). \tag{9}$$

Naturally, recent observations will most likely provide more insights into future predictions than historically older observations. Additionally, it is not feasible to assume that future observations depend on all past observations [43]. The simplest formulation, Fig. 5(a), and hence ignoring the intrinsic order of the sequence, applies the Markov assumption stating a new state is only dependent on the current state, i.e.,

$$\mathbf{s}_{j+1}^{(i)} \perp\!\!\!\perp \mathbf{s}_{j-1}^{(i)} \mid \mathbf{s}_j^{(i)}. \tag{10}$$

Therefore, the joint probability is

$$P\left(\mathbf{s}_1^{(i)}, \mathbf{s}_2^{(i)}, \dots, \mathbf{s}_N^{(i)}\right) = P(\mathbf{s}_1^{(i)}) \prod_{j=2}^{N} P\left(\mathbf{s}_j^{(i)} \mid \mathbf{s}_{j-1}^{(i)}\right). \tag{11}$$

To enable prior observations to impact the modeling, one can transition to utilizing higher-order Markov chains, considering a greater number of preceding states. However, this will lead to an exponentially growing number of parameters the model requires, rendering it impractical.

To consider the more complex intrinsic order in sequences, a latent variable model that is not limited by the Markov assumption can be used, refer to Fig. 5(b). A latent variable model permits the creation of a rich model out of simple components. In this approach, a latent variable or hidden state, $\mathbf{h}_{j-1}^{(i)}$, stores the information of the sequential steps up to $j-1$, while still satisfying the conditional independence property [43], $\mathbf{h}_{j+1}^{(i)} \perp\!\!\!\perp \mathbf{h}_{j-1}^{(i)} \mid \mathbf{h}_j^{(i)}$, such that the joint probability distribution is

$$P\left(\mathbf{s}_1^{(i)}, \mathbf{s}_2^{(i)}, \dots, \mathbf{s}_N^{(i)}, \mathbf{h}_1^{(i)}, \mathbf{h}_2^{(i)}, \dots, \mathbf{h}_N^{(i)}\right) = P\left(\mathbf{h}_1^{(i)}\right) \left[\prod_{j=2}^{N} P\left(\mathbf{h}_j^{(i)} \mid \mathbf{h}_{j-1}^{(i)}\right)\right] \prod_{j=1}^{N} P\left(\mathbf{s}_j^{(i)} \mid \mathbf{h}_j^{(i)}\right). \tag{12}$$

Intuitively, when examining a sequence, one goal might be to predict the next value that might occur in the sequence. This can be achieved, for example, by evaluating the expected value of the likelihood of a new state $\mathbf{s}_j^{(i)}$. Let's define random variables for the subsequent state $(\mathbf{Y}^{(i)})$ as $\mathbf{Y}^{(i)} = \mathbf{s}_j^{(i)}$ and the sequence of preceding states $(\mathbf{X}^{(i)})$ as $\mathbf{X}^{(i)} = \mathbf{s}_{j-1}^{(i)}, \mathbf{s}_{j-2}^{(i)}, \dots, \mathbf{s}_1^{(i)}$.

$$\mathbb{E}\left[\mathbf{Y}^{(i)} \mid \mathbf{X}^{(i)}\right] = \mathbb{E}\left[P\left(\mathbf{s}_j^{(i)} \mid \mathbf{s}_{j-1}^{(i)}, \mathbf{s}_{j-2}^{(i)}, \dots, \mathbf{s}_1^{(i)}\right)\right] \tag{13}$$

Then, for example, a linear regression model can be employed to estimate the conditional expectation, $\mathbb{E}$, as follows,

$$\hat{y}^{(i)} = \hat{\mathbb{E}}\left[\mathbf{Y}^{(i)} \mid \mathbf{X}^{(i)}\right] \approx f(\mathbf{X}^{(i)}; \Theta) + \epsilon^i, \tag{14}$$

where $\epsilon$ is Gaussian white noise, $\mathcal{N}(0, \beta^{-1})$ [43]. The model $f$,

$$f(\mathbf{X}^{(i)}; \Theta) = \sum_{j=1}^{N} \theta^T x_j^{(i)} + \epsilon^{(i)} \tag{15}$$

is a linear combination of its parent nodes, which is known as an autoregressive model [75].

### B.1.1 Recurrent Neural Network (RNN)

Elman [32] developed a unique modeling technique, referred to as RNN, which is specifically designed to capture and utilize the temporal dependencies and relationships inherent in such data. Note that RNNs are increasingly being replaced by Transformer-based architectures, which can be more efficient at processing sequential data due to their parallel processing capabilities.

A RNN [31, 32] is a deep learning model that is trained to process a sequential input and convert it into a specific sequential output. Sequential data refers to data, such as words, sentences, or time series data, where sequential components are linked together based on complex semantic and syntactic rules. RNNs manage sequence dynamics through recurrent connections, which function like cycles within the network recursively evaluating the sequential elements. These recurrent connections are unrolled across sequential steps, applying the same parameters at each step [76], as illustrated in Fig. 6(a). While standard connections propagate activations synchronously within the same sequential steps, recurrent connections transmit information across adjacent sequential steps, also shown in Fig. 6(a). RNNs can be seen as feed-forward networks or multilayer perceptrons (MLPs) with shared parameters across sequential steps, typically representing steps in time. Sequentiality is not exclusive to RNNs; for instance, Convolutional Neural Networks (CNNs) can be adapted for data with varying lengths, such as images of different resolutions. Although RNNs have recently been overshadowed by Transformer models, they remain essential for complex sequential modeling. For a more detailed discussion on RNNs, refer to the comprehensive reviews by [76] and [77].

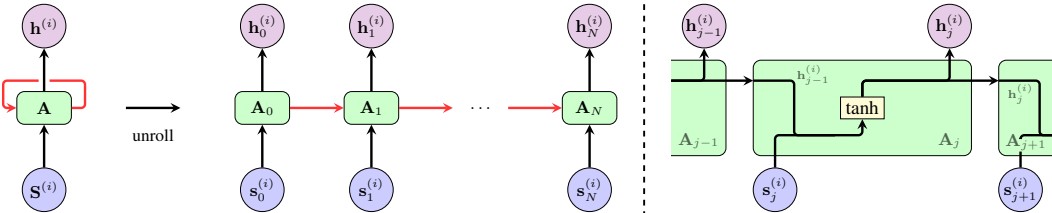

(a) Unrolling computational cycles in a RNN.      (b) Computational logic in a RNN cell.

Figure 6: Recurrent Neural Network: (a) illustrates the high level computational concept of cycles in the networks used in RNN and how they are unrolled. Recurrent connections are highlighted in red. (b) illustrates the computational logic of each cycle applied as a neural network for an adjacent element of the input sequence.

Adopting a Deep Learning (DL) perspective, the sequence in Fig. 5(a) could also be considered a computational graph, for which Eq. (16) describes the recurrent or recursive computation [76].

$$\mathbf{s}_j^{(i)} = f(\mathbf{s}_{j-1}^{(i)}; \Theta_j) \tag{16}$$

This builds the foundation of the RNN architecture. In a similar fashion to the latent variable model described previously, a RNN often is formulated using a hidden state, $\mathbf{h}_{j-1}^{(i)}$, which stores the information of the preceding states in a higher dimension. This allows the hidden state to be calculated at any step by the hidden state of the previous step and the current state, i.e.,

$$\mathbf{h}_j^{(i)} = f\left(\mathbf{s}_j^{(i)}, \mathbf{h}_{j-1}^{(i)}; \Theta_j\right). \tag{17}$$

If the function $f$ is sufficiently powerful, the latent variable model can be exact, as $\mathbf{h}_j^{(i)}$ can store all previously observed data. However, this can lead to high computational and storage costs. A simple deep neural network, a MLP layer, is used in RNNs to describe the function $f$ that allows to compute hidden states $\mathbf{h}_j^{(i)}$ that are used to approximate the likelihood, $P\left(\mathbf{s}_j^{(i)} \mid \mathbf{s}_1^{(i)}, \mathbf{s}_2^{(i)}, \ldots, \mathbf{s}_{j-1}^{(i)}\right)$.

$$\mathbf{a}_j^{(i)} = \Theta_{i,a}^T \mathbf{s}_j^{(i)} + \mathbf{b}_{ia} + \Theta_{h,a} \mathbf{h}_{j-1}^{(i)} + \mathbf{b}_{h,a} \tag{18}$$

$$\mathbf{h}_j^{(i)} = \tanh\left(\mathbf{a}_j^{(i)}\right) \tag{19}$$

As defined in Section 3, $\mathbf{s}_j^{(i)} \in \mathbb{R}^D$, hence, $\Theta_{i,a} \in \mathbb{R}^{D \times H}$, $\Theta_{h,a} \in \mathbb{R}^{H \times H}$ and $\mathbf{b}_{i,a}, \mathbf{b}_{ha} \in \mathbb{R}^H$, where $D$ is the dimensionality of a sequential element of the input sequence and $H$ the dimensionality

of the hidden state. Based on the hidden state, a simple MLP layer can be applied to compute an output,

$$\mathbf{o}_j^{(i)} = \Theta_{h,o}^T \mathbf{h}_j^{(i)} + \mathbf{b}_{h,o}, \tag{20}$$

where $\Theta_{h,o} \in \mathbb{R}^{H \times F}$ and $\mathbf{b}_{h,o} \in \mathbb{R}^F$. $F$ defines the dimensionality of the output. This allows to compute the conditional expectations of the current state $\mathbf{s}_j^{(i)}$. For example, to compute a classification, $\hat{y}_j^{(i)}$, one would apply the softmax to the outputs, $\hat{y}_j^{(i)} = \text{softmax}(\mathbf{o}_j^{(i)})$. The computational logic of the aforementioned mathematical formulations are schematically displayed in Fig. 6(b).

### B.1.2 Long Short-Term Memory (LSTM)

The Elman-style RNNs encounter difficulties in learning long-term dependencies, as identified by Elman [32]. These challenges, articulated by [78] and [79], arise due to vanishing or exploding gradients during backpropagation. In lengthy sequences, recurrent computations are repeatedly applied to the weights. Since these weights are shared across sequential steps, if $\Theta_{ia} \ll 1$, it results in vanishing gradients, whereas if $\Theta_{ia} \gg 1$, it leads to exploding gradients [76]. While gradient clipping mitigates exploding gradients, vanishing gradients require more sophisticated solutions. One of the earliest and most effective methods to address vanishing gradients is the LSTM model introduced by [1]. LSTMs are similar to standard RNNs but replace each recurrent node with a memory cell. Each memory cell includes an internal state with a self-connected recurrent edge, allowing gradients to propagate across many time steps without vanishing or exploding. The term 'long short-term memory' reflects the model's ability to maintain both long-term memory, through slowly changing weights that encode general data knowledge, and short-term memory, through transient activations passed between nodes.

A LSTM cell contains an internal cell state, denoted as $\mathbf{c}_j^{(i)}$.

It includes several multiplicative gates:

- the input gate, $\mathbf{i}_j^{(i)}$, decides if an input, determined by the input node gate, $\mathbf{g}_j^{(i)}$, should affect the internal cell state, $\mathbf{c}_j^{(i)}$,

- the forget gate, $\mathbf{f}_j^{(i)}$, determines if the internal cell state, $\mathbf{c}_j^{(i)}$, should be reset, and

- the output gate, $\mathbf{o}_j^{(i)}$, controls whether the internal cell state should influence the cell's output, $\mathbf{h}_j^{(i)}$.

As in the RNN, the input gate, the forget gate, the input node gate and the output gate are a latent variable model in the form of a MLP layer, with fully connected layers for the input $\mathbf{s}_j^{(i)}$ and the hidden state $\mathbf{h}_{j-1}^{(i)}$, where $\sigma$ denotes a sigmoid activation function. Hence, the gates are described as follows:

$$\mathbf{f}_j^{(i)} = \sigma(\Theta_{i,f}^T \mathbf{s}_j^{(i)} + \mathbf{b}_{i,f} + \Theta_{h,f} \mathbf{h}_{j-1}^{(i)} + \mathbf{b}_{h,f}) \tag{21}$$

$$\mathbf{i}_j^{(i)} = \sigma(\Theta_{i,i}^T \mathbf{s}_j^{(i)} + \mathbf{b}_{i,i} + \Theta_{h,i} \mathbf{h}_{j-1}^{(i)} + \mathbf{b}_{h,i}) \tag{22}$$

$$\mathbf{g}_j^{(i)} = \tanh\left(\Theta_{i,g}^T \mathbf{s}_j^{(i)} + \mathbf{b}_{i,g} + \Theta_{h,g} \mathbf{h}_{j-1}^{(i)} + \mathbf{b}_{h,g}\right) \tag{23}$$

$$\mathbf{o}_j^{(i)} = \sigma\left(\Theta_{i,o}^T \mathbf{s}_j^{(i)} + \mathbf{b}_{i,o} + \Theta_{h,o} \mathbf{h}_{j-1}^{(i)} + \mathbf{b}_{h,o}\right) \tag{24}$$

$$\mathbf{c}_j^{(i)} = \mathbf{f}_j^{(i)} \odot \mathbf{c}_{j-1}^{(i)} + \mathbf{i}_j^{(i)} \odot \mathbf{g}_j^{(i)} \tag{25}$$

$$\mathbf{h}_j^{(i)} = \mathbf{o}_j^{(i)} \odot \tanh(\mathbf{c}_j^{(i)}) \tag{26}$$

Given $\mathbf{s}_j^{(i)} \in \mathbb{R}^D$, hence, $\Theta_{i,f}, \Theta_{i,i}, \Theta_{i,g}, \Theta_{i,o} \in \mathbb{R}^{D \times H}$, $\Theta_{h,f}, \Theta_{h,i}, \Theta_{h,g}, \Theta_{h,o} \in \mathbb{R}^{H \times H}$ and $\mathbf{b}_{i,f}, \mathbf{b}_{i,i}, \mathbf{b}_{i,g}, \mathbf{b}_{i,o} \in \mathbb{R}^H$ and $\mathbf{b}_{h,f}, \mathbf{b}_{h,i}, \mathbf{b}_{h,g}, \mathbf{b}_{h,o} \in \mathbb{R}^H$. The computational logic of the aforementioned mathematical formulations are schematically displayed in Fig. 7(a).

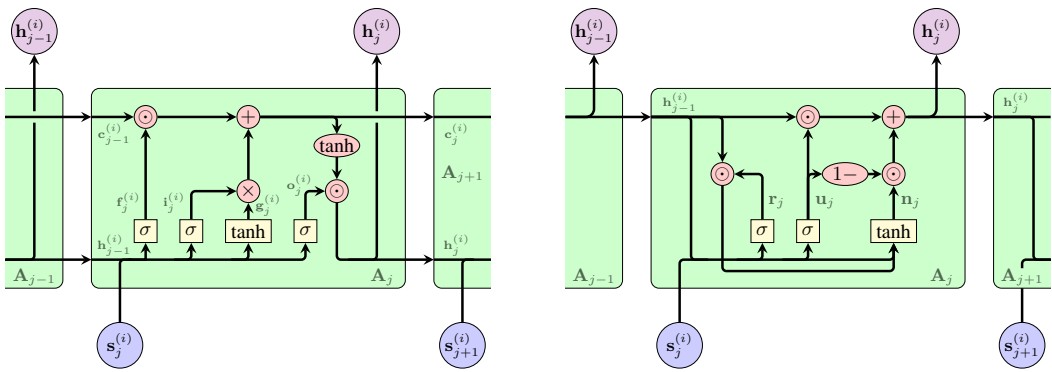

(a) Computational logic in a LSTM cell.          (b) Computational logic in a GRU cell.

Figure 7: Illustration of the computational logic of each cycle applied as a neural network for an adjacent element of the input sequence for a LSTM cell (a) and a GRU cell (b).

### B.1.3    Gated Recurrent Unit (GRU)

The Gated Recurrent Unit (GRU) [14] provides a streamlined architecture of the LSTM memory cell, retaining internal state and multiplicative gating mechanisms while speeding up computation and, often, achieving comparable performance [80]. In a GRU, the LSTM's three gates are replaced by two: the reset gate, $\mathbf{r}_j^{(i)}$, and the update gate, $\mathbf{u}_j^{(i)}$.

The reset gate controls how much of the previous state to remember, while the update gate manages how much of the new state is a copy of the old one. As in previous RNN models, the gates' outputs are produced by a MLP layer with a sigmoid activation function. Similar to the input node gate in the LSTM memory cell, the new gate, $\mathbf{n}_j^{(i)}$, computes the new temporary hidden state. The information used from this state is determined by the reset gate. The final hidden state is determined by incorporating the update gate, $\mathbf{u}_j^{(i)}$, and the new temporary hidden state, $\mathbf{n}_j^{(i)}$. Specifically, the final hidden state, $\mathbf{h}_j^{(i)}$, is computed as a weighted sum of the temporary hidden state, $\mathbf{n}_j^{(i)}$ and the old hidden state, $\mathbf{h}_{j-1}^{(i)}$, with the update gate, $\mathbf{u}_j^{(i)}$, serving as the weight.

$$\mathbf{r}_j^{(i)} = \sigma\left(\Theta_{i,r}^T \mathbf{s}_j^{(i)} + \mathbf{b}_{i,r} + \Theta_{h,r}\mathbf{h}_{j-1}^{(i)} + \mathbf{b}_{h,r}\right) \tag{27}$$

$$\mathbf{u}_j^{(i)} = \sigma\left(\Theta_{i,u}^T \mathbf{s}_j^{(i)} + \mathbf{b}_{i,u} + \Theta_{h,u}\mathbf{h}_{j-1}^{(i)} + \mathbf{b}_{h,u}\right) \tag{28}$$

$$\mathbf{n}_j^{(i)} = \tanh\left(\Theta_{i,n}^T \mathbf{s}_j^{(i)} + \mathbf{b}_{i,n} + \mathbf{r}_j^{(i)} \odot \left(\Theta_{h,n}\mathbf{h}_{j-1}^{(i)} + \mathbf{b}_{h,n}\right)\right) \tag{29}$$

$$\mathbf{h}_j^{(i)} = (1 - \mathbf{u}_j^{(i)}) \odot \mathbf{n}_j^{(i)} + \mathbf{u}_j^{(i)} \odot \mathbf{h}_{j-1}^{(i)} \tag{30}$$

Given $\mathbf{s}_j^{(i)} \in \mathbb{R}^D$, thus, $\Theta_{i,r}, \Theta_{i,u}, \Theta_{i,n} \in \mathbb{R}^{D \times H}$, $\Theta_{h,r}, \Theta_{h,u}, \Theta_{h,n} \in \mathbb{R}^{H \times H}$ and $\mathbf{b}_{i,r}, \mathbf{b}_{i,u}, \mathbf{b}_{i,n} \in \mathbb{R}^H$ and $\mathbf{b}_{h,r}, \mathbf{b}_{h,u}, \mathbf{b}_{h,n} \in \mathbb{R}^H$. The computational logic of the aforementioned mathematical formulations are illustrated in Fig. 7(b).

### B.1.4    Transformer

The seminal Transformer architecture in [2] includes encoders and decoders. Since then, there has been a tendency to use decoder architectures mainly for generative tasks [81–83] and encoder architectures for tasks requiring understanding [42, 84, 23, 21]. STaRFormer follows this trend and chooses only Transformer encoder layers as the central component. Additionally, encoders allow for a general framework for learning task-specific reconstructions that can be applied to a wide range of tasks. It allows one to handle any task, such as classification, regression, generative forecasting, or anomaly detection, by simply adjusting the output layer the latent embedding gets passed to.

As introduced in [2], the **encoder layer** consists of two sub-layers, a multi-head self-attention mechanism, and a fully connected neural network. Both layers are followed by a residual connection and a normalization layer. The **self-attention** in [2] is a mechanism that allows each element of a sequence to consider the entire sequence when computing its representation. This capability helps the model to grasp the context surrounding each token in a sequence, making it highly effective in sequential data tasks. This allows the model to effectively capture both long-term and short-term dependencies within the sequence. This feature addresses the limitations of previous DL approaches like LSTM [1] or RNN [32], which can struggle with capturing such dependencies.

In order for attention to work, each sequence has been embedded as a vector representation. Then, a series of query ($\mathbf{Q}$), key ($\mathbf{K}$) and value ($\mathbf{V}$) terms are formed; $\mathbf{Q}$ is a representation the model focuses on, $\mathbf{K}$ determines the relevance of each element, and $\mathbf{V}$ is a representation used to form output scores. The Scaled Dot-Product Attention approach in [2] computes a weighted sum of the input values with the attention weights, where the weights are determined by the similarity between input elements computed via the softmax function. The attention scores are normalized by the square root of the dimension of the key vectors to stabilize gradients during training, i.e.,

$$\text{Attention}(\mathbf{Q}, \mathbf{K}, \mathbf{V}) = \text{softmax}\left(\frac{\mathbf{Q}\mathbf{K}^T}{\sqrt{d_k}}\right)\mathbf{V}, \tag{31}$$

where $\mathbf{Q}, \mathbf{K}, \mathbf{V} \in \mathbb{R}^{N \times B \times D}$ and $d_k$ is the dimension of key vectors. Often, not just a single self-attention mechanism is performed, but rather a mechanism referred to as *multi-head* attention. In multi-head attention, the queries, keys, and values are linearly projected $n_{\text{head}}$ times and then concatenated and projected to the model's embedding dimension, i.e.,

$$\text{MultiHeadAttention}(\mathbf{Q}, \mathbf{K}, \mathbf{V}) = \left(\bigoplus_{i}^{n_{\text{head}}} \mathbf{H}_i\right)\mathbf{W}^{\mathbf{O}}, \tag{32}$$

where $n_{\text{head}}$ is a tunable hyperparameter and $\mathbf{H}_i = \text{Attention}(\mathbf{Q}\mathbf{W}_i^{\mathbf{Q}}, \mathbf{K}\mathbf{W}_i^{\mathbf{K}}, \mathbf{V}\mathbf{W}_i^{\mathbf{V}})$ represents an attention head. Here, $\mathbf{W}_i^{\mathbf{Q}} \in \mathbb{R}^{d_{\text{model}} \times d_k}$, $\mathbf{W}_i^{\mathbf{K}} \in \mathbb{R}^{d_{\text{model}} \times d_k}$, $\mathbf{W}_i^{\mathbf{V}} \in \mathbb{R}^{d_{\text{model}} \times d_v}$ and $\mathbf{W}_i^{\mathbf{V}} \in \mathbb{R}^{n_{\text{head}} d_v \times d_{\text{model}}}$ are projection matrices where $d_*$ indicate the respective dimensions.

A necessity for Transformer models is the encoding process of the sequential inputs. When the sequential input is vectorized, the input representation loses the sequential information, i.e., the order of the sequence. Hence, an underlying property of the data type is lost. This is why it is essential to inject the sequential information about the relative or absolute sequential position into the vector representation. To do so, [2] introduces sinusoidal **positional encodings**, which are added to the encoded sequence before it is passed to the Transformer encoder layer, i.e.:

$$PE_{(\text{pos}, 2i)} = \sin\left(\text{pos}/10000^{2i/d_{\text{model}}}\right) \tag{33}$$

$$PE_{(\text{pos}, 2i+1)} = \cos\left(\text{pos}/10000^{2i/d_{\text{model}}}\right) \tag{34}$$

As mentioned before, in certain time series, the sequence lengths $N$ can vary. Consequently, to process batches of sequences with differing lengths, padding is necessary. To ensure that padded elements are not considered during the attention mechanism, it is crucial to introduce a **batch-wise masking** strategy. This mask is passed to the attention process to prevent artificially padded elements from being attended to, thereby preserving the integrity of the sequential data. This mask might further be required in different output heads or loss formulations.

## B.2 Downstream Tasks

**Classification.** Instead of performing autoregressive predictions [21] or predictions based on the concatenation of the entire embedded representation [23], a special token is used ins STaRFormer for classification tasks. This is a design choice and no requirement. The token effectively captures the dependencies between sequential elements via the encoder's self-attention mechanism. When performing classification tasks, a multilayer perceptron (MLP) performs the classification based on this token. The reduced version of the MLP layer used in STaRFormer, is a fully connected neural network followed by a sigmoid or softmax activation, depending if multi-class or binary predictions are required, i.e.,

$$\hat{\mathbf{y}} = \sigma(\mathbf{Z}_{\text{CLS}}\Theta + \mathbf{b}), \tag{35}$$

while the default version adds an additional activation layer and a normalization layer, i.e.,

$$\hat{\mathbf{y}} = \sigma((\text{Norm}(\sigma(\mathbf{Z}_{\text{CLS}}\Theta + b)))\Theta + \mathbf{b}). \tag{36}$$

where $\hat{\mathbf{y}} \in \mathbb{R}^{B \times K}$, $\mathbf{Z}_{\text{CLS}} = \mathbf{Z}_{0,:,:} \mid \in \mathbb{R}^{B \times F}$ is the specialized token of the latent embedding and $\Theta \in \mathbb{R}^{F \times K}$ and $\mathbf{b} \in \mathbb{R}^{K}$ are the model weights and bias, where $K$ defines the output dimension of the final layer. The inner activation function, $\sigma$ is a tuneable hyperparameter. The output of the MLP layer, i.e., the prediction $\hat{y}^{(i)}$, is passed through either the cross-entropy (CE) loss function for multi-class predictions, $\mathfrak{L}_{\text{task}} = \sum_{i=1}^{C} y^{(i)} \log(\hat{y}^{(i)})$, or the binary cross-entropy (BCE) loss for binary predictions, $\mathfrak{L}_{\text{task}} = y^{(i)} \log(\hat{y}^{(i)}) + (1 - y^{(i)}) \log(\hat{y}^{(i)})$.

**Anomaly Detection.** The same output head configuration is employed for anomaly detection, with only the output dimensionality $K$ adjusted accordingly. In this setup, a dedicated classification token is unnecessary, as predictions are made at the element level. Therefore, the raw latent embedding $\mathbf{Z} \in \mathbb{R}^{B \times N \times F}$ serves as the input to the output head. Moreover, as the sequences possess element-wise anomalous labels, anomaly detection tasks requires predictions at each time step. Consequently, the model must generate $N$ predictions for an input sequence $\mathbf{S}^{(i)} \in \mathbb{R}^{N \times D}$, hence $\hat{\mathbf{y}} \in \mathbb{R}^{B \times N}$.

**Regression.** The objective is to predict a scalar value for each sequence, as defined in Section 3. We adopt the output-head configuration used for classification, but fix the output dimension to $K = 1$. Moreover, the model output does not require an outer activation function, such as softmax, since we aim to predict scalar values. Accordingly, the reduced version is

$$\hat{\mathbf{y}} = \hat{\mathbf{Z}}\Theta + b. \tag{37}$$

The default version incorporates an additional hidden layer, i.e.,

$$\hat{\mathbf{y}} = \text{Norm}(\sigma(\text{Norm}(\sigma(\hat{\mathbf{Z}}\Theta + b))\Theta + b))\Theta + b. \tag{38}$$

Here, $\hat{\mathbf{Z}} \in \mathbb{R}^{B \times F}$ is computed via an average pooling operation over each sequence, i.e., the mean per sequence, and $\hat{\mathbf{y}} \in \mathbb{R}^{B}$ denotes the predicted scalar value per sequence in a batch.

## B.3 Semi-supervised Task Informed Representation Learning in STaRFormer

Fig. 8 display the Contrastive Learning (CL) approach applied in STaRFormer. The CL approach ultimately depends on the accumulated latent representations, $\mathbf{Z}$ and $\tilde{\mathbf{Z}}$, that are created via the Dynamic Attention-based Regional Masking (DAReM) scheme. During the computation of the multi-head attention, while executing training of a downstream task, STaRFormer dynamically collects the attention weights, $\mathbf{A}$. To aggregate the attention, STaRFormer employs a slightly modified attention rollout technique [58] rather than mere summation or aggregation. This approach enables a better consideration of the flow of information within the Transformer layers. The operation accounts for padded sequences by masking the attention if necessary, i.e.,

$$\tilde{\mathbf{A}} = \begin{cases} (\frac{1}{2}\mathbf{A}_{i,:,:,:} \odot \mathbf{M} + \frac{1}{2}\mathbf{I}_N) \otimes \tilde{\mathbf{A}}_{i-1,:,:,:} & \text{if } i > 0, \\ \mathbf{A}_{i,:,:,:} \odot \mathbf{M} & \text{if } i = 0 \end{cases} \tag{39}$$

where $\tilde{\mathbf{A}} \in \mathbb{R}^{B \times N \times N}$ represents the aggregated attention, and $\mathbf{M}$ the mask accounting for the padded input sequences. STaRFormer then computes the attention scores $\sigma$ as in [21], i.e.,

$$\sigma_{i,k'} = \frac{\sum_{j=1}^{N} \tilde{\mathbf{A}}_{i,j,k}}{\sum_{k=1}^{N} \sum_{j=1}^{N} \tilde{\mathbf{A}}_{i,j,k}}, \tag{40}$$

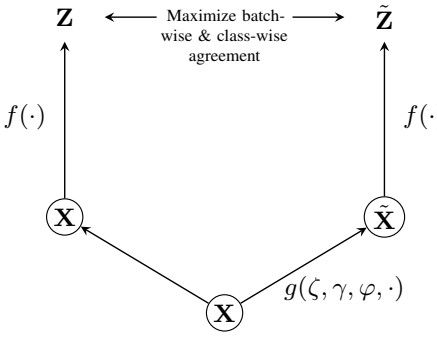

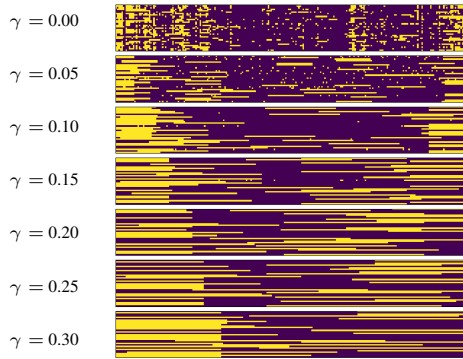

Figure 8: STaRFormer's CL approach: The masking, $g$, generates two correlated views. The encoder, $f$, is trained to maximize the trade-off between batch- and class-wise agreement of the latent embeddings $\mathbf{Z}$ and $\tilde{\mathbf{Z}}$ while training for a downstream task.

Figure 9: Seven different regional masks for the same batch, with sequences aligned horizontally and stacked vertically (per $\mathbf{X}$). The masked regions in DAReM are defined by different values of $\gamma$, with $\varphi \approx 0.2$ and $\zeta = 0.3$ held constant and depicted in yellow.

---

**Algorithm 1** Dynamic Attention-based Regional Masking (DAReM)

---

**Require:** $\mathbf{A}, \mathbf{n}, B, \zeta, \gamma, \varphi$
  $\tilde{\mathbf{A}} \leftarrow \texttt{attention-rollout}(\mathbf{A})$
  $\sigma \leftarrow \texttt{attention-scores}(\tilde{\mathbf{A}})$
  mask-indices $\leftarrow []$
  **for** $i$ **in range** $(B)$ **do**
    $\sigma_{\text{top}} \leftarrow \texttt{topk}(\zeta, \mathbf{n}_i)$
    $\sigma_{\text{top1}} \leftarrow \sigma_{\text{top}}[0]$
    $b_{\text{top1}} \leftarrow \mathbf{n}_i \cdot \gamma$
    mask-indices$_i \leftarrow []$
    $m_i \leftarrow \texttt{range}(\max(0, \sigma_{\text{top1}} - b_{\text{top1}}), \min(\mathbf{n}_i, \sigma_{\text{top1}} + b_{\text{top1}} + 1))$
    mask-indices$_i$.append$(m_i)$
    **if** $\texttt{len}(\text{mask-indices}_i) \leq \mathbf{n}_i \cdot \varphi$ **then**
      **for** $i, \sigma_{\text{top}k}$ **in** $\texttt{enumerate}(\sigma_{\text{top}}[1:])$ **do**
        **if** $\texttt{len}(\text{mask-indices}_i) \leq \mathbf{n}_i \cdot \varphi$ **then**
          $b_{\text{top}k} \leftarrow \mathbf{n}_i \cdot \gamma$
          mask-indices$_i$.extend$(m_i)$
        **end if**
      **end for**
    **end if**
    **if** $\texttt{len}(\text{mask} - \text{indices}_i) \leq \mathbf{n}_i \cdot \varphi$ **then**
      $m_j \leftarrow \texttt{random}(\text{available-indices}, n_{\text{diff}})$
      mask-indices$_i$.extend$(m_j)$
    **end if**
    mask-indices.append(mask-indices$_i$)
  **end for**
  **return** mask-indices

---

where $\tilde{\mathbf{A}}_{i,j,k}$ is the attention weight assigned to $\mathbf{s}_k^{(i)}$ during the update of $\mathbf{s}_j^{(i)}$ in Eq. (31). A greater $\sigma_{i,k'}$ value indicates a higher importance of the $k$-th element in $\mathbf{S}^{(i)}$. Algorithm 1 implements the arithmetic's of the DAReM scheme introduced in STaRFormer. In the algorithm, $\mathbf{A}$ refers to the attention weights collected from the multi-head attention layer in the encoder, $\mathbf{n}$ refers to an array stating the sequence lengths per element in the mini-batch and $B$ refers to the batch-size of the mini-batch. The masking parameters $\varphi$, $\gamma$ and $\zeta$ are introduced in Section 3.1.1. We illustrate several regional masks for the same batch, with the region parameter $\gamma$ varied while $\varphi$ and $\zeta$ remain fixed in Fig. 9. To note is that if the region of the most important sequential element is already

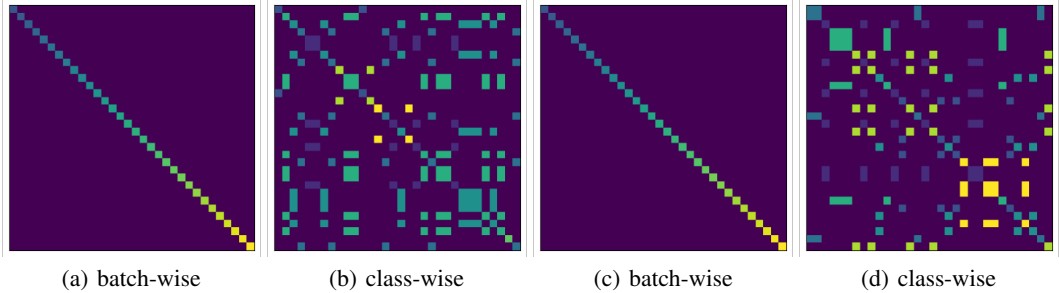

| (a) batch-wise | (b) class-wise | (c) batch-wise | (d) class-wise |

Figure 10: Example visualizations from two different batches of the PAM dataset: images (a) and (b) are from one batch, while images (c) and (d) are from another batch, both have a mini-batch of size $B = 32$. Positive pairs within each batch are color-coded. The darkest shade represents negative pairs.

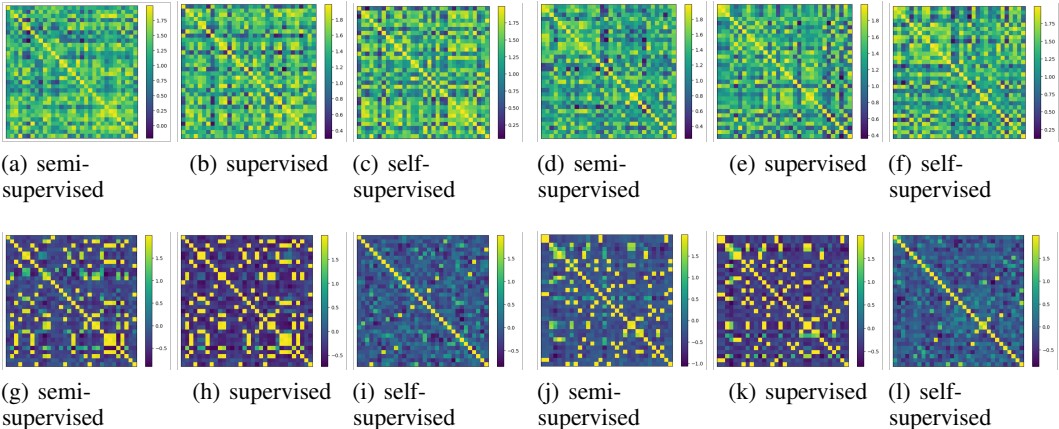

(a) semi-supervised (b) supervised (c) self-supervised (d) semi-supervised (e) supervised (f) self-supervised

(g) semi-supervised (h) supervised (i) self-supervised (j) semi-supervised (k) supervised (l) self-supervised

Figure 11: Similarity heat maps illustrating the contrastive loss formulation in STaRFormer for two mini-batches of size $B = 32$ from the PAM dataset. The top row displays similarities between latent embeddings $\hat{\mathbf{Z}}^{(i)}$ and $\hat{\tilde{\mathbf{Z}}}^{(i)}$ for an untrained model, while the bottom row shows similarities between $\hat{\mathbf{Z}}^{(i)}$ and $\hat{\tilde{\mathbf{Z}}}^{(i)}$ for a trained model. Plots (a)-(c) and (g)-(i) pertain to one batch (same batch as in plots (a) and (b) in Fig. 10), whereas plots (d)-(f) and (j)-(l) pertain to another batch (same batch as in plots (c) and (d) in Fig. 10).

greater than threshold $\varphi$, only that region is masked, and the other selected $\sigma$ values are dropped. If all important sequential regions are masked and the threshold still allows samples to be masked, then random samples are selected using the available indices of $\sigma$. We opt to select the following bounds for the masking parameters: $\varphi \in (0.0, 0.5]$, $\gamma = \{5j \times 10^{-2} \mid j \in \{0, 1, 2, 3, 4, 5\}\}$ and $\zeta = \{j \times 10^{-1} \mid j \in \{1, 2, 3, 4, 5\}\}$.

In Section 3.1.2, we introduce the implementation of a semi-supervised CL paradigm as employed in the STaRFormer framework. This methodology exploits the inherent batch-wise and class-wise agreement between masked ($\tilde{\mathbf{Z}}^{(i)}$) and unmasked ($\mathbf{Z}^{(i)}$) latent representations allowing to facilitate semi-supervised CL. Fig. 10 provides a visual illustration of positive pair selection under the semi-supervised CL framework, using two distinct batches from the Physical Activity Monitoring (PAM) dataset. It demonstrates the construction of batch-wise and class-wise positive pairs for contrastive learning. For batch-wise pairs, the corresponding diagonal elements are selected (Fig. 10(a) and 10(c)), whereas for class-wise pairs, the corresponding diagonal and off-diagonal elements are selected.

In Fig. 11, the results of deploying different possible CL paradigms are graphically represented. Specifically, as discussed in Section 4.4.2, we analyze three different learning paradigms: semi-supervised, supervised, and self-supervised. Fig. 11 is a continuation of the visual analysis initiated

in Fig. 10, employing the same two batches for consistency. The upper row of Fig. 11 presents the similarity heat maps for a model prior to training, while the lower row illustrates the heat maps post-training. A color-coded scheme is utilized to convey similarity levels, with yellow indicating high similarity and purple denoting low similarity. The matrices' diagonal entries quantify the self-similarity among batch elements. In contrast, the off-diagonal entries measure the degree of similarity between disparate batch elements. The two rows within Fig. 11 visualize the learning objective of CL as described in Section 3.1.2 and Fig. 8, where similar samples are pulled closer together, while dissimilar samples are pushed further apart. In the top row, the untrained model evaluates relatively high similarity across all element pairs within the batch, regardless of the CL paradigm used, whereas the trained model clearly distinguishes between similar and dissimilar samples in the batch in accordance to the contrastive paradigm applied.

When the model is trained to prioritize batch-wise similarity under the self-supervised contrastive learning paradigm, the heat maps reveal, as expected, brightly colored diagonal entries in yellow, indicating the intended emphasis on self-similarity. This is illustrated in Fig. 11(i) and 11(l). Conversely, the off-diagonal entries are predominantly cast in darker shades ranging from blue to purple, indicating a stark contrast in similarity and, thus, a clear distinction between different elements.

Training with an emphasis on class-wise similarity yields a different pattern; refer to the heat maps in Fig. 11(h) and 11(k). Here, in addition to self-similar elements, the heat maps distinctly accentuate high similarity among elements belonging to the same class, while elements of disparate classes are clearly differentiated by lower similarity scores, reflecting the model's class-wise learning.

The semi-supervised training paradigm offers a composite view, where the model demonstrably assimilates both batch-wise and class-wise similarities (see Fig. 11(g) and 11(j)). This dual learning is evidenced by the pronounced similarity not only along the self-similar diagonal entries but also between class-aligned, diagonal, and off-diagonal elements. However, the off-diagonal entries not associated with class similarity do not display the darkened hues observed in the strictly supervised model. This absence of dark hues suggests a more tempered and generalized learning process, where the model avoids overfitting to specific batch-wise or class-wise similarities, potentially achieving a more holistic representation of the data.

### B.3.1 Additional Information - Formulation 1

The resulting cosine similarity matrix computed from the reduced latent space representation $\hat{\mathbf{Z}}_{i,j}$ and $\hat{\tilde{\mathbf{Z}}}_{i,j}$ represents the inter-batch similarity between all sequences in a batch. Hence, elements at position where $i = j$ in the similarity matrix originate from the same input sequence $\mathbf{S}^{(i)}$. Thus, given $\hat{\mathbf{Z}}_{i,j} \in \mathbb{R}^{B \times F}$, STaRFormer can form $B$ positive and $B(B-1)$ negative batch-wise pairs. By having $C$ classes per $\mathbf{X}$, we obtain $\sum_{c=1}^{C} n_c^2$ positive and $\left(\sum_{c=1}^{C} n_c\right)^2 - \left(\sum_{c=1}^{C} n_c^2\right)$ negative class-wise pairs; $n_c$ is the number of samples per class per $\mathbf{X}$.

### B.3.2 Additional Information - Formulation 2

As stated in Section 3.1, the element-wise formulation allows to create *intra-* and *inter* class positive pairs. *Intra-class* positive pairs are created between elements within a sequence whereas *inter-class* positive pairs are created between elements with other sequences in a mini-batch.

**Inter-class.** We chose to select an *inter-class* positive pair if $\mathbb{I}^{(i,j)}_{\mathrm{inter},\left[\mathbf{Y}_{\mathrm{l}}^{(i,j)}=\mathbf{Y}_{\mathrm{r}}^{(i,j)}\right]}$ is 1, i.e., the class in the left and right label is equal to each other, non-negative and the elements are not from the same sequence. The following equation, for completeness, defines the indicator function applied for the inter-class class-wise contrastive loss formulation in Section 3.1.2.

$$\mathbb{I}^{(i,j)}_{\mathrm{inter},\left[\mathbf{Y}_{\mathrm{l}}^{(i,j)}\mathbf{Y}_{\mathrm{r}}^{(i,j)}\right]} := \begin{cases} 1 & \text{if } \mathfrak{S}_i \neq \mathfrak{S}_j \wedge \mathbf{Y}_{\mathrm{l}}^{(i,j)} = \mathbf{Y}_{\mathrm{r}}^{(i,j)} \wedge \mathbf{Y}_{\mathrm{l}}^{(i,j)} > -1 \wedge \mathbf{Y}_{\mathrm{r}}^{(i,j)} > -1 \\ 0 & \text{otherwise} \end{cases} \tag{41}$$

This definition accounts for padded elements, where the label tensor equals $-1$.

**Intra-class.** We chose to select an *intra-class* positive pair if $\mathbb{I}^{(i,j)}_{\text{intra},\left[\mathbf{Y}_{\text{l}}^{(i,j)}=\mathbf{Y}_{\text{r}}^{(i,j)}\right]}$ is 1, i.e., the class in the left and right label is equal to each other, non-negative and the elements are from the same sequential input element. In this case, a few modifications are necessary. The left and right label tensors are created as $\mathbf{Y}_{\text{l}} \in \mathbb{R}^{B \times N \times 1}$ and $\mathbf{Y}_{\text{r}} \in \mathbb{R}^{1 \times B \times N}$ respectively. Additionally, the cosine similarity needs to be computed between each element of a sequence; thus, we require a three-dimensional similarity matrix. As described in Section 3.1, this requires $\bigotimes_{\text{bmm}}$ in the similarity computation. Thus, the cosine similarity is defined as:

$$\text{sim}_{\text{intra}}(\mathbf{U}, \mathbf{V}) = \frac{\mathbf{U} \bigotimes_{\text{bmm}} \mathbf{V}}{\|\mathbf{U}\|\|\mathbf{V}\|} \tag{42}$$

The latent embeddings $\mathbf{Z}_{\text{perm}}$ and $\tilde{\mathbf{Z}}_{\text{perm}}$ are permuted equivalents of $\mathbf{Z}$ and $\tilde{\mathbf{Z}}$, where $\mathbf{Z}_{\text{perm}} \in \mathbb{R}^{B \times N \times D}$ and $\tilde{\mathbf{Z}}_{\text{perm}} \in \mathbb{R}^{B \times D \times N}$. For completeness, the following equation defines the indicator function applied for the intra-class class-wise contrastive loss formulation in Section 3.1.2.

$$\mathbb{I}^{(i,j)}_{\text{intra},\left[\mathbf{Y}_{\text{l}}^{(i,j)}=\mathbf{Y}_{\text{r}}^{(i,j)}\right]} := \begin{cases} 1 & \text{if } i \neq j \wedge \mathbf{Y}_{\text{l}}^{(i,j)} = \mathbf{Y}_{\text{r}}^{(i,j)} \wedge \mathbf{Y}_{\text{l}}^{(i,j)} > -1 \wedge \mathbf{Y}_{\text{r}}^{(i,j)} > -1 \\ 0 & \text{otherwise} \end{cases} \tag{43}$$

This definition accounts for padded elements, where the label tensor equals $-1$.

### B.4 Limitations of STaRFormer

A key limitation of the proposed approach is the computational overhead introduced by the attention-based masking mechanism (DAReM), which requires the computation of attention weights scaling with $\mathcal{O}(N^2)$ complexity, which becomes increasingly computationally expensive proportional to the length of the sequences. Additionally, integrating CL and DAReM during training further increases training time and computational demands, as each input must be processed two times on top of the increased workload by applying CL in the first place. However, these overheads are confined to the training phase and do not impact inference performance, where only the downstream prediction task is executed. Despite the additional computational overhead introduced by CL and DAReM, STaRFormer maintains comparable batch sizes during training. For instance, STaRFormer trains with batch sizes of 512 and 256 on DKT and Geolife (GL), respectively, matching those of the baseline transformer model. In practice, batch size limitations are primarily constrained by sequence length due to the quadratic complexity of attention-weight computation.

The outlined limitations become more pronounced at scale, particularly when training on large datasets such as DKT. While the additional computational cost is negligible for small datasets, it becomes a significant factor during large-scale training, especially in the context of hyperparameter tuning. Although this work prioritizes predictive performance over computational efficiency, we acknowledge the potential for optimizing the implementation of DAReM to mitigate training overhead and more efficient attention computation to improve scalability.

## C  Datasets

In Table 8, we display the different attributes of each dataset used in this work. For GL and DKT, we use five different seeds to ensure a fair evaluation of STaRFormer's performance. For DKT, we keep the test set fixed across all splits. As default, we set the seed to 42 and use 123, 0, 63, and 2024 additionally. For the benchmarks of GL, the UEA benchmark (UEA) [70] and the TSR benchmark (TSR) [73], we only report the best performing model in the paper of a single seed, to be consistent with previous literature.

Table 8: Time Series Datasets Overview

| # | Task | Type | Dataset | Train Samples | Test Sample | Classes | Max Length | Dimension | Literature | Link |
|---|------|------|---------|--------------|-------------|---------|-----------|-----------|-----------|------|
| 1 | | Non-Stationary | Digital Key Trajectories (DKT) | 447,765 | 111,944 | 2 | 677 | 8 | - | - |
| 2 | | | Geolife (GL) | 6,434 | 1,556 | 4 | 7,990 | 10 | [63] | geolife-link |
| 3 | | Irregularly Sampled | PhysioNet Sepsis Early Prediction Challenge 2019 (P19) | 34,922 | 3,881 | 2 | 60 | 34 | [66] | p19-link |
| 4 | | | PhysioNet Mortality Prediction Challenge 2012 (P12) | 10,789 | 1,199 | 2 | 215 | 36 | [67] | p12-link |
| 5 | | | Physical Activity Monitoring (PAM) | 4799 | 534 | 8 | 600 | 17 | [68] | pam-link |
| 6 | | | Articulary Word Recognition (AWR) | 275 | 300 | 25 | 144 | 9 | [85] | awr-link |
| 7 | | | Atrial Fibrillation (AF) | 15 | 15 | 3 | 640 | 2 | [67] | af-link |
| 8 | | | Basic Motions (BM) | 40 | 40 | 4 | 100 | 6 | [67] | bm-link |
| 9 | | | Character Trajectories (CT) | 1,422 | 1,436 | 20 | 182 | 3 | [86] | ct-link |
| 10 | | | Cricket (CK) | 108 | 72 | 12 | 1,197 | 6 | [87] | ck-link |
| 11 | | | Duck Duck Geese (DDK) | 60 | 40 | 5 | 270 | 1,345 | [88] | ddk-link |
| 12 | | | Eigen Worms (EW) | 131 | 128 | 5 | 17,984 | 6 | [89] | ew-link |
| 13 | | | Epilepsy (EP) | 137 | 138 | 4 | 206 | 3 | [90] | ep-link |
| 14 | | | ERing (ER) | 30 | 30 | 6 | 65 | 4 | [91] | er-link |
| 15 | | | Ethanol Concentration (EC) | 261 | 263 | 4 | 1,751 | 3 | [92] | ec-link |
| 16 | Classification | | Face Detection (FD) | 5,890 | 3,524 | 2 | 62 | 144 | [93] | fd-link |
| 17 | | | Finger Movements (FM) | 316 | 100 | 2 | 50 | 28 | [94] | fm-link |
| 18 | | | Hand Movement Direction (HMD) | 320 | 147 | 4 | 400 | 10 | [95] | hmd-link |
| 19 | | | Handwritting (HW) | 150 | 850 | 26 | 152 | 3 | [96] | hw-link |
| 20 | | Regular | Heartbeat (HB) | 204 | 205 | 2 | 405 | 61 | [97] | hb-link |
| 21 | | | Insect Wingbeat (IW) | 30,000 | 20,000 | 10 | 78 | 200 | [98] | iw-link |
| 22 | | | Japenese Vowels (JV) | 270 | 370 | 9 | 29 | 12 | [99] | jv-link |
| 23 | | | Libras (LI) | 180 | 180 | 15 | 45 | 2 | [100] | li-link |
| 24 | | | LSST (LSST) | 2,459 | 2,466 | 14 | 36 | 6 | [101] | lsst-link |
| 25 | | | Motor Imagery (MI) | 278 | 100 | 2 | 3,000 | 64 | [102] | mi-link |
| 26 | | | NATOPS (NT) | 180 | 180 | 6 | 51 | 24 | [103] | nt-link |
| 27 | | | PEMS-SF (PS) | 267 | 173 | 7 | 144 | 963 | [104] | ps-link |
| 28 | | | Pen Digits (PD) | 7,494 | 3,498 | 10 | 8 | 2 | [105] | pd-link |
| 29 | | | Phoneme Spectra (PSp) | 3,315 | 3,353 | 39 | 217 | 11 | [106] | psp-link |
| 30 | | | Racket Sports (RS) | 151 | 152 | 4 | 30 | 6 | [107] | rs-link |
| 31 | | | Self Regulation SCP1 (SCP1) | 268 | 293 | 2 | 896 | 6 | [108] | scp1-link |
| 32 | | | Self Regulation SCP2 (SCP2) | 200 | 180 | 2 | 1,152 | 7 | [109] | scp2-link |
| 33 | | | Spoken Arabic Digits (SAD) | 6,599 | 2,199 | 10 | 65 | 13 | [110] | sad-link |
| 34 | | | Stand Walk Jump (SWJ) | 12 | 15 | 3 | 2,500 | 4 | [67] | swj-link |
| 35 | | | UWave Gesture Library (UW) | 2,238 | 2,241 | 8 | 315 | 3 | [111] | uw-link |
| 36 | Anomaly Detection | - | A Labeled Anomaly Detection Dataset (Yahoo) | 367 | 367 | 2 | 840 | 1 | [72] | yahoo-link |
| 37 | | | KPI | 58 | 58 | 2 | 74,581 | 1 | [71] | kpi-link |
| 38 | | | Appliances Energy (AE) | 96 | 42 | - | 144 | 24 | [112] | ae-link |
| 39 | | | Australia Rainfall (AR) | 112,186 | 48,081 | - | 24 | 3 | [113] | ar-link |
| 40 | | | Beijing PM10 Quality (BPM10) | 12,432 | 5,100 | - | 24 | 9 | [114] | bpm10-link |
| 41 | | | Beijing PM25 Quality (BPM25) | 12,432 | 5,100 | - | 24 | 9 | [115] | bpm25-link |
| 42 | | | Benzene Concentration (BC) | 3,433 | 5,445 | - | 240 | 8 | [116] | bc-link |
| 43 | | | BIDMC32HR (BIDMCHR) | 5,471 | 2,399 | - | 4,000 | 2 | [117] | bidmchr-link |
| 44 | | | BIDMC32RR (BIDMCRR) | 5,550 | 2,399 | - | 4,000 | 2 | [118] | bidmcrr-link |
| 45 | | | BIDMC32SpO2 (BIDMCSPO2) | 5,550 | 2,399 | - | 4,000 | 2 | [119] | bidmcspo2-link |
| 46 | | | Covid3Month (C3M) | 140 | 61 | - | 84 | 1 | [120] | c3m-link |
| 47 | Regression | - | Flood Modeling 1 (FM1) | 471 | 202 | - | 266 | 1 | [121] | fm1-link |
| 48 | | | Flood Modeling 2 (FM2) | 389 | 167 | - | 266 | 1 | [122] | fm2-link |
| 49 | | | Flood Modeling 3 (FM3) | 429 | 184 | - | 266 | 1 | [123] | fm3-link |
| 50 | | | Household Power Consumption 1 (HPC1) | 746 | 694 | - | 1,440 | 5 | [124] | hpc1-link |
| 51 | | | Household Power Consumption 2 (HPC2) | 746 | 694 | - | 1,440 | 5 | [125] | hpc2-link |
| 52 | | | IEEEPPG | 1,768 | 1,328 | - | 1,000 | 5 | [126] | ieeeppg-link |
| 53 | | | Live Fuel Moisture Content (LFMC) | 3,493 | 1,510 | - | 365 | 7 | [127] | lfmc-link |
| 54 | | | News Headline Sentiment (NHS) | 58,213 | 24,951 | - | 144 | 3 | [128] | nhs-link |
| 55 | | | News Title Sentiment (NTS) | 58,213 | 24,951 | - | 144 | 3 | [129] | nts-link |
| 56 | | | PPG Dalia (PPG) | 43,215 | 21,482 | - | 256-512 | 4 | [130] | ppg-link |

## C.1  Classification Time Series Datasets

This section introduces the datasets used to evaluate the performance for time series classification.

### C.1.1  Non-Stationary Spatiotemporal Time Series Datasets

In this section, we provide details about the non-stationary spatiotemporal datasets used in Section 4.

#### C.1.1.1  Real-World Digital Key Trajectories (DKT) Dataset

The DKT dataset comprises multivariate time series data, capturing x- and y-positions sequentially to predict the intent of the smart device carrier. In total, the DKT dataset comprises 559,709 anonymized customer trajectories, recorded over a span of three months from a subset of BMW's fleet of vehicles. This dataset of labeled trajectories was obtained using high-precision localization with UWB technology and the DK. It includes various vehicle types, from small hatchbacks to large SUVs. Each trajectory is associated with a binary label, $y \in \{0, 1\}$, indicating whether a specific action is

Table 9: DKT label distribution (in %) and number of samples per data-subset for seed 42.

| | Label Distribution (%) | | Num. of Samples |
|---|---|---|---|
| | 0 | 1 | |
| Train Dataset | 48.50 | 51.50 | 358,211 |
| Val Dataset | 48.81 | 51.19 | 89,554 |
| Test Dataset | 48.67 | 51.33 | 111,944 |
| Total | 48.65 | 51.34 | 599,709 |

taking place (1) or not (0). The label distribution is approximately 48/52, with 52 % corresponding to label 1. However, localization accuracy can be affected by various external factors and ranging algorithms, as discussed in Appendix A. Consequently, the DKT data includes irregularly sampled and non-stationary sequential data. Table 9 displays the label distribution in the DKT dataset.

### C.1.1.2   Geolife (GL) Dataset

The GL GPS trajectory dataset was collected by 182 users as part of the Microsoft Research Asia Geolife project over a span of more than five years (from April 2007 to August 2012) [63]. Each GPS trajectory in this dataset is a sequence of time-stamped points, providing information on latitude, longitude, and altitude. The dataset comprises 17,621 trajectories, covering a total distance of approximately 1.2 million kilometers and a total duration exceeding 48,000 hours. These trajectories were recorded using various GPS loggers and GPS-enabled phones, featuring a range of sampling rates. This dataset captures a wide array of users' outdoor movements, including everyday activities like commuting to work or home, as well as recreational and sports activities such as shopping, sightseeing, dining, hiking, and cycling. The GL trajectory dataset is valuable for research in multiple fields, including mobility pattern mining, user activity recognition, location-based social networks, location privacy, and location recommendation [63].

Our pre-processing of the data before it is usable in training includes:

- filtering for labeled and unlabeled samples

- removing samples that have fewer than five sequential elements

- the data is restricted to the Beijing metropolitan area, which is approximately $96\%$ of the entire labeled data

- converting longitude, latitude, and altitude to SI units, i.e., meters (m)

- considering the imbalance in data distribution, all trajectories labeled airplane, boat, motorcycle, subway and train are dropped, runs are considered a walk and taxis considered a car

- filtering trajectories that surpass a certain speed limit, indicating that the label is wrong. As walks and runs are combined, we consider an average pace of 12 km/h, i.e., 5 min/km, as the speed boundary for the walk-run class, 60 km/h for bikes, 100 km/h for busses, and 120 km/h for cars. The last two correspond to the speed limits in China.

- removing outliers

After the data has been preprocessed, it consists of 7,990 samples. We use a ratio of 7/1/2 to split the data into training, validation, and testing respectively, following [17]. The breakdown of the class distribution is provided in Table 10.

Table 10: Label distribution (in %) of pre-processed GL dataset of seed 42.

| Label | Num. of Samples | Label Distribution ($\%$) |
|-------|-----------------|---------------------------|
| bike  | 1,534           | 19.20                     |
| bus   | 1,745           | 21.84                     |
| car   | 1,186           | 14.84                     |
| walk  | 3,525           | 44.12                     |
| Total | 7,990           |                           |

In Fig. 12, the trajectories of the dataset are visualized for the Beijing metropolitan area.

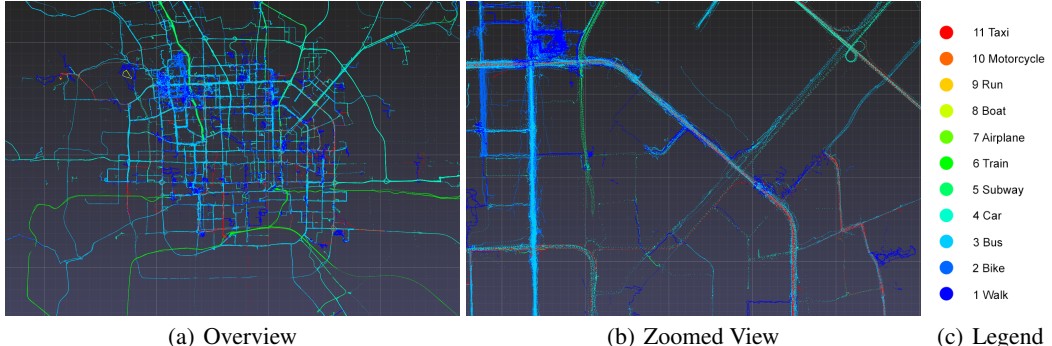

|  (a) Overview | (b) Zoomed View | (c) Legend |

Figure 12: This figure presents two images depicting the collected trajectories from the GL dataset in the Beijing Metropolitan Area. Image (a) provides a complete overview, image (b) shows a zoomed-in version, and image (c) includes the legend that explains the color coding used for the trajectories in both (a) and (b). Images are taken from `https://heremaps.github.io/pptk/tutorials/viewer/geolife.html`.

### C.1.2 Irregular Sampled Time Series Datasets

In this section, we provide details about the irregular sampled time series datasets used in Section 4.

#### C.1.2.1 PhysioNet Sepsis Early Prediction Challenge 2019 (P19) Dataset

The PhysioNet Sepsis Early Prediction Challenge 2019 (P19) dataset comprises time series records from 38,803 ICU patients, each monitored via 34 physiological sensors. From the original 40,336 patients, samples with extremely short or long sequences (fewer than 2 or more than 60 observations) were excluded. Each patient is also associated with a static feature vector encoding demographic and clinical attributes, including age, gender, ICU type, ICU stay duration, and time elapsed between hospital and ICU admission. The prediction task involves a binary label indicating whether sepsis will occur within the subsequent 6 hours [20]. The dataset is highly imbalanced, as displayed in Table 11. To ensure a fair evaluation, the performance is averaged over five consistent data splits.

Table 11: P19 Label Distribution in percentage (%).

|  | Split 0 | | Split 1 | | Split 2 | | Split 3 | | Split 4 | | Num. of Samples |
| --- | --- | --- | --- | --- | --- | --- | --- | --- | --- | --- | --- |
|  | 0 | 1 | 0 | 1 | 0 | 1 | 0 | 1 | 0 | 1 | |
| Training Dataset | 95.78 | 4.22 | 95.84 | 4.16 | 95.83 | 4.17 | 95.80 | 4.20 | 95.83 | 4.17 | 31,042 |
| Validation Dataset | 96.29 | 3.71 | 96.01 | 3.99 | 95.88 | 4.12 | 95.80 | 4.20 | 95.46 | 4.54 | 3,380 |
| Test Dataset | 95.54 | 4.46 | 95.34 | 4.66 | 95.54 | 4.46 | 95.88 | 4.12 | 96.01 | 3.99 | 3,881 |
| Total | | | | | | | | | | | 38,803 |

#### C.1.2.2 PhysioNet Mortality Prediction Challenge 2012 (P12) Dataset

After filtering out 12 entries lacking time series data, the PhysioNet Mortality Prediction Challenge 2012 (P12) contains data from 11,988 ICU patients. Each sample includes multivariate time series from 36 sensors (excluding weight), collected over the first 48 hours of the ICU stay. A static feature vector with nine demographic and clinical variables (e.g., age, gender) accompanies each sample. The binary prediction target denotes the ICU length of stay, where $\leq 3$ days is the negative class and $> 3$ days the positive class. The dataset is heavily imbalanced, as displayed in Table 12. To ensure a fair evaluation, the performance is averaged over five consistent data splits.

Table 12: P12 Label Distribution in percentage (%).

| | Split 0 | | Split 1 | | Split 2 | | Split 3 | | Split 4 | | Num. of Samples |
|---|---|---|---|---|---|---|---|---|---|---|---|
| | 0 | 1 | 0 | 1 | 0 | 1 | 0 | 1 | 0 | 1 | |
| Training Dataset | 85.60 | 14.40 | 85.99 | 14.01 | 85.77 | 14.23 | 85.83 | 14.17 | 86.08 | 13.92 | 9,590 |
| Validation Dataset | 85.65 | 14.35 | 83.99 | 16.01 | 85.40 | 14.60 | 85.32 | 14.68 | 85.82 | 14.18 | 1,199 |
| Test Dataset | 87.16 | 12.84 | 85.74 | 14.26 | 86.07 | 13.93 | 85.65 | 14.35 | 83.15 | 16.85 | 1,199 |
| Total | | | | | | | | | | | 11,988 |

### C.1.2.3 Physical Activity Monitoring (PAM) Dataset

The PAM dataset, derived from PAMAP2 (Physical Activity Monitoring), records physical activities of nine subjects using three inertial measurement units. To adapt it for irregular time series classification, the ninth subject, due to insufficient sensor readout length, is excluded. The continuous signals are segmented into samples with a time window of 600 and a $50\%$ overlap rate. Initially, PAM includes 18 daily activities, but those with fewer than 500 samples are excluded, leaving eight activities. After these modifications, the PAM dataset comprises 5,333 segments (samples) of sensory signals. Each sample is captured by 17 sensors and contains 600 continuous observations at a sampling frequency of 100 Hz. To simulate irregular time series data, $60\%$ of the observations are randomly removed. For fairness in comparison, the removed observations are randomly selected but consistent across all experimental settings and approaches. The PAM dataset is labeled into 8 classes, each representing a physical activity, and does not include static attributes. For more detailed descriptions please refer to [20]. The samples are roughly balanced across the 8 categories, as displayed in Table 13. To ensure a fair evaluation, the performance is averaged over five consistent data splits. The pre-processed data of PAMAP2 as well as the data splits can be accessed via the link provided in Table 8.

Table 13: PAM Label Distribution in percentage of Split 0 (in %).

| | 1 | 2 | 3 | 4 | 5 | 6 | 7 | 8 | Num. of Samples |
|---|---|---|---|---|---|---|---|---|---|
| Training Dataset | 22.08 | 11.77 | 10.31 | 11.86 | 15.56 | 5.91 | 11.67 | 10.83 | 4,266 |
| Validation Dataset | 24.95 | 11.26 | 9.01 | 12.20 | 15.20 | 4.69 | 11.44 | 11.26 | 533 |
| Test Dataset | 23.22 | 12.17 | 11.05 | 13.67 | 15.73 | 6.37 | 7.49 | 10.30 | 534 |
| Total | | | | | | | | | 5,333 |

### C.1.3 Regular Time Series Datasets

This section provides details about the regular sampled time series datasets used in Section 4.

### C.1.3.1 UEA Benchmark Datasets

The datasets from the UEA Archive [70] are commonly used to benchmark machine learning models on time series classification tasks. For a detailed overview of the datasets, please refer to Table 8. For all datasets separate testing and training datasets are provided, hence only the training set is split with a ratio of 9/1 into training and validation. This is executed consistently for all datasets from the UEA benchmark mentioned below. For all other datasets, the test set is also used for validation.

### C.1.3.1 Dataset Selection

We follow the curation of a diverse subset from Time Series Transformer (TST) [23] for the ablation study in Section 4.4.1. The diverse selection of datasets from the UEA benchmark ensures variability across key characteristics: sample dimensionality, sequence length, dataset size, and task difficulty. Our selection encompasses both high-performing ('easy') and low-performing ('challenging') datasets, as referenced by the baselines employed. Below is a brief justification for each selected multivariate dataset:

1. **Eigen Worms (EW)**: Low dimensionality, few samples, very long sequence length, moderate class count, relatively challenging dataset.

2. **Ethanol Concentration (EC)**: Low dimensionality, few samples, moderate sequence length, moderate class count, a challenging dataset [23].

3. **Face Detection (FD)**: Very high dimensionality, large sample size, short sequences, binary classification [23].

4. **Handwritting (HW)**: Low dimensionality, limited samples, moderate sequence length, many classes [23].

5. **Heartbeat (HB)**: High dimensionality, small sample size, moderate sequence length, binary classification [23].

6. **Japenese Vowels (JV)**: Variable sequence lengths, moderate dimensionality, few samples, moderate class count, baselines perform well [23].

7. **Pen Digits (PD)**: Low dimensionality, many samples, short sequence length, many classes, baselines perform well.

8. **PEMS-SF (PS)**: Extremely high dimensionality, few samples, moderate sequence length, moderate class count, baselines perform well [23].

9. **Self Regulation SCP1 (SCP1)**: Low dimensionality, few samples, long sequences, binary classification; baselines perform well [23].

10. **Self Regulation SCP2 (SCP2)**: Similar to SCP1 but with increased task complexity [23].

11. **Spoken Arabic Digits (SAD)**: Moderate dimensionality, large sample size, heterogeneous sequence lengths, moderate class count, baselines perform well [23].

12. **UWave Gesture Library (UW)**: Low dimensionality, few samples, moderate sequence length, moderate class count, baselines perform well [23].

## C.2 Anomaly Detection Time Series Datasets

This section introduces the benchmark datasets used to evaluate the performance for univariate time series anomaly detection.

### C.2.1 A Labeled Anomaly Detection Dataset (Yahoo) Webscope

Yahoo created a comprehensive public dataset, aiming to aid anomaly detection research [72]. This dataset includes both synthetic and real internet traffic data, with the latter manually labeled, acknowledging potential human error. Further it includes a variety of anomaly types such as outliers and change-points [24]. The dataset encompasses 367 hourly sampled time series with tagged anomaly points. The sequences are split as described in Section 4.2.

### C.2.2 KPI

The KPI dataset was released in an AIOPS challenge [71]. It includes multiple minutely sampled real KPI curves from many internet companies [24]. In total, it has 58 sequences, with the longest sequences exceeding 70,000 elements. The sequences are split as described in Section 4.2.

### C.2.3 Window Creation for Long Sequences

To create a sliding window mechanism that creates instance segments of a sequence, two variables are defined. $W$ defines the size of the window and $S$ the size of the stride. If $W \geq S$, there is no overlap between segments. The total number of segments is computed as:

$$N_w = \left\lfloor \frac{N - W}{S} \right\rfloor + 1 \tag{44}$$

where $N$ is the length of a sequence $\mathbf{S}^{(i)} \in \mathbb{R}^{N \times D}$. Then, a window can be defined as

$$\mathbf{W}^{(i)} = \mathbf{S}^{(i)}_{j*S:j*S+W,:} \tag{45}$$

where $j = \{0, 1, \ldots, N_w\}$ and $\mathbf{W}^{(i)} \in \mathbb{R}^{W \times D}$.

# D Experiments

## D.1 Evaluation Metrics

Typically, for benchmarking classification tasks, the accuracy on the test set is reported. In addition, for the DKT dataset, we want to focus on minimizing false positive predictions and thus record the $F_\beta$-score, Eq. (46) and (47), explicitly.

$$\text{F}_\beta\text{-score} = \frac{(1+\beta^2)\cdot TP}{(1+\beta^2)\cdot TP + FP + \beta^2 \cdot FN} \tag{46}$$

$$= \frac{(1+\beta^2)\cdot TP}{(1+\beta^2)\cdot TP + FP + \beta^2 \cdot FN} \quad\bigg|\cdot \frac{TP}{TP}$$

$$= \frac{(1+\beta^2)\cdot TP^2}{\beta^2 \cdot TP \cdot (TP+FN) + (TP+FP)\cdot TP} \quad\bigg|\cdot \frac{1/((TP+FN)\cdot(TP+FP))}{1/((TP+FN)\cdot(TP+FP))}$$

$$= (1+\beta^2)\cdot \frac{\overbrace{\frac{TP}{TP+FP}}^{\text{Precision}} \cdot \overbrace{\frac{TP}{TP+FN}}^{\text{Recall}}}{\beta^2 \cdot \frac{TP(TP+FN)}{(TP+FP)(TP+FN)} + \frac{TP(TP+FP)}{(TP+FP)(TP+FN)}}$$

$$= (1+\beta^2)\cdot \frac{\text{Precision}\cdot\text{Recall}}{\beta^2 \cdot \underbrace{\frac{TP}{TP+FP}}_{\text{Precision}} + \underbrace{\frac{TP}{TP+FN}}_{\text{Recall}}}$$

$$= (1+\beta^2)\cdot \frac{\text{Precision}\cdot\text{Recall}}{\beta^2 \cdot \text{Precision} + \text{Recall}} \tag{47}$$

The $\text{F}_\beta$-score balances precision and recall through the weighting parameter $\beta$. For $\beta = 1$, it equals the $\text{F}_1$-score. A $\beta$-value $< 1$ emphasizes precision, reducing false positives, while $\beta > 1$ prioritizes recall, reducing false negatives. We choose $\beta = 0.5$. Excellent $\text{F}_\beta$-scores range from $0.8$ - $0.9$, whereas scores below $0.5$ are considered poor.

Depending on the dataset, other metrics used for classification and anomaly detection tasks include $\text{F}_1$-Score, Precision, Recall, Area Under the Receiver Operating Characteristic (AUROC), Area Under the Precision-Recall Curve (AUPRC) and Mean Absolute Error (MAE).

For regression, we follow the 'average relative mean difference', $r_j$, the evaluation metric used in previous literature [21, 23]. For each model $j$ over $N$ datasets, the average relative mean difference is defined as:

$$r_j = \frac{1}{N}\sum_{i=1}^{N} \frac{R(i,j) - \bar{R}_i}{\bar{R}_i}, \tag{48}$$

and

$$\bar{R}_i = \frac{1}{M}\sum_{k=1}^{M} R(i,j), \tag{49}$$

where $M$ is the number of models, $R(i,j)$ is the Root Mean Squared Error (RMSE) of the model $j$ on dataset $i$.

## D.2 DKT Robustness Analysis

Table 19 documents the complete results presented in Table 1, detailing performance across five training seeds for various baseline models on the DKT dataset. Although STaRFormer is able to outperform the baseline models, some achieve nearly similar performance. Thus, we conduct further examinations to evaluate the performance.

We discovered that the labeling process during the trajectory recording leads to an overlap of positive and negative labels for some visually similar trajectories. This overlap creates a performance ceiling that we believe is inherent to the dataset. Despite efforts to overfit the model during training, the maximum accuracy attained was approximately 90%. This suggests that the performance metrics are approaching the upper limit, given the current data collection methods. Consequently, we performed a robustness analysis to explore not only the defined metrics but also the sensitivity of model predictions to potential noise from the sensors used for the data collection. In this analysis, we utilized the coefficient of variation (CV), as shown in Eq. (50), to assess the variability in the model's predictions.

$$\text{CV} = \frac{\sigma}{\mu} \tag{50}$$

**Experimental Setup.** We selected longer sequences from the DKT test set, specifically those where $\text{len}(\mathbf{S}^{(i)}) > 100$ elements. Then, Gaussian white noise is added to the final 10 and 30 elements of the selected sequences. Consequently, for each sequence, we obtained 10/30 additional corrupted sequences in the respective setups. We then evaluated all sequences and calculated their corresponding CV values.

**Results.** Table 14 presents the results of the robustness analysis, comparing four models under varying levels of input noise. The evaluation focuses on two primary metrics: CV, which reflects the stability of model predictions, and the Mean Absolute Error (MAE) between original and noisy predictions, which quantifies sensitivity to perturbations. A lower CV value is indicative of better inherent robustness, implying more consistent predictions across samples. Conversely, a lower MAE signifies that the model's outputs are less influenced by noise. However, excessively low MAE values may also suggest that a model is insensitive to real-world variability, potentially leading to underfitting or lack of generalization.

The results clearly show that all models experience a degradation in robustness as the noise level increases from 10 to 30 corrupted sequential elements. This trend is evident in the positive deltas across all metrics ($\Delta_{\text{original}}$, $\Delta_{\text{noisy}}$, and $\Delta_{\text{MAE}}$), signifying increased variability and error due to noise. Among the models evaluated, STaRFormer demonstrates superior robustness characteristics: yielding the lowest CV values for both original and noisy data for both perturbations (10 and 30), exhibiting the smallest deltas ($\Delta_{\text{original}} = 0.063$, $\Delta_{\text{noisy}} = 0.098$) which indicates that its predictions are least affected by increased noise, and reporting moderate MAE values (0.039 at 10 and 0.074 at 30) suggesting a balanced trade-off between robustness and sensitivity, avoiding excessive rigidity.

In contrast, the Transformer model shows the highest deltas and CV values across the evaluation, indicating the most substantial degradation in performance under noise. This emphasizes the advantages of our approach, demonstrating that our method facilitates the creation of more robust latent representations, which consequently enhances the overall robustness of the downstream task performance, even if the improvements in downstream task metrics are small. Additionally, it is notable that the LSTM model consistently exhibits the lowest MAE between the original and corrupted sequences under both perturbation levels (10, 30). This suggests a high degree of robustness to input noise. However, the minimal deviation in predictions may also reflect an excessive rigidity or insensitivity to input variability, potentially harming the model's ability to generalize effectively.

Overall, STaRFormer emerges as the most robust model, maintaining consistent prediction quality while allowing for some degree of variability, which is critical for generalization.

Table 14: Robustness analysis results.

| Method | CV 10 | | | CV 30 | | | Deltas ($\Delta$) | | |
| --- | --- | --- | --- | --- | --- | --- | --- | --- | --- |
| | original | noisy | MAE | original | noisy | MAE | $\Delta_{\text{original}}$ | $\Delta_{\text{noisy}}$ | $\Delta_{\text{MAE}}$ |
| LSTM | 0.845 | 0.847 | **0.003** | 0.959 | 0.993 | **0.035** | 0.114 | 0.146 | **0.032** |
| GRU | 0.810 | 0.846 | 0.036 | 0.949 | 1.020 | 0.071 | 0.139 | 0.174 | 0.035 |
| Transformer | 1.018 | 1.068 | 0.050 | 1.223 | 1.309 | 0.086 | 0.205 | 0.241 | 0.036 |
| STaRFormer | **0.765** | **0.804** | 0.039 | **0.828** | **0.902** | 0.074 | **0.063** | **0.098** | 0.035 |

## D.3 Experiment Runs

We use a Rate Scheduler (Reduce Learning Rate on Plateau Learning), Early Stopping, and the Adam optimizer for all experiments. All configurations of the model, the datasets, and all other relevant hyperparameters are extensively documented in the accompanying GitHub repository, `https://github.com/STaR-Former/starformer`, and can be found in the 'experiment/final' subfolder in 'configs'.

### D.3.1 Compute Resources and Execution Times

We conducted all experiments using Amazon EC2 instances (`https://aws.amazon.com/ec2/instance-types/?nc1=h_ls`), which offer a broad range of instance types with configurable

combinations of CPU, memory, storage, and networking capabilities. These allow for flexible resource allocation tailored to specific computational requirements. For this study, we primarily utilized AWS accelerated computing instances, specifically the P3 and G5 families.

Due to the asynchronous scheduling of experiments and the dynamic nature of cloud resource availability, we employed different GPU and CPU configurations depending on instance accessibility at the time of execution. Nevertheless, all experiments were run exclusively on AWS EC2 P3 or G5 instances, utilizing either an NVIDIA Tesla V100-SXM2-16GB or an NVIDIA A10G-24GB GPU. During the course of experimentation, AWS deprecated the P3 instance family, rendering them inaccessible for future runs. As a result, we transitioned to the G5 instance family. For a comprehensive overview of G5 instance specifications, please refer to the official AWS G5 documentation (`https://aws.amazon.com/ec2/instance-types/g5/`).

**Hyperparameter Tuning.**    We performed extensive hyperparameter tuning for all experiments involving STaRFormer to identify optimal configurations. For large-scale datasets, such as DKT, this process was computationally intensive and required sustained multi-GPU workloads over several days. Specifically, for DKT, we executed multiple sweep agents on an 8-GPU EC2 P3 instance over the span of one week to converge on the final configuration. In contrast, tuning for smaller datasets, particularly those in the UEA benchmark benchmark, required significantly less time. For example, in the case of Japenese Vowels (JV), a single sweep was completed in approximately five hours. All hyperparameter sweep configurations used in this study are available in the accompanying GitHub repository.

**Execution Times.**    The following table reports training durations, memory usage, and resource allocation for all executed runs, including both our model and baseline implementations. Note that many ablation studies reused already documented configurations with slight changes. Given the high similarity in resource profiles across these repeated runs and the large number of total ablations (approximately 210), only representative execution times are documented from the original configuration. Find the complete documentation for the individual execution times in Tables 15, 16, 17 and 18.

Note: In some instances, multiple experiments were simultaneously executed on the same GPU to optimize memory utilization. Consequently, resource contention led to increased training durations, with some recorded runtimes exceeding those expected under isolated, single-workload GPU execution.

Table 15: Combined training and testing times for non-stationary (DKT, GL) and irregularly sampled (P19, P12, PAM) datasets. All training times for a single GPU are reported to ensure a fair comparison. However, some are estimates (indicated by *), as they were trained using data-distributed parallel strategies on multi-GPU workloads.

| Dataset | Method | Hardware | Memory (GB) | | Splits (Seed) 0 (42) | 1 (123) | 2 (0) | 3 (63) | 4 (2024) | Average | Std. Dev. |
|---|---|---|---|---|---|---|---|---|---|---|---|
| DKT | RNN | A10G | 24 | Time (h) | 3.600 | 1.713 | 2.157 | 2.805 | 2.749 | 2.605 | 0.716 |
| | | | | Epochs | 54 | 42 | 45 | 47 | 46 | 46.800 | 4.438 |
| | | | | Memory Allocation | 0.115 | 0.059 | 0.107 | 0.167 | 0.164 | 0.123 | 0.045 |
| | LSTM | A10G | 24 | Time (h) | 4.773 | 5.225 | 4.588 | 4.041 | 4.672 | 4.660 | 0.424 |
| | | | | Epochs | 98 | 63 | 102 | 105 | 68 | 87.200 | 20.042 |
| | | | | Memory Allocation | 0.080 | 0.125 | 0.128 | 0.122 | 0.122 | 0.115 | 0.020 |
| | GRU | A10G | 24 | Time (h) | 4.805 | 3.180 | 4.798 | 4.630 | 2.851 | 4.053 | 0.957 |
| | | | | Epochs | 94 | 146 | 124 | 104 | 192 | 132.000 | 39.013 |
| | | | | Memory Allocation | 0.129 | 0.118 | 0.129 | 0.118 | 0.072 | 0.113 | 0.024 |
| | Transformer (TST) | Tesla V100 - SXM2 | 16 | Time (h) | 3.670 | 1.846 | 2.900 | 2.315 | 2.455 | 2.637 | 0.689 |
| | | | | Epochs | 98 | 65 | 105 | 83 | 87 | 87.600 | 15.356 |
| | | | | Memory Allocation | 0.118 | 0.118 | 0.103 | 0.118 | 0.118 | 0.115 | 0.006 |
| | TARNet | A10G | 24 | Time (h) | 28.318 | 14.239 | 22.375 | 17.861 | 18.943 | 20.347 | 5.317 |
| | | | | Epochs | 300 | 300 | 300 | 300 | 300 | 300.000 | 0.000 |
| | | | | Memory Allocation | 0.574 | 0.499 | 0.599 | 0.649 | 0.448 | 0.554 | 0.080 |
| | TimesURL | A10G | 24 | Time (h) | 118.398 | 114.323 | 115.659 | 115.687 | 115.605 | 115.934 | 1.493 |
| | | | | Epochs | 100 | 100 | 100 | 100 | 100 | 100.000 | 0.000 |
| | | | | Memory Allocation | 0.681 | 0.706 | 0.756 | 0.530 | 0.555 | 0.646 | 0.098 |
| | STaRFormer-RM* | Tesla V100 - SXM2 | 16 | Time (h) | 112.921 | 80.068 | 50.440 | 66.424 | 61.878 | 74.346 | 24.034 |
| | | | | Epochs | 95 | 78 | 80 | 89 | 69 | 82.200 | 10.085 |
| | | | | Memory Allocation | 0.939 | 0.484 | 0.516 | 0.939 | 0.516 | 0.679 | 0.238 |
| | STaRFormer* | Tesla V100 - SXM2 | 16 | Time (h) | 73.729 | 58.723 | 58.415 | 68.840 | 62.267 | 64.395 | 6.696 |
| | | | | Epochs | 102 | 84 | 88 | 82 | 100 | 91.200 | 9.230 |
| | | | | Memory Allocation | 0.846 | 0.566 | 0.941 | 0.939 | 0.484 | 0.755 | 0.215 |
| GL | Transformer (TST) | Tesla V100 - SXM2 | 16 | Time (h) | 0.419 | 0.244 | 0.272 | 0.320 | 0.294 | 0.310 | 0.067 |
| | | | | Epochs | 115 | 61 | 69 | 84 | 74 | 80.600 | 20.959 |
| | | | | Memory Allocation | 0.112 | 0.079 | 0.112 | 0.079 | 0.112 | 0.099 | 0.018 |
| | TARNet | A10G | 24 | Time (h) | 1.541 | - | - | - | - | 1.541 | - |
| | | | | Epochs | 200 | - | - | - | - | 200.000 | - |
| | | | | Memory Allocation | 0.574 | - | - | - | - | 0.574 | - |
| | TimesURL | A10G | 24 | Time (h) | 9.903 | - | - | - | - | 9.903 | - |
| | | | | Epochs | 300 | - | - | - | - | 300.000 | - |
| | | | | Memory Allocation | 0.383 | - | - | - | - | 0.383 | - |
| | STaRFormer-RM* | Tesla V100 - SXM2 | 16 | Time (h) | 0.854 | 1.287 | 1.343 | 1.210 | 0.838 | 1.107 | 0.242 |
| | | | | Epochs | 57 | 62 | 69 | 56 | 56 | 60.000 | 5.612 |
| | | | | Memory Allocation | 0.174 | 0.254 | 0.254 | 0.254 | 0.174 | 0.222 | 0.044 |
| | STaRFormer* | Tesla V100 - SXM2 | 16 | Time (h) | 1.117 | 1.081 | 1.105 | 1.752 | 1.442 | 1.299 | 0.293 |
| | | | | Epochs | 70 | 67 | 68 | 96 | 91 | 78.400 | 13.939 |
| | | | | Memory Allocation | 0.878 | 0.674 | 0.639 | 0.732 | 0.641 | 0.713 | 0.100 |
| P19 | Base | Tesla V100 - SXM2 | 16 | Time (h) | 1.077 | 2.328 | 2.217 | 2.475 | 2.305 | 2.080 | 0.569 |
| | | | | Epochs | 125 | 86 | 74 | 86 | 77 | 89.600 | 20.501 |
| | | | | Memory Allocation | 0.549 | 0.928 | 0.928 | 0.928 | 0.928 | 0.852 | 0.170 |
| | STaRFormer-RM* | Tesla V100 - SXM2 | 16 | Time (h) | 1.501 | 1.080 | 1.246 | 1.705 | 2.236 | 1.554 | 0.450 |
| | | | | Epochs | 72 | 52 | 60 | 82 | 98 | 72.800 | 18.144 |
| | | | | Memory Allocation | 0.186 | 0.188 | 0.188 | 0.186 | 0.189 | 0.187 | 0.001 |
| | STaRFormer* | Tesla V100 - SXM2 | 16 | Time (h) | 3.445 | 4.247 | 4.021 | 5.613 | 4.953 | 4.456 | 0.843 |
| | | | | Epochs | 106 | 136 | 124 | 76 | 158 | 140.000 | 27.604 |
| | | | | Memory Allocation | 0.259 | 0.260 | 0.260 | 0.259 | 0.260 | 0.260 | 0.000 |
| P12 | Base | Tesla V100 - SXM2 | 16 | Time (h) | 1.199 | 0.519 | 0.285 | 0.323 | 0.286 | 0.523 | 0.390 |
| | | | | Epochs | 102 | 46 | 39 | 53 | 82 | 64.400 | 26.633 |
| | | | | Memory Allocation | 0.928 | 0.860 | 0.799 | 0.791 | 0.658 | 0.807 | 0.100 |
| | STaRFormer-RM* | Tesla V100 - SXM2 | 16 | Time (h) | 0.930 | 0.831 | 0.441 | 0.516 | 0.405 | 0.625 | 0.240 |
| | | | | Epochs | 82 | 97 | 53 | 62 | 49 | 68.600 | 20.354 |
| | | | | Memory Allocation | 0.984 | 0.581 | 0.564 | 0.590 | 0.746 | 0.693 | 0.179 |
| | STaRFormer* | Tesla V100 - SXM2 | 16 | Time (h) | 0.820 | 1.161 | 2.417 | 1.149 | 1.640 | 1.438 | 0.621 |
| | | | | Epochs | 85 | 117 | 259 | 122 | 163 | 149.200 | 67.351 |
| | | | | Memory Allocation | 0.515 | 0.879 | 0.705 | 0.670 | 0.585 | 0.671 | 0.138 |
| PAM | Base | Tesla V100 - SXM2 | 16 | Time (h) | 0.166 | 0.078 | 0.122 | 0.160 | 0.166 | 0.138 | 0.039 |
| | | | | Epochs | 86 | 74 | 59 | 74 | 76 | 73.800 | 9.654 |
| | | | | Memory Allocation | 0.234 | 0.186 | 0.234 | 0.234 | 0.234 | 0.224 | 0.021 |
| | STaRFormer-RM* | Tesla V100 - SXM2 | 16 | Time (h) | 0.559 | 0.675 | 0.308 | 0.308 | 0.663 | 0.503 | 0.183 |
| | | | | Epochs | 80 | 101 | 100 | 103 | 101 | 97.000 | 9.566 |
| | | | | Memory Allocation | 0.268 | 0.268 | 0.099 | 0.099 | 0.266 | 0.200 | 0.092 |
| | STaRFormer* | Tesla V100 - SXM2 | 16 | Time (h) | 0.369 | 0.702 | 0.442 | 0.452 | 0.561 | 0.505 | 0.130 |
| | | | | Epochs | 88 | 170 | 103 | 103 | 132 | 119.200 | 32.568 |
| | | | | Memory Allocation | 0.371 | 0.371 | 0.371 | 0.371 | 0.371 | 0.371 | 0.000 |

Table 16: Combined training and testing times for datasets from the UEA and the anomaly detection benchmarks (Yahoo, KPI). All training times are reported for a single GPU to ensure a fair comparison.

| Dataset | Hardware | Memory (GB) | Method | Time (h) | Epochs | Memory Allocation |
|---|---|---|---|---|---|---|
| EW | Tesla V100 - SXM2 | 16 | Base | 0.015 | 44 | 0.097 |
| | | | STaRFormer-RM | 0.016 | 37 | 0.030 |
| | | | STaRFormer | 0.033 | 87 | 0.180 |
| EC | Tesla V100 - SXM2 | 16 | Base | 0.119 | 65 | 0.559 |
| | | | STaRFormer-RM | 0.296 | 73 | 0.700 |
| | | | STaRFormer | 0.035 | 71 | 0.557 |
| FD | Tesla V100 - SXM2 | 16 | Base | 0.416 | 87 | 0.559 |
| | | | STaRFormer-RM | 0.201 | 32 | 0.700 |
| | | | STaRFormer | 0.342 | 102 | 0.557 |
| HW | Tesla V100 - SXM2 | 16 | Base | 0.051 | 111 | 0.131 |
| | | | STaRFormer-RM | 0.104 | 197 | 0.142 |
| | | | STaRFormer | 0.114 | 162 | 0.277 |
| HB | Tesla V100 - SXM2 | 16 | Base | 0.018 | 50 | 0.345 |
| | | | STaRFormer-RM | 0.023 | 48 | 0.130 |
| | | | STaRFormer | 0.023 | 34 | 0.481 |
| JV | Tesla V100 - SXM2 | 16 | Base | 0.019 | 108 | 0.701 |
| | | | STaRFormer-RM | 0.068 | 232 | 0.122 |
| | | | STaRFormer | 0.041 | 103 | 0.126 |
| PD | Tesla V100 - SXM2 | 16 | Base | 0.199 | 113 | 0.031 |
| | | | STaRFormer-RM | 1.110 | 262 | 0.031 |
| | | | STaRFormer | 2.017 | 266 | 0.427 |
| PS | Tesla V100 - SXM2 | 16 | Base | 0.058 | 195 | 0.119 |
| | | | STaRFormer-RM | 0.123 | 177 | 0.154 |
| | | | STaRFormer | 0.159 | 291 | 0.237 |
| SCP1 | Tesla V100 - SXM2 | 16 | Base | 0.049 | 87 | 0.788 |
| | | | STaRFormer-RM | 0.118 | 112 | 0.609 |
| | | | STaRFormer | 0.043 | 103 | 0.245 |
| SCP2 | Tesla V100 - SXM2 | 16 | Base | 0.026 | 69 | 0.279 |
| | | | STaRFormer-RM | 0.052 | 32 | 0.831 |
| | | | STaRFormer | 0.054 | 42 | 0.442 |
| SAD | Tesla V100 - SXM2 | 16 | Base | 0.143 | 59 | 0.102 |
| | | | STaRFormer-RM | 0.690 | 51 | 0.175 |
| | | | STaRFormer | 1.567 | 105 | 0.145 |
| UW | Tesla V100 - SXM2 | 16 | Base | 0.034 | 142 | 0.838 |
| | | | STaRFormer-RM | 0.090 | 98 | 0.523 |
| | | | STaRFormer | 0.037 | 96 | 0.295 |
| Yahoo | A10G | 24 | Base | 0.036 | 39 | 0.601 |
| | | | STaRFormer-RM | 0.141 | 46 | 0.904 |
| | | | STaRFormer | 0.461 | 61 | 0.660 |
| KPI | A10G | 24 | Base | 0.967 | 102 | 0.422 |
| | | | STaRFormer-RM | 1.429 | 54 | 0.422 |
| | | | STaRFormer | 2.374 | 100 | 0.415 |

Table 17: Combined training and testing times of STaRFormer of remaining datasets from the UEA and TSR benchmarks. All training times are reported for a single GPU to ensure a fair comparison.

| Benchmark | Dataset | Hardware | Memory (GB) | Time (h) | Epochs | Memory Allocation (max 1) |
|---|---|---|---|---|---|---|
| UEA | AWR | A10G | 24 | 0.227 | 145 | 0.472 |
| | AF | A10G | 24 | 0.009 | 53 | 0.010 |
| | BM | A10G | 24 | 0.011 | 76 | 0.010 |
| | CT | A10G | 24 | 1.221 | 163 | 0.526 |
| | CK | A10G | 24 | 0.085 | 116 | 0.934 |
| | DDK | A10G | 24 | 0.024 | 36 | 0.634 |
| | EP | A10G | 24 | 0.045 | 56 | 0.397 |
| | ER | A10G | 24 | 0.103 | 113 | 0.383 |
| | FM | A10G | 24 | 0.052 | 36 | 0.619 |
| | HMD | A10G | 24 | 0.049 | 54 | 0.320 |
| | IW | A10G | 24 | 8.768 | 84 | 0.743 |
| | LI | A10G | 24 | 0.170 | 189 | 0.657 |
| | LSST | A10G | 24 | 0.360 | 45 | 0.284 |
| | MI | A10G | 24 | 0.031 | 17 | 0.927 |
| | NT | A10G | 24 | 0.117 | 156 | 0.265 |
| | PSp | A10G | 24 | 0.859 | 64 | 0.417 |
| | RS | A10G | 24 | 0.124 | 122 | 0.597 |
| | SWJ | A10G | 24 | 0.024 | 96 | 0.902 |
| TSR | AE | A10G | 24 | 0.085 | 827 | 0.089 |
| | AR | A10G | 24 | 22.039 | 200 | 0.298 |
| | BPM10 | A10G | 24 | 1.663 | 164 | 0.191 |
| | BPM25 | A10G | 24 | 1.108 | 127 | 0.150 |
| | BC | A10G | 24 | 2.144 | 263 | 0.529 |
| | BIDMCHR | A10G | 24 | 3.998 | 104 | 0.510 |
| | BIDMCRR | A10G | 24 | 1.249 | 110 | 0.460 |
| | BIDMCSPO2 | A10G | 24 | 2.733 | 104 | 0.340 |
| | C3M | A10G | 24 | 0.014 | 175 | 0.037 |
| | FM1 | A10G | 24 | 0.965 | 419 | 0.664 |
| | FM2 | A10G | 24 | 0.249 | 157 | 0.320 |
| | FM3 | A10G | 24 | 0.388 | 294 | 0.368 |
| | HPC1 | A10G | 24 | 0.786 | 146 | 0.670 |
| | HPC2 | A10G | 24 | 0.643 | 159 | 0.513 |
| | IEEEPPG | A10G | 24 | 2.439 | 173 | 0.705 |
| | LFMC | A10G | 24 | 1.650 | 190 | 0.358 |
| | NHS | A10G | 24 | 10.479 | 120 | 0.326 |
| | NTS | A10G | 24 | 9.910 | 107 | 0.586 |
| | PPG | A10G | 24 | 17.886 | 276 | 0.314 |

Table 18: Combined training and testing times of TARNet for the datasets from the TSR benchmarks. All training times are reported for a single GPU to ensure a fair comparison.

| Benchmark | Dataset | Hardware | Memory (GB) | Time (h) | Epochs | Memory Allocation (max 1) |
|---|---|---|---|---|---|---|
| | AE | A10G | 24 | 0.011 | 200 | - |
| | AR | A10G | 24 | 1.291 | 300 | - |
| | BPM10 | A10G | 24 | 0.336 | 200 | - |
| | BPM25 | A10G | 24 | 0.262 | 200 | - |
| | BC | A10G | 24 | 0.529 | 200 | - |
| | BIDMCHR | A10G | 24 | 2.094 | 100 | - |
| | BIDMCRR | A10G | 24 | 1.264 | 100 | - |
| | BIDMCSPO2 | A10G | 24 | 1.781 | 300 | - |
| | C3M | A10G | 24 | 0.017 | 300 | - |
| TSR | FM1 | A10G | 24 | 0.108 | 300 | - |
| | FM2 | A10G | 24 | 0.070 | 300 | - |
| | FM3 | A10G | 24 | 0.065 | 300 | - |
| | HPC1 | A10G | 24 | 0.462 | 100 | - |
| | HPC2 | A10G | 24 | 0.465 | 100 | - |
| | IEEEPPG | A10G | 24 | 1.745 | 200 | - |
| | LFMC | A10G | 24 | 0.921 | 300 | - |
| | NHS | A10G | 24 | 4.581 | 300 | - |
| | NTS | A10G | 24 | 3.709 | 300 | - |
| | PPG | A10G | 24 | 4.773 | 300 | - |

### D.3.2 DKT

Table 19: Run documentation of the methods trained on the DKT dataset. The respective seeds for each run are stated in the parenthesis in the header row.

| Method | Metrics | 0 (42) | 1 (123) | 2 (0) | 3 (63) | 4 (2024) | Median | Average | Std. Dev. |
|---|---|---|---|---|---|---|---|---|---|
| RNN Baseline | Accuracy | 0.74794 | 0.76239 | 0.76147 | 0.75659 | 0.73939 | 0.755 | 0.754 | 0.010 |
| | $F_{0.5}$-Score | 0.74776 | 0.76224 | 0.76288 | 0.75698 | 0.73917 | 0.755 | 0.754 | 0.010 |
| LSTM Baseline | Accuracy | 0.83982 | 0.84202 | 0.84576 | 0.84732 | 0.8438 | 0.844 | 0.844 | 0.003 |
| | $F_{0.5}$-Score | 0.84046 | 0.84194 | 0.84581 | 0.84359 | 0.84383 | 0.844 | 0.843 | 0.002 |
| GRU Baseline | Accuracy | 0.84088 | 0.8399 | 0.83505 | 0.84337 | 0.84284 | 0.841 | 0.840 | 0.003 |
| | $F_{0.5}$-Score | 0.84097 | 0.8399 | 0.83547 | 0.84323 | 0.84282 | 0.841 | 0.840 | 0.003 |
| Transformer | Accuracy | 0.84847 | 0.84589 | 0.84962 | 0.84851 | 0.85059 | 0.849 | 0.849 | 0.002 |
| | $F_{0.5}$-Score | 0.84855 | 0.84577 | 0.84953 | 0.84844 | 0.8505 | 0.849 | 0.849 | 0.002 |
| TARNet | Accuracy | 0.78050 | 0.76269 | 0.78482 | 0.78121 | 0.79360 | 0.781 | 0.781 | 0.011 |
| | $F_{0.5}$-Score | 0.78408 | 0.76311 | 0.78469 | 0.78152 | 0.79545 | 0.784 | 0.782 | 0.012 |
| TimesURL | Accuracy | 0.72720 | 0.7261 | 0.7244 | 0.7205 | 0.7207 | 0.724 | 0.724 | 0.003 |
| | $F_{0.5}$-Score | - | - | - | - | - | - | - | - |
| STaRFormer with RM | Accuracy | 0.84535 | 0.84509 | 0.84687 | 0.84477 | 0.84408 | 0.845 | 0.845 | 0.001 |
| | $F_{0.5}$-Score | 0.84535 | 0.8448 | 0.84675 | 0.8447 | 0.84405 | 0.845 | 0.845 | 0.001 |
| STaRFormer with DAReM | Accuracy | 0.85498 | 0.85069 | 0.8493 | 0.85366 | 0.84931 | 0.851 | 0.852 | 0.003 |
| | $F_{0.5}$-Score | 0.8549 | 0.8507 | 0.84916 | 0.85355 | 0.84952 | 0.851 | 0.852 | 0.003 |

Table 20: Run documentation of the ablation study on the DKT dataset evaluating the impact of our semi-supervised CL approach on the model performance. The respective seeds for each run are stated in the parenthesis in the header row.

| Ablation | Metrics | 0 (42) | 1 (123) | 2 (0) | 3 (63) | 4 (2024) | Median | Average | Std. Dev. |
|---|---|---|---|---|---|---|---|---|---|
| semi-supervised ($\lambda_{CL} \approx 0.796$) | Accuracy | 0.85498 | 0.85069 | 0.8493 | 0.85366 | 0.84931 | 0.851 | 0.852 | 0.003 |
| | $F_{0.5}$-Score | 0.8549 | 0.8507 | 0.84916 | 0.85355 | 0.84952 | 0.851 | 0.852 | 0.003 |
| w/o self-supervised ($\lambda_{CL} \approx 0.796$) | Accuracy | 0.84916 | 0.84950 | 0.8445 | 0.84680 | 0.84917 | 0.848 | 0.848 | 0.002 |
| | $F_{0.5}$-Score | 0.84991 | 0.84944 | 0.8444 | 0.84663 | 0.84917 | 0.849 | 0.848 | 0.002 |
| w/o supervised ($\lambda_{CL} \approx 0.796$) | Accuracy | 0.84782 | 0.84824 | 0.84742 | 0.84762 | 0.84865 | 0.848 | 0.848 | 0.001 |
| | $F_{0.5}$-Score | 0.84770 | 0.84818 | 0.84371 | 0.84760 | 0.84854 | 0.848 | 0.847 | 0.002 |
| semi-supervised ($\lambda_{CL} = 0.1$) | Accuracy | 0.84943 | 0.84861 | 0.84736 | 0.84613 | 0.84908 | 0.848 | 0.848 | 0.001 |
| | $F_{0.5}$-Score | 0.84000 | 0.84874 | 0.84761 | 0.84593 | 0.84897 | 0.847 | 0.846 | 0.004 |
| semi-supervised ($\lambda_{CL} = 1.0$) | Accuracy | 0.85223 | 0.85337 | 0.85004 | 0.85030 | 0.84819 | 0.851 | 0.851 | 0.002 |
| | $F_{0.5}$-Score | 0.85227 | 0.85342 | 0.84994 | 0.85023 | 0.84820 | 0.851 | 0.851 | 0.002 |
| semi-supervised ($\lambda_{CL} = 5.0$) | Accuracy | 0.84629 | 0.84900 | 0.84900 | 0.85030 | 0.85132 | 0.849 | 0.849 | 0.002 |
| | $F_{0.5}$-Score | 0.84620 | 0.84891 | 0.84887 | 0.85023 | 0.85123 | 0.849 | 0.849 | 0.002 |
| semi-supervised ($\lambda_{CL} = 10.0$) | Accuracy | 0.84525 | 0.84455 | 0.84567 | 0.84309 | 0.85009 | 0.845 | 0.846 | 0.003 |
| | $F_{0.5}$-Score | 0.84517 | 0.84454 | 0.84553 | 0.84299 | 0.85000 | 0.845 | 0.846 | 0.003 |

Table 21: Run documentation of the ablation study evaluating how the size of the masked regions affects the model performance on the DKT dataset. The respective seeds for each run are stated in the parenthesis in the header row.

| # | Ablation | | | Metrics | 0 (42) | 1 (123) | 2 (0) | 3 (63) | 4 (2024) | Median | Average | Std. Dev. |
|---|---|---|---|---|---|---|---|---|---|---|---|---|
| | $\varphi$ | $\zeta$ | $\gamma$ | | | | | | | | | |
| default | 0.427 | 0.2 | 0.25 | Accuracy | 0.85498 | 0.85069 | 0.8493 | 0.85366 | 0.84931 | 0.852 | 0.852 | 0.003 |
| | | | | $F_{0.5}$-Score | 0.8549 | 0.8507 | 0.84916 | 0.85355 | 0.84952 | 0.852 | 0.852 | 0.003 |
| 1 | 0.427 | 0.2 | 0.0 | Accuracy | 0.84898 | 0.85157 | 0.84685 | 0.84893 | 0.85129 | 0.850 | 0.850 | 0.002 |
| | | | | $F_{0.5}$-Score | 0.84895 | 0.85139 | 0.84671 | 0.84883 | 0.8518 | 0.850 | 0.850 | 0.002 |
| 2 | 0.427 | 0.2 | 0.05 | Accuracy | 0.84958 | 0.84961 | 0.84611 | 0.85328 | 0.84941 | 0.850 | 0.850 | 0.003 |
| | | | | $F_{0.5}$-Score | 0.84593 | 0.84594 | 0.84608 | 0.85324 | 0.84937 | 0.848 | 0.848 | 0.003 |
| 3 | 0.427 | 0.2 | 0.10 | Accuracy | 0.84890 | 0.85015 | 0.84488 | 0.85408 | 0.84799 | 0.849 | 0.849 | 0.003 |
| | | | | $F_{0.5}$-Score | 0.84886 | 0.85005 | 0.8488 | 0.85403 | 0.84794 | 0.850 | 0.849 | 0.002 |
| 4 | 0.427 | 0.2 | 0.15 | Accuracy | 0.85227 | 0.85061 | 0.84839 | 0.84867 | 0.85171 | 0.850 | 0.850 | 0.002 |
| | | | | $F_{0.5}$-Score | 0.85216 | 0.85059 | 0.84849 | 0.84854 | 0.8516 | 0.850 | 0.850 | 0.002 |
| 4 | 0.427 | 0.2 | 0.20 | Accuracy | 0.85221 | 0.84962 | 0.85106 | 0.84994 | 0.85138 | 0.851 | 0.851 | 0.001 |
| | | | | $F_{0.5}$-Score | 0.85223 | 0.84957 | 0.85114 | 0.84978 | 0.85132 | 0.851 | 0.851 | 0.001 |
| 6 | 0.427 | 0.2 | 0.25 | Accuracy | 0.85498 | 0.85069 | 0.8493 | 0.85366 | 0.84931 | 0.852 | 0.852 | 0.003 |
| | | | | $F_{0.5}$-Score | 0.8549 | 0.8507 | 0.84916 | 0.85355 | 0.84952 | 0.852 | 0.852 | 0.003 |
| 7 | 0.427 | 0.2 | 0.30 | Accuracy | 0.85126 | 0.84934 | 0.84974 | 0.849 | 0.8518 | 0.850 | 0.850 | 0.001 |
| | | | | $F_{0.5}$-Score | 0.8512 | 0.84929 | 0.84963 | 0.84888 | 0.85176 | 0.850 | 0.850 | 0.001 |

### D.3.3 Baseline Implementation

We selected two distinct state-of-the-art methodologies from literature to serve as additional baseline methods on the DKT dataset. Specifically, we chose Task-Aware Reconstruction for Time Series Transformer (TARNet) [21] and Self-Supervised Contrastive Learning for Universal Time Series Representation Learning (TimesURL) [27]. To utilize the official code baselines, we adapted our data loading procedures accordingly. Due to the absence of specified hyperparameters, for the Transformer backend in TARNet, we applied the same hyperparameters as those used for STaRFormer. For TimesURL, we employed the model's default parameters. It is important to note that TimesURL inherently utilizes a grid search strategy to find the optimal the Support Vector Machines (SVM) for the downstream task. For the DKT dataset, as the grid search is quite expensive, we limited it to two folds instead of the default five.

### D.3.4 Geolife

Table 22: Run documentation of the ablation study on the three ablations of STaRFormer on the GL dataset [63]; (i) (Base), (ii) STaRFormer-RM and (iii) STaRFormer. Here only the architecture of the model is changed, keeping everything else fixed. The respective seeds for each run are stated in the parenthesis in the header row.

| Ablation | Metrics | 0 (42) | 1 (123) | 2 (0) | 3 (63) | 4 (2024) | Median | Average | Std. Dev. |
|---|---|---|---|---|---|---|---|---|---|
| Base | Accuracy | 0.88614 | 0.89796 | 0.87687 | 0.86696 | 0.87750 | 0.878 | 0.881 | 0.012 |
| | $F_{0.5}$-Score | 0.86227 | 0.87591 | 0.84978 | 0.84574 | 0.85735 | 0.857 | 0.858 | 0.012 |
| STaRFormer with RM | Accuracy | 0.91518 | 0.90074 | 0.88235 | 0.89093 | 0.8825 | 0.891 | 0.894 | 0.014 |
| | $F_{0.5}$-Score | 0.90047 | 0.88006 | 0.85443 | 0.8704 | 0.85923 | 0.870 | 0.873 | 0.018 |
| STaRFormer with DAReM | Accuracy | 0.93238 | 0.89904 | 0.89904 | 0.89183 | 0.89625 | 0.899 | 0.904 | 0.016 |
| | $F_{0.5}$-Score | 0.91589 | 0.87505 | 0.87469 | 0.87114 | 0.87595 | 0.875 | 0.883 | 0.019 |

Table 23: Run documentation of the ablation study evaluating the impact of our semi-supervised CL approach on the model performance on the GL dataset [63]. The respective seeds for each run are stated in the parenthesis in the header row.

| Ablation | Metrics | 0 (42) | 1 (123) | 2 (0) | 3 (63) | 4 (2024) | Median | Average | Std. Dev. |
|---|---|---|---|---|---|---|---|---|---|
| semi-supervised ($\lambda_{CL} \approx 0.773$) | Accuracy | 0.93238 | 0.89904 | 0.89904 | 0.89183 | 0.89625 | 0.899 | 0.904 | 0.016 |
| | $F_{0.5}$-Score | 0.91586 | 0.87505 | 0.87469 | 0.87114 | 0.87595 | 0.875 | 0.883 | 0.019 |
| w/o self-supervised ($\lambda_{CL} \approx 0.773$) | Accuracy | 0.92188 | 0.88882 | 0.88882 | 0.89964 | 0.90125 | 0.900 | 0.900 | 0.014 |
| | $F_{0.5}$-Score | 0.89874 | 0.86552 | 0.86048 | 0.87804 | 0.88177 | 0.878 | 0.877 | 0.015 |
| w/o supervised ($\lambda_{CL} \approx 0.773$) | Accuracy | 0.92175 | 0.87921 | 0.89423 | 0.88942 | 0.89125 | 0.891 | 0.895 | 0.016 |
| | $F_{0.5}$-Score | 0.90169 | 0.87604 | 0.86763 | 0.86696 | 0.87048 | 0.870 | 0.877 | 0.014 |
| semi-supervised ($\lambda_{CL} = 0.1$) | Accuracy | 0.93125 | 0.89904 | 0.89363 | 0.88522 | 0.89063 | 0.894 | 0.900 | 0.018 |
| | $F_{0.5}$-Score | 0.91487 | 0.87578 | 0.86639 | 0.86519 | 0.87052 | 0.871 | 0.879 | 0.021 |
| semi-supervised ($\lambda_{CL} = 1.0$) | Accuracy | 0.92452 | 0.89483 | 0.90144 | 0.89844 | 0.89000 | 0.898 | 0.902 | 0.013 |
| | $F_{0.5}$-Score | 0.90499 | 0.87011 | 0.87759 | 0.87942 | 0.86601 | 0.878 | 0.880 | 0.015 |
| semi-supervised ($\lambda_{CL} = 5.0$) | Accuracy | 0.92925 | 0.90264 | 0.90505 | 0.89663 | 0.90438 | 0.904 | 0.908 | 0.013 |
| | $F_{0.5}$-Score | 0.91513 | 0.88069 | 0.88002 | 0.87347 | 0.88488 | 0.881 | 0.887 | 0.016 |
| semi-supervised ($\lambda_{CL} = 10.0$) | Accuracy | 0.91838 | 0.90565 | 0.90385 | 0.90925 | 0.89125 | 0.906 | 0.906 | 0.010 |
| | $F_{0.5}$-Score | 0.90368 | 0.88400 | 0.87978 | 0.88987 | 0.86998 | 0.884 | 0.885 | 0.013 |

Table 24: Run documentation of the ablation study evaluating how the size of the masked regions affects the model performance on the GL dataset [63]. The respective seeds for each run are stated in the parenthesis in the header row.

| # | Ablation | | | Metrics | 0 (42) | 1 (123) | 2 (0) | 3 (63) | 4 (2024) | Median | Average | Std. Dev. |
| | $\varphi$ | $\zeta$ | $\gamma$ | | | | | | | | | |
|---|---|---|---|---|---|---|---|---|---|---|---|---|
| default | 0.399 | 0.1 | 0.05 | Accuracy | 0.93238 | 0.89904 | 0.89904 | 0.89183 | 0.89625 | 0.904 | 0.904 | 0.016 |
| | | | | $F_{0.5}$-Score | 0.91586 | 0.87505 | 0.87469 | 0.87114 | 0.87595 | 0.883 | 0.883 | 0.019 |
| 1 | 0.399 | 0.1 | 0.0 | Accuracy | 0.92875 | 0.88041 | 0.89183 | 0.89844 | 0.88875 | 0.898 | 0.898 | 0.019 |
| | | | | $F_{0.5}$-Score | 0.91007 | 0.87835 | 0.86425 | 0.87641 | 0.86787 | 0.879 | 0.879 | 0.018 |
| 2 | 0.399 | 0.1 | 0.05 | Accuracy | 0.93238 | 0.89904 | 0.89904 | 0.89183 | 0.89625 | 0.904 | 0.904 | 0.016 |
| | | | | $F_{0.5}$-Score | 0.91586 | 0.87505 | 0.87469 | 0.87114 | 0.87595 | 0.883 | 0.883 | 0.019 |
| 3 | 0.399 | 0.1 | 0.10 | Accuracy | 0.92375 | 0.90352 | 0.89543 | 0.89663 | 0.89375 | 0.903 | 0.903 | 0.012 |
| | | | | $F_{0.5}$-Score | 0.90316 | 0.88335 | 0.86971 | 0.87761 | 0.87655 | 0.882 | 0.882 | 0.013 |
| 4 | 0.399 | 0.1 | 0.15 | Accuracy | 0.92875 | 0.90445 | 0.89543 | 0.89603 | 0.89063 | 0.903 | 0.903 | 0.015 |
| | | | | $F_{0.5}$-Score | 0.90977 | 0.88539 | 0.86977 | 0.87752 | 0.86884 | 0.882 | 0.882 | 0.017 |
| 5 | 0.399 | 0.1 | 0.20 | Accuracy | 0.91750 | 0.90565 | 0.89483 | 0.89363 | 0.89312 | 0.901 | 0.901 | 0.011 |
| | | | | $F_{0.5}$-Score | 0.89494 | 0.8839 | 0.86964 | 0.87391 | 0.87221 | 0.879 | 0.879 | 0.010 |
| 6 | 0.399 | 0.1 | 0.25 | Accuracy | 0.92813 | 0.88642 | 0.89663 | 0.89724 | 0.89625 | 0.901 | 0.901 | 0.016 |
| | | | | $F_{0.5}$-Score | 0.90813 | 0.88427 | 0.8712 | 0.87805 | 0.87629 | 0.884 | 0.884 | 0.014 |
| 7 | 0.399 | 0.1 | 0.30 | Accuracy | 0.92287 | 0.90204 | 0.90144 | 0.89784 | 0.88875 | 0.903 | 0.903 | 0.013 |
| | | | | $F_{0.5}$-Score | 0.90571 | 0.88161 | 0.87608 | 0.87777 | 0.86723 | 0.882 | 0.882 | 0.014 |

### D.3.5 PAM

Table 25: Run documentation of the ablation study on the three ablations of STaRFormer on the PAM dataset [68]; (i) Base, (ii) STaRFormer-RM and (iii) STaRFormer. Here only the architecture of the model is changed, keeping everything else fixed.

| Ablation | Metrics | 0 | 1 | 2 | 3 | 4 | Median | Average | Std. Dev. |
|---|---|---|---|---|---|---|---|---|---|
| Base | Accuracy | 0.97917 | 0.96875 | 0.96181 | 0.96544 | 0.94476 | 0.965 | 0.964 | 0.013 |
| | Precision | 0.98309 | 0.97635 | 0.96303 | 0.97274 | 0.94997 | 0.973 | 0.969 | 0.013 |
| | Recall | 0.97998 | 0.96654 | 0.97068 | 0.97579 | 0.95619 | 0.971 | 0.970 | 0.009 |
| | $F_1$-Score | 0.98184 | 0.9705 | 0.96647 | 0.97414 | 0.95184 | 0.971 | 0.969 | 0.011 |
| STaRFormer with RM | Accuracy | 0.97812 | 0.96122 | 0.97031 | 0.9625 | 0.94901 | 0.963 | 0.964 | 0.011 |
| | Precision | 0.978 | 0.96542 | 0.9654 | 0.9687 | 0.96077 | 0.965 | 0.968 | 0.006 |
| | Recall | 0.97711 | 0.97067 | 0.97067 | 0.96327 | 0.95837 | 0.971 | 0.968 | 0.007 |
| | $F_1$-Score | 0.97735 | 0.96777 | 0.96777 | 0.96591 | 0.95893 | 0.968 | 0.968 | 0.007 |
| STaRFormer with DAReM | Accuracy | 0.98307 | 0.98047 | 0.97786 | 0.96011 | 0.97917 | 0.979 | 0.976 | 0.009 |
| | Precision | 0.97796 | 0.97683 | 0.97132 | 0.97032 | 0.96945 | 0.971 | 0.973 | 0.004 |
| | Recall | 0.98016 | 0.9745 | 0.97467 | 0.97275 | 0.97717 | 0.975 | 0.976 | 0.003 |
| | $F_1$-Score | 0.97893 | 0.97542 | 0.97278 | 0.97146 | 0.97299 | 0.973 | 0.974 | 0.003 |

Table 26: Run documentation of the ablation study evaluating the impact of our semi-supervised CL approach on the model performance on the PAM dataset [68].

| Ablation | Metrics | 0 | 1 | 2 | 3 | 4 | Median | Average | Std. Dev. |
|---|---|---|---|---|---|---|---|---|---|
| semi-supervised ($\lambda_{CL} \approx 0.567$) | Accuracy | 0.98307 | 0.98407 | 0.97786 | 0.96011 | 0.97917 | 0.979 | 0.976 | 0.009 |
| | Precision | 0.97796 | 0.97683 | 0.97132 | 0.97032 | 0.96945 | 0.971 | 0.973 | 0.004 |
| | Recall | 0.98016 | 0.9745 | 0.97467 | 0.97275 | 0.97717 | 0.975 | 0.976 | 0.003 |
| | $F_1$-Score | 0.97893 | 0.97452 | 0.97278 | 0.97146 | 0.97299 | 0.973 | 0.974 | 0.003 |
| w/o self-supervised ($\lambda_{CL} \approx 0.567$) | Accuracy | 0.97786 | 0.96402 | 0.97656 | 0.96532 | 0.9272 | 0.965 | 0.962 | 0.021 |
| | Precision | 0.97559 | 0.97403 | 0.97192 | 0.97513 | 0.95906 | 0.974 | 0.971 | 0.007 |
| | Recall | 0.97187 | 0.9711 | 0.96882 | 0.97989 | 0.96069 | 0.971 | 0.970 | 0.007 |
| | $F_1$-Score | 0.97349 | 0.97201 | 0.97013 | 0.97708 | 0.95936 | 0.972 | 0.970 | 0.007 |
| w/o supervised ($\lambda_{CL} \approx 0.567$) | Accuracy | 0.98047 | 0.95017 | 0.98177 | 0.96271 | 0.95017 | 0.963 | 0.965 | 0.016 |
| | Precision | 0.97777 | 0.97697 | 0.97531 | 0.97483 | 0.9708 | 0.975 | 0.975 | 0.003 |
| | Recall | 0.97677 | 0.97115 | 0.98065 | 0.97062 | 0.97006 | 0.971 | 0.974 | 0.005 |
| | $F_1$-Score | 0.97722 | 0.97388 | 0.9777 | 0.97265 | 0.96993 | 0.974 | 0.974 | 0.003 |
| semi-supervised ($\lambda_{CL} = 0.1$) | Accuracy | 0.95147 | 0.94709 | 0.97526 | 0.95881 | 0.97786 | 0.959 | 0.962 | 0.014 |
| | Precision | 0.96899 | 0.95585 | 0.96865 | 0.96658 | 0.97207 | 0.969 | 0.966 | 0.006 |
| | Recall | 0.97509 | 0.95055 | 0.97504 | 0.97247 | 0.96874 | 0.972 | 0.968 | 0.010 |
| | $F_1$-Score | 0.97161 | 0.95235 | 0.97129 | 0.96895 | 0.97003 | 0.970 | 0.967 | 0.008 |
| semi-supervised ($\lambda_{CL} = 1.0$) | Accuracy | 0.97135 | 0.97786 | 0.98047 | 0.96532 | 0.96402 | 0.971 | 0.972 | 0.007 |
| | Precision | 0.97248 | 0.97765 | 0.97733 | 0.97445 | 0.96996 | 0.974 | 0.974 | 0.003 |
| | Recall | 0.96366 | 0.96661 | 0.97721 | 0.97883 | 0.97242 | 0.972 | 0.972 | 0.007 |
| | $F_1$-Score | 0.96764 | 0.97155 | 0.97718 | 0.97635 | 0.97107 | 0.972 | 0.973 | 0.004 |
| semi-supervised ($\lambda_{CL} = 5.0$) | Accuracy | 0.92803 | 0.96922 | 0.98438 | 0.96792 | 0.98307 | 0.969 | 0.967 | 0.023 |
| | Precision | 0.95442 | 0.98047 | 0.98276 | 0.97708 | 0.97937 | 0.979 | 0.975 | 0.012 |
| | Recall | 0.9394 | 0.97688 | 0.97828 | 0.98013 | 0.97473 | 0.977 | 0.970 | 0.017 |
| | $F_1$-Score | 0.94606 | 0.9785 | 0.98039 | 0.97846 | 0.97682 | 0.978 | 0.972 | 0.015 |
| semi-supervised ($\lambda_{CL} = 10.0$) | Accuracy | 0.94886 | 0.96141 | 0.98698 | 0.96922 | 0.98568 | 0.969 | 0.970 | 0.016 |
| | Precision | 0.96938 | 0.96931 | 0.98449 | 0.97912 | 0.98321 | 0.979 | 0.977 | 0.007 |
| | Recall | 0.96927 | 0.96652 | 0.98293 | 0.98193 | 0.97697 | 0.977 | 0.976 | 0.007 |
| | $F_1$-Score | 0.96907 | 0.96733 | 0.98352 | 0.98043 | 0.97947 | 0.979 | 0.976 | 0.007 |

Table 27: Run documentation of the ablation study evaluating how the size of the masked regions affects the model performance on the PAM dataset [68].

| # | Ablation | | | Metrics | 0 | 1 | 2 | 3 | 4 | Median | Average | Std. Dev. |
|---|---|---|---|---|---|---|---|---|---|---|---|---|
| | $\varphi$ | $\zeta$ | $\gamma$ | | | | | | | | | |
| default | 0.207 | 0.3 | 0.10 | Accuracy | 0.98307 | 0.98047 | 0.97786 | 0.96011 | 0.97917 | 0.976 | 0.976 | 0.009 |
| | | | | Precision | 0.97796 | 0.97683 | 0.97132 | 0.97032 | 0.96945 | 0.973 | 0.973 | 0.004 |
| | | | | Recall | 0.98016 | 0.9745 | 0.97467 | 0.97275 | 0.97717 | 0.976 | 0.976 | 0.003 |
| | | | | $F_1$-Score | 0.97893 | 0.97542 | 0.97278 | 0.97146 | 0.97299 | 0.974 | 0.974 | 0.003 |
| 1 | 0.207 | 0.3 | 0.0 | Accuracy | 0.97656 | 0.96251 | 0.97786 | 0.96662 | 0.96402 | 0.967 | 0.970 | 0.007 |
| | | | | Precision | 0.97485 | 0.9759 | 0.97289 | 0.97544 | 0.97176 | 0.975 | 0.974 | 0.002 |
| | | | | Recall | 0.97116 | 0.96728 | 0.9759 | 0.98193 | 0.96955 | 0.971 | 0.973 | 0.006 |
| | | | | $F_1$-Score | 0.97282 | 0.97135 | 0.9742 | 0.97817 | 0.97045 | 0.973 | 0.973 | 0.003 |
| 2 | 0.207 | 0.3 | 0.05 | Accuracy | 0.96875 | 0.97656 | 0.96141 | 0.94105 | 0.94235 | 0.961 | 0.958 | 0.016 |
| | | | | Precision | 0.96279 | 0.97400 | 0.96650 | 0.96174 | 0.96014 | 0.965 | 0.965 | 0.006 |
| | | | | Recall | 0.96229 | 0.96516 | 0.97287 | 0.96853 | 0.95972 | 0.966 | 0.966 | 0.005 |
| | | | | $F_1$-Score | 0.96450 | 0.96917 | 0.96938 | 0.96449 | 0.95961 | 0.965 | 0.965 | 0.004 |
| 3 | 0.207 | 0.3 | 0.10 | Accuracy | 0.98307 | 0.98047 | 0.97786 | 0.96011 | 0.97917 | 0.976 | 0.976 | 0.009 |
| | | | | Precision | 0.97796 | 0.97683 | 0.97132 | 0.97032 | 0.96945 | 0.973 | 0.973 | 0.004 |
| | | | | Recall | 0.98016 | 0.9745 | 0.97467 | 0.97275 | 0.97717 | 0.976 | 0.976 | 0.003 |
| | | | | $F_1$-Score | 0.97893 | 0.97542 | 0.97278 | 0.97146 | 0.97299 | 0.974 | 0.974 | 0.003 |
| 4 | 0.207 | 0.3 | 0.15 | Accuracy | 0.97266 | 0.96271 | 0.98828 | 0.95881 | 0.97183 | 0.972 | 0.971 | 0.011 |
| | | | | Precision | 0.9703 | 0.9747 | 0.98254 | 0.96929 | 0.97944 | 0.975 | 0.975 | 0.006 |
| | | | | Recall | 0.96576 | 0.96958 | 0.98727 | 0.96638 | 0.9842 | 0.970 | 0.975 | 0.010 |
| | | | | $F_1$-Score | 0.96763 | 0.97153 | 0.98478 | 0.96737 | 0.98167 | 0.972 | 0.975 | 0.008 |
| 5 | 0.207 | 0.3 | 0.20 | Accuracy | 0.96532 | 0.96122 | 0.97031 | 0.9625 | 0.94901 | 0.963 | 0.962 | 0.008 |
| | | | | Precision | 0.96551 | 0.97312 | 0.96542 | 0.9687 | 0.96077 | 0.966 | 0.967 | 0.005 |
| | | | | Recall | 0.97354 | 0.96209 | 0.97067 | 0.96372 | 0.95837 | 0.964 | 0.966 | 0.006 |
| | | | | $F_1$-Score | 0.96855 | 0.96711 | 0.96777 | 0.96591 | 0.95893 | 0.967 | 0.966 | 0.004 |
| 6 | 0.207 | 0.3 | 0.25 | Accuracy | 0.97917 | 0.96122 | 0.97031 | 0.9625 | 0.94901 | 0.963 | 0.964 | 0,011 |
| | | | | Precision | 0.97775 | 0.97312 | 0.96542 | 0.9687 | 0.96077 | 0.969 | 0.969 | 0,007 |
| | | | | Recall | 0.97103 | 0.96209 | 0.97067 | 0.96372 | 0.95837 | 0.964 | 0.965 | 0,006 |
| | | | | $F_1$-Score | 0.9742 | 0.96771 | 0.96777 | 0.96591 | 0.95893 | 0.968 | 0.967 | 0,005 |
| 7 | 0.207 | 0.3 | 0.30 | Accuracy | 0.97266 | 0.96122 | 0.97031 | 0.9625 | 0.94901 | 0.963 | 0.963 | 0.009 |
| | | | | Precision | 0.96562 | 0.97312 | 0.96542 | 0.9687 | 0.96077 | 0.966 | 0.967 | 0.005 |
| | | | | Recall | 0.96632 | 0.96209 | 0.97067 | 0.96372 | 0.95837 | 0.964 | 0.964 | 0.005 |
| | | | | $F_1$-Score | 0.96585 | 0.96711 | 0.96777 | 0.96591 | 0.95893 | 0.966 | 0.965 | 0.004 |

## D.3.6  P19 Runs

Table 28: Run documentation of the ablation study on the three ablations of STaRFormer on the P19 dataset [66]; (i) Base, (ii) STaRFormer-RM and (iii) STaRFormer. Here, only the architecture of the model is changed, keeping everything else fixed.

| Ablation | Metrics | 0 | 1 | 2 | 3 | 4 | Median | Average | Std. Dev. |
|---|---|---|---|---|---|---|---|---|---|
| Base | AUROC | 0.9095 | 0.88573 | 0.88623 | 0.8879 | 0.86183 | 0.886 | 0.886 | 0.017 |
| | AUPRC | 0.63754 | 0.57881 | 0.60019 | 0.58167 | 0.55528 | 0.582 | 0.591 | 0.031 |
| | Accuracy | 0.9714 | 0.9665 | 0.97114 | 0.97011 | 0.96856 | 0.970 | 0.970 | 0.002 |
| | $F_1$-Score | 0.80053 | 0.77317 | 0.78357 | 0.76393 | 0.75725 | 0.773 | 0.776 | 0.017 |
| | Recall | 0.87247 | 0.84609 | 0.89798 | 0.85501 | 0.81718 | 0.855 | 0.858 | 0.030 |
| | Precision | 0.75358 | 0.72758 | 0.72314 | 0.71227 | 0.71775 | 0.723 | 0.727 | 0.016 |
| STaRFormer with RM | AUROC | 0.921 | 0.87785 | 0.88049 | 0.88521 | 0.86792 | 0.880 | 0.886 | 0.020 |
| | AUPRC | 0.67494 | 0.57197 | 0.57515 | 0.58078 | 0.5798 | 0.580 | 0.597 | 0.044 |
| | Accuracy | 0.9714 | 0.96702 | 0.96882 | 0.97088 | 0.75293 | 0.969 | 0.926 | 0.097 |
| | $F_1$-Score | 0.81059 | 0.76449 | 0.74701 | 0.76374 | 0.97063 | 0.764 | 0.811 | 0.092 |
| | Recall | 0.85729 | 0.86743 | 0.91123 | 0.8728 | 0.86125 | 0.867 | 0.874 | 0.022 |
| | Precision | 0.77563 | 0.70946 | 0.6806 | 0.70699 | 0.69718 | 0.707 | 0.714 | 0.036 |
| STaRFormer with DAReM | AUROC | 0.91218 | 0.89776 | 0.89279 | 0.89397 | 0.87469 | 0.894 | 0.894 | 0.013 |
| | AUPRC | 0.66579 | 0.60643 | 0.61391 | 0.60891 | 0.57196 | 0.609 | 0.613 | 0.034 |
| | Accuracy | 0.97346 | 0.96212 | 0.96985 | 0.97243 | 0.97114 | 0.971 | 0.970 | 0.005 |
| | $F_1$-Score | 0.81241 | 0.77328 | 0.77309 | 0.78401 | 0.77452 | 0.775 | 0.783 | 0.017 |
| | Precision | 0.89539 | 0.79238 | 0.886 | 0.85432 | 0.87199 | 0.872 | 0.860 | 0.041 |
| | Recall | 0.76017 | 0.75682 | 0.7142 | 0.73374 | 0.71976 | 0.734 | 0.737 | 0.021 |

## D.3.7 P12 Runs

Table 29: Run documentation of the ablation study on the three ablations of STaRFormer on the P12 dataset [67]; (i) Base, (ii) STaRFormer-RM and (iii) STaRFormer. Here, only the architecture of the model is changed, keeping everything else fixed.

| Ablation | Metrics | 0 | 1 | 2 | 3 | 4 | Median | Average | Std. Dev. |
|---|---|---|---|---|---|---|---|---|---|
| Base | AUROC | 0.85655 | 0.85965 | 0.7408 | 0.85859 | 0.83593 | 0.85655 | 0.830 | 0.051 |
| | AUPRC | 0.51476 | 0.55478 | 0.33127 | 0.54069 | 0.52082 | 0.52082 | 0.492 | 0.092 |
| | Accuracy | 0.60384 | 0.86822 | 0.86322 | 0.82402 | 0.84487 | 0.84487 | 0.801 | 0.111 |
| | $F_1$-Score | 0.54218 | 0.57294 | 0.55269 | 0.7109 | 0.60415 | 0.57294 | 0.597 | 0.068 |
| | Recall | 0.60725 | 0.79105 | 0.70883 | 0.68606 | 0.74284 | 0.70883 | 0.707 | 0.068 |
| | Precision | 0.73951 | 0.55995 | 0.54662 | 0.76901 | 0.585 | 0.585 | 0.640 | 0.106 |
| STaRFormer with RM | AUROC | 0.85727 | 0.86518 | 0.83434 | 0.86184 | 0.82935 | 0.857 | 0.850 | 0.017 |
| | AUPRC | 0.51466 | 0.5758 | 0.47908 | 0.53086 | 0.49449 | 0.515 | 0.519 | 0.037 |
| | Accuracy | 0.8849 | 0.8799 | 0.86072 | 0.8824 | 0.77815 | 0.880 | 0.857 | 0.045 |
| | $F_1$-Score | 0.61662 | 0.63967 | 0.86072 | 0.8824 | 0.68701 | 0.687 | 0.737 | 0.125 |
| | Recall | 0.79122 | 0.82993 | 0.70894 | 0.77173 | 0.66954 | 0.772 | 0.754 | 0.064 |
| | Precision | 0.58794 | 0.60576 | 0.70578 | 0.69176 | 0.75015 | 0.692 | 0.668 | 0.069 |
| STaRFormer with DAReM | AUROC | 0.85853 | 0.85989 | 0.84508 | 0.86469 | 0.83435 | 0.859 | 0.853 | 0.012 |
| | AUPRC | 0.51325 | 0.53224 | 0.51175 | 0.5426 | 0.50128 | 0.513 | 0.520 | 0.017 |
| | Accuracy | 0.88657 | 0.87239 | 0.86986 | 0.88407 | 0.84237 | 0.872 | 0.871 | 0.018 |
| | $F_1$-Score | 0.67798 | 0.61871 | 0.65169 | 0.70709 | 0.63258 | 0.652 | 0.658 | 0.035 |
| | Recall | 0.64426 | 0.59163 | 0.62328 | 0.67095 | 0.60915 | 0.623 | 0.628 | 0.031 |
| | Precision | 0.75933 | 0.78038 | 0.72938 | 0.78793 | 0.71802 | 0.759 | 0.755 | 0.031 |

## D.3.8 UEA Benchmark

Table 30: Complete results of the UEA benchmark for 30 multivariate time series datasets [70].

| Dataset | ViTST† | DTWD* | Weasel-Muse* | TST (TimesURL)+ | T-Loss+ | TS-TCC+ | TNC+ | TS2Vec+ | InfoTS++ | Rocket* | Mini-Rocket* | TST (TARNet)* | InfoTS$_a$++ | TimesURL+ | TARNet* | STaR-Former |
|---|---|---|---|---|---|---|---|---|---|---|---|---|---|---|---|---|
| AWR | - | 0.987 | 0.990 | 0.977 | 0.943 | 0.953 | 0.973 | 0.987 | 0.987 | **0.993** | **0.993** | 0.947 | **0.993** | 0.990 | 0.977 | **0.993** |
| AF | - | 0.220 | 0.333 | 0.067 | 0.133 | 0.267 | 0.133 | 0.200 | 0.200 | 0.067 | 0.133 | 0.533 | 0.267 | 0.400 | **1.000** | 0.667 |
| BM | - | 0.975 | **1.000** | 0.975 | **1.000** | **1.000** | 0.975 | 0.975 | 0.975 | **1.000** | **1.000** | 0.925 | **1.000** | **1.000** | **1.000** | **1.000** |
| CT | - | 0.989 | 0.990 | 0.975 | 0.993 | 0.985 | 0.967 | **0.995** | 0.974 | 0.991 | 0.990 | 0.971 | 0.987 | 0.990 | 0.994 | 0.994 |
| CK | - | 0.100 | **1.000** | **1.000** | 0.973 | 0.917 | 0.958 | 0.972 | 0.986 | **1.000** | 0.986 | 0.847 | **1.000** | **1.000** | **1.000** | **1.000** |
| DDK | - | 0.600 | 0.575 | 0.622 | 0.650 | 0.380 | 0.460 | 0.680 | 0.549 | 0.500 | 0.750 | 0.300 | 0.600 | 0.720 | 0.750 | **0.760** |
| EW | 0.878 | 0.618 | **0.890** | 0.748 | 0.840 | 0.779 | 0.840 | 0.847 | 0.733 | 0.650 | 0.790 | 0.720 | 0.748 | 0.870 | 0.420 | 0.850 |
| EP | - | 0.618 | **1.000** | 0.949 | 0.971 | 0.957 | 0.957 | 0.964 | 0.971 | 0.986 | **1.000** | 0.775 | 0.993 | 0.978 | **1.000** | 0.986 |
| ER | - | 0.133 | 0.122 | 0.874 | 0.133 | 0.904 | 0.852 | 0.874 | 0.949 | **0.989** | 0.974 | 0.930 | 0.953 | 0.985 | 0.919 | 0.959 |
| EC | **0.456** | 0.323 | 0.430 | 0.262 | 0.205 | 0.285 | 0.297 | 0.308 | 0.281 | 0.450 | 0.430 | 0.337 | 0.323 | 0.304 | 0.323 | 0.393 |
| FD | 0.632 | 0.529 | 0.545 | 0.534 | 0.513 | 0.544 | 0.536 | 0.501 | 0.543 | 0.638 | 0.612 | 0.625 | 0.525 | 0.608 | 0.641 | **0.697** |
| FM | - | 0.530 | 0.490 | 0.560 | 0.580 | 0.460 | 0.470 | 0.480 | 0.630 | 0.520 | 0.550 | 0.590 | 0.620 | **0.660** | 0.620 | 0.640 |
| HMD | - | 0.231 | 0.365 | 0.243 | 0.351 | 0.243 | 0.324 | 0.338 | 0.392 | 0.486 | 0.392 | **0.675** | 0.514 | 0.432 | 0.392 | 0.635 |
| HW | 0.766 | 0.286 | **0.605** | 0.225 | 0.451 | 0.498 | 0.249 | 0.515 | 0.452 | 0.596 | 0.520 | 0.359 | 0.554 | 0.462 | 0.281 | 0.373 |
| HB | 0.766 | 0.717 | 0.727 | 0.746 | 0.741 | 0.751 | 0.746 | 0.683 | 0.722 | 0.741 | 0.771 | **0.782** | 0.771 | 0.746 | 0.780 | 0.772 |
| IW | - | - | - | 0.105 | 0.156 | 0.264 | 0.469 | 0.466 | 0.470 | 0.179 | 0.229 | **0.687** | 0.472 | 0.473 | 0.137 | 0.681 |
| JV | 0.946 | 0.949 | 0.973 | 0.978 | 0.989 | 0.930 | 0.978 | 0.984 | 0.984 | 0.978 | 0.986 | 0.822 | **0.995** | 0.986 | 0.992 | 0.990 |
| LI | - | 0.870 | 0.878 | 0.656 | 0.883 | 0.822 | 0.817 | 0.867 | 0.883 | 0.906 | 0.822 | 0.861 | 0.889 | 0.922 | **1.000** | 0.894 |
| LSST | - | 0.551 | 0.590 | 0.408 | 0.509 | 0.474 | 0.595 | 0.537 | 0.591 | 0.635 | 0.653 | 0.576 | 0.593 | 0.602 | **0.976** | 0.679 |
| MI | - | 0.500 | 0.500 | 0.500 | 0.580 | 0.610 | 0.500 | 0.510 | 0.630 | 0.460 | 0.610 | 0.610 | 0.610 | **0.680** | 0.630 | 0.670 |
| NT | - | 0.883 | 0.870 | 0.850 | 0.917 | 0.822 | 0.911 | 0.928 | 0.933 | 0.872 | 0.933 | 0.939 | 0.939 | 0.961 | 0.911 | **0.989** |
| PS | 0.913 | 0.711 | - | 0.740 | 0.676 | 0.734 | 0.699 | 0.682 | 0.751 | 0.832 | 0.809 | 0.930 | 0.757 | 0.821 | 0.936 | **0.943** |
| PD | - | 0.977 | 0.948 | 0.560 | 0.981 | 0.974 | 0.979 | 0.989 | **0.990** | 0.981 | 0.967 | 0.981 | 0.989 | 0.989 | 0.976 | 0.983 |
| PSp | - | 0.151 | 0.190 | 0.085 | 0.220 | 0.252 | 0.207 | 0.233 | 0.249 | 0.273 | **0.291** | 0.111 | 0.233 | 0.237 | 0.165 | 0.178 |
| RS | - | 0.803 | 0.934 | 0.809 | 0.855 | 0.816 | 0.776 | 0.855 | 0.855 | 0.901 | 0.868 | 0.796 | 0.829 | 0.862 | **0.987** | 0.947 |
| SCP1 | 0.898 | 0.775 | 0.710 | 0.754 | 0.843 | 0.823 | 0.799 | 0.812 | 0.874 | 0.867 | 0.915 | **0.961** | 0.887 | 0.908 | 0.816 | 0.913 |
| SCP2 | 0.561 | 0.539 | 0.460 | 0.550 | 0.539 | 0.533 | 0.550 | 0.578 | 0.578 | 0.555 | 0.506 | 0.604 | 0.572 | 0.600 | 0.622 | **0.635** |
| SAD | 0.985 | 0.963 | 0.982 | 0.923 | 0.905 | 0.970 | 0.934 | 0.988 | 0.947 | 0.997 | 0.963 | **0.998** | 0.932 | 0.985 | 0.985 | 0.989 |
| SWJ | - | 0.200 | 0.333 | 0.267 | 0.333 | 0.330 | 0.400 | 0.467 | 0.467 | 0.467 | 0.330 | 0.600 | 0.467 | 0.467 | 0.533 | **0.733** |
| UW | 0.862 | 0.903 | 0.916 | 0.575 | 0.875 | 0.753 | 0.759 | 0.906 | 0.884 | **0.931** | 0.785 | 0.913 | 0.884 | 0.919 | 0.878 | 0.894 |
| Avg. Accuracy | 0.790 | 0.608 | 0.691 | 0.617 | 0.658 | 0.668 | 0.670 | 0.704 | 0.714 | 0.715 | 0.719 | 0.729 | 0.730 | 0.752 | 0.755 | **0.795** |
| Rank | - | - | - | 13 | 12 | 11 | 10 | 9 | 8 | 7 | 6 | 5 | 4 | 3 | 2 | **1** |
| Avg. Rank | - | - | - | 10.6 | 8.6 | 9.2 | 9.9 | 7.4 | 6.8 | 5.5 | 5.7 | 6.5 | 5.3 | 3.9 | 4.9 | **2.8** |
| Top Scores | 1 | 0 | 5 | 1 | 1 | 1 | 0 | 1 | 1 | 5 | 4 | 6 | 3 | 4 | 7 | **9** |
| 1-v-1 | 8 | 28 | 20 | 29 | 27 | 27 | 29 | 25 | 27 | 19 | 22 | 23 | 23 | 19 | 21 | - |
| DS Count | 10 | 29 | 28 | 30 | 30 | 30 | 30 | 30 | 30 | 30 | 30 | 30 | 30 | 30 | 30 | 30 |
| Accuracy 28 | - | 0.604 | 0.691 | 0.631 | 0.675 | 0.680 | 0.677 | 0.713 | 0.722 | 0.730 | 0.733 | 0.724 | 0.738 | 0.760 | 0.770 | **0.793** |
| Rank 28 | - | 15 | 10 | 14 | 13 | 11 | 12 | 9 | 8 | 6 | 5 | 7 | 4 | 3 | 2 | **1** |
| Avg. Rank 28 | - | 11.2 | 7.8 | 11.7 | 9.1 | 10.3 | 11.0 | 8.1 | 7.5 | 5.8 | 6.0 | 7.5 | 5.8 | 4.1 | 5.2 | **3.1** |
| Accuracy 9 | 0.776 | 0.702 | 0.737 | 0.674 | 0.717 | 0.708 | 0.715 | 0.734 | 0.727 | 0.756 | 0.751 | 0.771 | 0.736 | 0.770 | 0.717 | **0.793** |
| Rank 9 | 2 | 15 | 7 | 16 | 12 | 14 | 13 | 9 | 10 | 5 | 6 | 3 | 8 | 4 | 11 | **1** |
| Avg. Rank 9 | 6.4 | 11.8 | 9.0 | 12.3 | 11.1 | 11.3 | 10.8 | 9.0 | 10.0 | 6.7 | 7.4 | 3.9 | 8.4 | 5.3 | 6.3 | **3.3** |

The model results marked with * are taken from the [21], + from [27], ++ from [26] and † from [3].

### D.3.9 STaRFormer Approach Ablation Runs

Table 31: STaRFormer architecture ablation study results (Accuracy).

|  | Base | STaRFormer-RM | STaRFormer |
|---|---|---|---|
| DKT | $0.849 \pm 0.002$ | $0.845 \pm 0.001$ | $\mathbf{0.852} \pm 0.003$ |
| GL | $0.881 \pm 0.012$ | $0.894 \pm 0.014$ | $\mathbf{0.904} \pm 0.015$ |
| P19 | $\mathbf{0.970} \pm 0.002$ | $\mathbf{0.970} \pm 0.002$ | $\mathbf{0.970} \pm 0.005$ |
| P12 | $0.801 \pm 0.111$ | $0.857 \pm 0.045$ | $\mathbf{0.871} \pm 0.018$ |
| PAM | $0.964 \pm 0.013$ | $0.964 \pm 0.011$ | $\mathbf{0.976} \pm 0.009$ |
| EW | 0.752 | 0.799 | **0.850** |
| EC | 0.371 | **0.402** | 0.393 |
| FD | 0.687 | 0.673 | **0.697** |
| HW | 0.336 | 0.327 | **0.373** |
| HB | **0.786** | 0.772 | 0.772 |
| JV | **0.990** | 0.982 | 0.990 |
| PD | 0.982 | 0.980 | **0.983** |
| PS | 0.909 | 0.922 | **0.943** |
| SCP1 | 0.906 | 0.891 | **0.913** |
| SCP2 | 0.630 | 0.620 | **0.635** |
| SAD | **0.990** | 0.983 | 0.989 |
| UW | 0.881 | 0.838 | **0.894** |
| Yahoo | 0.988 | 0.991 | **0.992** |
| KPI | **0.982** | 0.980 | 0.981 |
| Avg. Acc. | 0.824 | 0.826 | **0.841** |
| Rank | 3 | 2 | **1** |
| Avg Rank | 2.1 | 2.5 | **1.2** |
| Top Scores | 5 | 2 | **15** |
| **1-v-1** |  |  |  |
| Base | - | 12 | 3 |
| RM | 6 | - | 1 |
| STaRFormer | 14 | 16 | - |

In Table 31, we display the complete metric scores summarized in Table 6. To ensure consistence, we report the accuracy for every dataset, as it is available for each dataset. However, this is not the ideal metric for many datasets. Heavily skewed datasets like P19 and P12 or anomaly detection datasets, where anomalous elements appear much less frequently than regular elements, $F_1$ would be a better score to consider.

### D.3.10 Univariate Anomaly Detection Benchmarks

Table 32: Run documentation of the ablation study on the three ablations of STaRFormer on the univariate anomaly detection benchmark datasets; (i) Base, (ii) STaRFormer-RM and (iii) STaRFormer. Here, only the architecture of the model is changed, keeping everything else fixed.

| Method | Yahoo | | | | KPI | | | |
|---|---|---|---|---|---|---|---|---|
|  | $F_1$ | Precision | Recall | Accuracy | $F_1$ | Precision | Recall | Accuracy |
| Base | 0.685 | 0.671 | 0.767 | 0.988 | 0.814 | 0.857 | 0.780 | **0.982** |
| STaRFormer-RM | 0.737 | **0.801** | 0.696 | 0.991 | 0.737 | **0.910** | 0.670 | 0.980 |
| STaRFormer | **0.789** | 0.772 | **0.807** | **0.992** | **0.830** | 0.852 | **0.811** | 0.981 |

### D.3.11 TSR Benchmark

Table 33: Complete results of the TSR Benchmark for 19 time series datasets.

| Dataset | FPCR* | FPCR-Bspline* | SVR* | SVR Optimised* | Random Forest* | XG-Boost* | 1-NN-ED* | 5-NN-ED* | 1-NN-DTWD* | 5-NN-DTWD* | Rocket* | FCN* | ResNet* | Inception* | TAR-Net | STaR-Former |
|---|---|---|---|---|---|---|---|---|---|---|---|---|---|---|---|---|
| AE | 5.405 | 5.405 | 3.458 | 3.455 | 3.455 | 3.489 | 5.232 | 4.227 | 6.037 | 4.020 | 2.299 | 2.866 | 3.065 | 4.435 | 3.161 | **1.844** |
| AR | 8.436 | 8.436 | 8.651 | 8.651 | 8.390 | 8.493 | 30.254 | 10.233 | 12.002 | 11.951 | 8.124 | 8.426 | 8.179 | 8.841 | 8.390 | **4.719** |
| BPM10 | 99.726 | 99.732 | 110.574 | 110.574 | 94.072 | **93.138** | 139.230 | 115.669 | 139.135 | 115.503 | 120.058 | 94.349 | 95.489 | 96.750 | 116.871 | 113.421 |
| BPM25 | 69.379 | 69.370 | 75.734 | 71.437 | 63.301 | **59.496** | 88.194 | 74.156 | 88.256 | 72.718 | 62.770 | 59.727 | 64.463 | 62.228 | 85.271 | 84.004 |
| BC | 11.088 | 11.095 | 4.791 | 4.791 | 0.856 | **0.638** | 6.536 | 5.845 | 4.984 | 4.868 | 3.361 | 4.988 | 4.061 | 1.585 | 4.073 | 2.913 |
| BIDMCHR | 13.981 | 13.981 | 13.580 | 13.393 | 15.016 | 13.964 | 14.837 | 14.756 | 15.291 | 15.127 | 13.944 | 13.131 | 10.741 | 9.425 | 14.072 | **8.068** |
| BIDMCRR | 3.365 | 3.365 | 4.160 | 3.174 | 4.350 | 4.368 | 4.387 | 4.135 | 3.529 | 3.432 | 4.093 | 3.578 | 3.921 | 3.018 | 3.487 | **2.973** |
| BIDMCSPO2 | 4.954 | 4.954 | 4.819 | 4.797 | 4.570 | 4.451 | 5.530 | 5.408 | 5.215 | 5.124 | 5.222 | 5.968 | 5.988 | 5.576 | 5.231 | **4.130** |
| C3M | 0.045 | 0.045 | 0.066 | 0.066 | 0.042 | 0.045 | 0.053 | 0.042 | 0.053 | 0.043 | 0.044 | 0.074 | 0.095 | 0.054 | 0.060 | **0.037** |
| FM1 | 0.019 | 0.019 | 0.078 | 0.046 | 0.016 | 0.016 | 0.015 | 0.016 | 0.012 | 0.010 | **0.002** | 0.007 | 0.009 | 0.017 | 0.017 | 0.013 |
| FM2 | 0.019 | 0.019 | 0.076 | 0.076 | 0.014 | 0.018 | 0.019 | 0.019 | 0.016 | 0.016 | **0.006** | 0.007 | 0.014 | 0.007 | 0.048 | **0.006** |
| FM3 | 0.021 | 0.021 | 0.035 | 0.035 | 0.020 | 0.021 | 0.020 | 0.021 | 0.014 | 0.013 | **0.004** | 0.008 | 0.016 | 0.008 | 0.048 | 0.017 |
| HPC1 | 147.549 | 147.549 | 519.156 | 152.391 | 248.859 | 231.090 | 473.933 | 432.595 | 427.043 | 297.222 | **132.799** | 162.244 | 193.207 | 153.716 | 519.454 | 147.250 |
| HPC2 | 46.925 | 46.930 | 57.340 | 55.981 | 46.932 | 44.373 | 71.479 | 64.273 | 58.803 | 51.495 | **32.607** | 46.829 | 39.080 | 39.410 | 50.917 | 42.102 |
| IEEEPPG | 31.381 | 31.381 | 36.302 | 37.254 | 32.109 | 31.488 | 33.209 | 27.111 | 37.140 | 33.573 | 36.515 | 34.326 | 33.151 | **23.904** | 31.245 | 30.012 |
| LFMC | 37.684 | 37.688 | 43.022 | 39.734 | 32.163 | 32.442 | 47.837 | 38.536 | 39.972 | 35.185 | 29.410 | 33.257 | 30.352 | **28.796** | 41.905 | 31.628 |
| NHS | **0.142** | **0.142** | 0.143 | 0.143 | 0.148 | **0.142** | 0.203 | 0.157 | 0.198 | 0.156 | **0.142** | 0.148 | 0.150 | 0.150 | 0.144 | **0.142** |
| NTS | 0.138 | 0.138 | 0.139 | 0.139 | 0.143 | 0.138 | 0.193 | 0.151 | 0.187 | 0.151 | **0.138** | **0.138** | **0.138** | 0.159 | 0.140 | **0.138** |
| PPG | 20.674 | 20.674 | 19.005 | 19.005 | 17.531 | 16.583 | 21.877 | 18.282 | 26.025 | 20.768 | 14.051 | 13.039 | 11.382 | **9.924** | 20.703 | 12.794 |
| Avg. Rel. Mean Difference ↓ | 0.028 | 0.029 | 0.387 | 0.208 | -0.121 | -0.132 | 0.288 | 0.051 | 0.125 | -0.034 | -0.245 | -0.160 | -0.119 | -0.220 | 0.170 | **-0.254** |
| Avg. Rel. Mean Difference Rank ↓ | 9 | 10 | 16 | 14 | 6 | 5 | 15 | 11 | 12 | 8 | 2 | 4 | 7 | 3 | 13 | **1** |
| Number of Top Scores ↑ | 1 | 1 | 0 | 0 | 0 | 4 | 0 | 0 | 0 | 0 | 7 | 1 | 1 | 3 | 0 | **9** |

The model results marked with * are taken from the official benchmark (http://tseregression.org/).

