# OpenReview forum: "STaRFormer: Semi-Supervised Task-Informed Representation Learning via Dynamic Attention-Based Regional Masking for Sequential Data"
_NeurIPS.cc/2025/Conference — NeurIPS 2025 poster_

### Official Review · Reviewer_BjfK · 2025-06-30

**Clarity:** 2
**Significance:** 3
**Originality:** 3
**Rating:** 4
**Confidence:** 4

**Summary:**

The paper proposes STaRFormer, a unified framework for time series modeling that addresses challenges such as non-stationarity and irregular sampling. It introduces a novel Dynamic Attention-based Regional Masking (DAReM) module that adaptively masks task-relevant regions based on attention scores. The framework employs a semi-supervised contrastive learning approach, combining self-supervised batch-wise contrastive loss with supervised class-wise loss to improve representation learning for downstream tasks. STaRFormer is evaluated on multiple benchmark datasets across different tasks, demonstrating state-of-the-art performance.

**Questions:**

As weakness

**Ethical Concerns:**

["NO or VERY MINOR ethics concerns only"]

**Final Justification:**

After reviewing the rebuttal and additional clarifications, I acknowledge that several of my earlier concerns, such as the readability of certain sections and the clarification of the task-specific design, were addressed.  I want to note that my evaluation is not intended to penalize the authors for being transparent about limitations, but on weighing the trade-offs between robustness/performance and computational efficiency/flexibility. While the efficiency limitation is acknowledged in the paper, the runtime and memory usage still indicate substantial overhead relative to lighter baselines for marginal accuracy improvements in certain settings. For these reasons, I believe my current rating reflects both the contributions of the work and the remaining concerns.

**Limitations:**

yes

**Quality:**

3

**Strengths And Weaknesses:**

Strengths:
1. The framework composes formulations for both sequential-level and element-level supervision, enabling adaptability to diverse downstream tasks such as classification and anomaly detection.

 2. The Dynamic Attention-based Regional Masking (DAReM) module introduces a novel masking strategy that leverages attention scores to identify and focus on task-relevant regions.

3. The paper presents extensive experiments and ablation studies, and the results show improvements compared to previous methods.

Weaknesses:
1.  It is unclear whether the trained STaRFormer model can generalize across multiple downstream tasks or if it must be trained separately from scratch for each task, limiting its flexibility and generability compared to task-agnostic frameworks.

2. The proposed DAReM masking method is not compared against other baseline masking strategies (e.g., random, uniform, or fixed-region masking), making it difficult to isolate and verify its effectiveness.

3. Figure 1(b) is not well explained in the main paper and is difficult to interpret without first reading Supplementary Section B.3.

 4. Important figures such as Fig. 8 (which is referred to in the main text) are only included in the supplementary material, reducing accessibility to explanations within the main paper.

---

> ### Author Rebuttal · Authors · 2025-07-29
>
> We express our gratitude to the reviewers for their careful reading and critical assessment of our work. We are greatly encouraged by the overall positive reception of our work, reflected in the reviewers' consensus across most reviewing criteria. We are glad to hear they found our approach of addressing irregularly sampled and non-stationary time series 'innovative' (nw2T), 'novel' (KH4k, 8jfK) and 'effective' (nw2T, 7K1F). Moreover, we are pleased that the reviewers acknowledged our efforts to evaluate our approach through 'comprehensive' (nw2T), 'thorough' (nw2T), and 'extensive' (7K1F, 8jfK) experiments and ablation studies. We are satisfied that these experiments demonstrate the effectiveness (nw2T, 7K1F) and strong performance (KH4k) of our approach compared to state-of-the-art baseline models. We will address your comments below and incorporate your feedback accordingly.
>
> **Q11)** _*It is unclear whether the trained STaRFormer model can generalize across multiple downstream tasks or if it must be trained separately from scratch for each task, limiting its flexibility and generability compared to task-agnostic frameworks.*_
>
> You raise a valid point that highlights a limitation inherent to any non-task-agnostic model. Thus, this limitation applies to all non-task-agnostic models and is not exclusive to STaRFormer. However, we do not believe this is unclear in the paper, as we explicitly state in lines 81-82 on page 2 and lines 117-119 on page 3 that we propose a task-coupled approach, thus implying that STaRFormer cannot be task-agnostic. Moreover, the flexibility within this context is constrained, as, to the best of our knowledge, all existing task-agnostic models have been evaluated on tasks for which they were previously trained to extract universal representations. Therefore, from the perspective of this study, these models merely decouple the training process between representation and task, which can actually introduce an additional layer of complexity, as it requires two distinct training pipelines. For instance, in the case of TimesURL, optimizing a Support Vector Classifier (SVC) for large datasets, such as DKT, can be quite challenging and time-consuming, as scikit-learn does not natively support GPUs. This is evident in Table 14 (Appendix Section D.3.1) for the DKT dataset, where the runtimes of TARNet, TimesURL, and STaRFormer can be compared. The runtime of TimesURL is significantly longer. Furthermore, it should be noted that task-agnostic models may experience reduced performance and effectiveness as a trade-off for their increased flexibility. Within this scope, the extent of performance degradation is unclear, and we are currently unable to validate or refute the degree of degradation due to the lack of supporting evidence.
>
> **Q12)** _*The proposed DAReM masking method is not compared against other baseline masking strategies (e.g., random, uniform, or fixed-region masking), making it difficult to isolate and verify its effectiveness.*_
>
> We would like to point out that, indeed, we have compared the masking scheme against random masking (RM) and no masking across all datasets used in our study. These results are summarized in section 4.3.1 and in Table 5, while extensive results can be found in Table 27. Considering the other three ablation studies, we deemed two variations sufficient in this ablation to convince the reader; however, upon your recommendation, we could add additional variations for the camera-ready paper.
>
> **Q13)** _*Figure 1(b) is not well explained in the main paper and is difficult to interpret without first reading Supplementary Section B.3.*_
>
> Thank you for providing this perspective. As mentioned in the response to **Q4)**, we had to transfer some contents to the appendix. This necessitates finding the appropriate balance between what to retain in the main body and what to relocate to the supplementary information. At the time of submission, we believed the explanation provided was sufficient for comprehension, and for readers with a particular interest in the masking procedure, we directed them to the relevant section in the appendix. Considering your suggestion, we could elaborate on this section in greater detail with additions from the supplementary material for the camera-ready version.
>
> **Q14)** _*Important figures such as Fig. 8 (which is referred to in the main text) are only included in the supplementary material, reducing accessibility to explanations within the main paper.*_
>
> From our perspective, Fig. 8 in particular, is a generic schematic that can also be found in other CL-related publications, such as the original SimCL (NT-Xent) paper [1]. Hence, although we originally had it included in the main PDF, we didn't deem it important enough to relocate any other content to the appendix. That is why we chose to link it to the main PDF so that it would be easily accessible to readers seeking additional information. Upon your recommendation, we would be happy to include Fig. 8 in the main PDF.
>
> If our responses have adequately addressed your questions and concerns, we would appreciate your consideration in updating your recommendation score. Otherwise, we welcome further discussion.
>
> [1] Ting Chen, Simon Kornblith, Mohammad Norouzi, and Geoffrey Hinton. A Simple Framework for Contrastive Learning of Visual Representations, June 2020. arXiv:2002.05709 [cs, stat].

---

> > ### Comment · Reviewer_BjfK · 2025-08-05
> >
> > Thank you for your detailed and thoughtful responses. I appreciate the effort made to clarify the scope of STaRFormer, particularly around task coupling, the DAReM module, and the supplementary figures. Your clarifications addressed many of my concerns.
> >
> > However, I remain slightly concerned about the computational efficiency and generalizability of the proposed method compared to task-agnostic alternatives. While your response (Q11) justifies the task-coupled design and highlights the training overhead of task-agnostic models such as TimesURL, I would argue that the runtime data in Table 14 still shows a substantial training and memory burden for STaRFormer itself, especially on the DKT dataset. For instance, STaRFormer incurs notably higher training time and high memory usage when compared to lighter models like Transformer (TST), which achieves only slightly lower accuracy (Table 1). Even though I acknowledge that STaRFormer demonstrates greater robustness under noise, this benefit seems to come at a significant computational cost.
> >
> > Given these considerations, I would like to maintain my current rating. This is a technically solid work with extensive experiments, but limitations remain in terms of efficiency and flexibility.

---

> > > ### Author Response · Authors · 2025-08-05
> > >
> > > Dear Reviewer,
> > >
> > > We value the time and effort you have invested in reviewing our paper, as well as your thoughtful responses, acknowledgment of our technically solid work and your generally positive feedback.
> > >
> > > We are pleased to hear we were able to address your initial concerns in the rebuttal and that you recognize the efforts we made to clarify any concerns you raised. However, you might also understand we are disappointed to hear that you do not consider updating your score, despite our attempts to address your feedback successfully. In the limitations section of the paper (page 9, line 342), we acknowledge that our approach is not as computationally efficient as other methodologies might be. The NeurIPS guideline advises authors to be transparent about the limitations of their work and not to be penalized for doing so. We have strived to be upfront about the trade-offs between the robustness of STaRFormer and its computational cost, particularly in comparison to more lightweight models like the Transformer (TST). The Transformer implementation used for comparison in the DKT dataset employs flash attention, which is the contributing factor to the significantly accelerated computation time compared to the regular attention approach used in STaRFormer. Additionally, our approach (DAReM + CL) necessitates two iterative steps during the forward pass, which contributes to the increased computational expenses. However, as outlined in the limitations, this procedure is only employed during training, ensuring that inference remains unaffected by this slowdown. Nevertheless, we emphasize that future work should focus on implementing flash attention to enhance efficiency and scalability for large-scale datasets (page 9, line 345). Additionally, regarding the increased flexibility of task-agnostic models, we are happy to expand on this point more explicitly in the limitations section of the paper. Although we initially deemed it implicit due to our task-bound model definition, we recognize that this is an important consideration and should be explicitly mentioned nevertheless. However, we would like to point out that we deem it a bit harsh to solely criticize this work based on this modeling choice, given that there are also many downsides to this added flexibility; the most obvious one being model performance. While the added flexibility is a nice addition, task-agnostic models can lack in performance compared to task-specific models. For example, on the DKT dataset and the GL dataset (Table 1), the model performance (in accuracy) of the best task-agnostic model from the UEA benchmark (TimesURL) is considerably worse than STaRFormer's. With that in mind, when developing a model for a specific task, such as an industrial application, would the theoretically increased flexibility genuinely enhance the solution approach? We would argue that the suitability of a model largely depends on the intended application. If the model is designed to be utilized across multiple tasks, we believe we would agree that a task-agnostic model would be the preferable choice. However, for a specific task, particularly within an industrial context for example, performance considerations on a specific task hold greater significance in our view.
> > >
> > > While we understand your perspective and the issues you have raised, considering that computational efficiency is addressed in the limitations, this leaves the lack of flexibility in the modeling architecture as the primary limiting factor in your evaluation of our work. We hope that these additional clarifications and considerations presented will encourage you to reconsider your current stance after further reflection.

---

> > > > ### Comment · Reviewer_BjfK · 2025-08-06
> > > >
> > > > Thank you for your additional clarifications. I appreciate the further explanation regarding the training-time overhead, particularly the impact of flash attention in TST and the two-step forward pass in STaRFormer, as well as the distinction between training and inference costs. I also understand your rationale for the task-coupled design and agree that, in certain contexts, task-specific performance can outweigh cross-task flexibility.
> > > >
> > > > My evaluation is based not on penalizing transparency, but on weighing the practical trade-offs between robustness/performance and computational efficiency/flexibility. While the efficiency limitation is acknowledged in the paper, the runtime and memory results in Table 14 still indicate substantial overhead relative to lighter baselines for marginal accuracy improvements in some settings. Given these factors, I believe my current rating reflects both the strengths and the remaining limitations of the work. I appreciate the thoroughness of your rebuttal and the value of the contributions.

---

> > > > > ### Author Response · Authors · 2025-08-06
> > > > >
> > > > > We sincerely thank the reviewer for the opportunity and the time dedicated to engaging in a constructive discussion.

---

### Official Review · Reviewer_7K1F · 2025-07-02

**Clarity:** 3
**Significance:** 3
**Originality:** 3
**Rating:** 4
**Confidence:** 1

**Summary:**

This paper propose a Transformer-based framework called STaRFormer for sequential data modeling. They state that traditional LSTM and Transformer models do not perform well on non-stationary and irregularly sampled time series. And to tackle these challenges, they introduce a novel semi-supervised, task-informed representation learning technique using a dual-encoder (towers) architecture: (1) In one tower, the model utilize masked token modeling during training for representation learning to improve the robutness of the model in handling potential non-stationary data during application, and (2) in another tower, the model perform supervised downstream task prediction. A contrastive loss is then used maximizes the agreement between the latent representations from both towers, improving the model's robustness and generalization. Experiments demonstrate the effectiveness of the model.

**Questions:**

I have attached my questions in the Weaknesses section.

**Ethical Concerns:**

["NO or VERY MINOR ethics concerns only"]

**Final Justification:**

The authors provided detailed clarifications on both points I raised: (1) why standard Transformers are less suited to irregularly sampled time series without timestamp information, and (2) the rationale behind importance-based masking in the context of contrastive learning. These responses have partially addressed my concerns, though some minor issues remain. In particular, for the second point, while the authors’ contrastive-learning-based justification is clear, the potential trade-off of masking highly informative features for representation quality remains a conceptual concern.

I would also like to note that works proposing new architectures should be encouraged when they provide reasonably solid empirical evaluation. Therefore I will maintain my positive rating.

**Limitations:**

Yes

**Quality:**

3

**Strengths And Weaknesses:**

**Disclaimer:** I am not an expert in this area, but will do my best to provide helpful evaluation:

## Strengths

1. The paper addresses the important and practical challenge of modeling sequential data that is non-stationary and irregularly sampled. This sounds to me a common issue in many real-world applications, such as digital car keys prediction that the authors used to motivate this work, making the problem highly relevant.

2. The proposed model, STaRFormer, is shown to be highly effective through extensive experiments. The improvement is not aways significant, but they are quite consistent across tasks.

3. The claims are well-supported by a comprehensive set of ablation studies.

## Weaknesses

I mainly have some concerns on the motivation on some design choices of the model:

1. The paper states that LSTMs and Transformers "typically assume that the data is fully observed, stationary, and sampled at regular intervals." This might be a reasonable assertion for LSTMs, which inherently process inputs in a sequential order. However, it is not clear to me why this will apply to Transformers as well. For instance, with architectural variants like relative positional embeddings, Transformers can be adapted to handle signals sampled at dynamic intervals. It would be nice to elaborate on why standard Transformers are still considered ill-suited for such data.

2. The proposed method dynamically masks regions that are deemed important based on attention scores, which is different from the random masking used in many self-supervised methods. My understanding is that the purpose of masking, as seen in paradigms like Masked Image Modeling (MIM), is to incentivize representation learning by forcing the model to learn the correpondance between regions. From this perspective, masking the *most important* regions seems counter-productive, as it removes the most informative signals for reconstruction. Could the authors provide more intuition on why this strategy is more effective than random masking?

---

> ### Author Rebuttal · Authors · 2025-07-29
>
> We express our gratitude to the reviewers for their careful reading and critical assessment of our work. We are greatly encouraged by the overall positive reception of our work, reflected in the reviewers' consensus across most reviewing criteria. We are glad to hear they found our approach of addressing irregularly sampled and non-stationary time series 'innovative' (nw2T), 'novel' (KH4k, 8jfK) and 'effective' (nw2T, 7K1F). Moreover, we are pleased that the reviewers acknowledged our efforts to evaluate our approach through 'comprehensive' (nw2T), 'thorough' (nw2T), and 'extensive' (7K1F, 8jfK) experiments and ablation studies. We are satisfied that these experiments demonstrate the effectiveness (nw2T, 7K1F) and strong performance (KH4k) of our approach compared to state-of-the-art baseline models. We will address your comments below and incorporate your feedback accordingly.
>
> **Q9)** _*The paper states that LSTMs and Transformers "typically assume that the data is fully observed, stationary, and sampled at regular intervals." This might be a reasonable assertion for LSTMs, which inherently process inputs in a sequential order. However, it is not clear to me why this will apply to Transformers as well. For instance, with architectural variants like relative positional embeddings, Transformers can be adapted to handle signals sampled at dynamic intervals. It would be nice to elaborate on why standard Transformers are still considered ill-suited for such data.*_
>
> Primarily, this is not feasible for us because, as in many of the underlying datasets, we are only provided with the time series features, not the actual timestamps a priori. This includes the main motivation (DKT) for the work from our industrial sponsor. Hence, it is not feasible to construct exact time intervals to effectively apply temporally contextualized embeddings. Upon your recommendation, we are happy to elaborate on this point in the camera-ready version, if you deem it necessary. For additional information, please refer to the answer to **Q8)**.
>
> **Q10)** _*The proposed method dynamically masks regions that are deemed important based on attention scores, which is different from the random masking used in many self-supervised methods. My understanding is that the purpose of masking, as seen in paradigms like Masked Image Modeling (MIM), is to incentivize representation learning by forcing the model to learn the correspondence between regions. From this perspective, masking the most important regions seems counter-productive, as it removes the most informative signals for reconstruction. Could the authors provide more intuition on why this strategy is more effective than random masking?*_
>
> Thank you for the insightful comment, reflecting the core arguments regarding the dynamic masking method presented in the paper. We believe your understanding to be indeed accurate. The key distinction in our approach lies in the fact that we do not base our model updates on the reconstruction term between the correlated views, as is commonly done in approaches like Masked Language Modeling (MLM) or Masked Image Modeling (MIM). Instead, we employ CL to align the two distinct latent spaces derived from masked and unmasked sequences. By forming positive pairs between batch-wise and class-wise samples, we consider the correspondence not only to the actual sequence but also to other sequences within the same class. In the class-wise case, this implies that there is no direct correspondence in the traditional sense. Our formulation differs from the 'classical' approach by not directly modeling the correspondence between two regions, as seen in methods like MIM, as you mentioned. Instead, we focus on the degree of difference in the latent space created by the masking, with the contrastive formulation aiming to align even disparate embeddings in the latent space. Thus, we aim to create more disparate representations by specifically masking their key important features. Through the creation of more diverse and challenging positive pairings, we strive to enhance and develop a more robust latent space, which the model uses to perform downstream tasks effectively.
>
> If our responses have adequately addressed your questions and concerns, we would appreciate your consideration in updating your recommendation score. Otherwise, we welcome further discussion.

---

> > ### Author Response · Authors · 2025-08-05
> >
> > Dear Reviewer,
> >
> > Thank you for your insightful and detailed review of our work. Even though you mentioned not being an expert in this area, you have provided conceptually intriguing insights and thorough questions/weaknesses. We appreciate your overall positive review of our work in general.
> >
> > In our rebuttal, we have addressed the two weaknesses you raised in your review. If our responses have satisfactorily addressed all your concerns, we would be grateful if you could take this into account in your final evaluation. Should there be any areas where our responses have not been sufficient, we welcome any opportunities for further clarification.
> >
> > Thank you once again for your valuable input.

---

### Official Review · Reviewer_KH4k · 2025-07-02

**Clarity:** 2
**Significance:** 2
**Originality:** 3
**Rating:** 3
**Confidence:** 3

**Summary:**

The paper proposes a time-series classification model combining multiple task objectives during training to learn better representations that support better classification performance. Among these multiple tasks are: (1) the classification loss, (2) the contrastive loss on instances produced via (a) task-aware masking and (b) class labels. Their results and ablations show a better performance over baselines.

**Questions:**

See Cons.

**Ethical Concerns:**

["NO or VERY MINOR ethics concerns only"]

**Final Justification:**

One of my remaining concern (and justification for my score) is the repeated mention of an *industrial partner*. Ideally, mentioning such details should be reserved for the final / camera ready version – and in my mind, it (softly) hurts the pure double-blind nature of the review process.

**Limitations:**

yes

**Paper Formatting Concerns:**

As listed in the Cons section above, I am concerned about the font size of the tables being too small.

**Quality:**

2

**Strengths And Weaknesses:**

## Pros

1. The idea of combining masking and contrastive loss in a semi-supervised setup on time-series data seems novel.
2. The experiments show good performance over the baselines.
3. Ablation shows benefits of DAReM over random masking.
4. Visualization shows a good separation of clusters.

## Cons

1.  I feel that there needs to be a better utilization of the space in the main paper.
    1. The paper shares too few details about DAReM in the main paper. For instance, it is not clear what φ, γ, ζ mean in more exact terms.
    2. It may be a good idea to perhaps offload less essential related work (which takes a whole page) to the appendix.
    3. The font size of the tables is too small.
2. In the ablation table (Table 6), the variations in the scores are too tiny. I wonder if this is expected or informative.
3. Mainly, I don’t see a connection between their initial motivation of tackling irregularly sampled time series and the main novelty of their method (DAReM + CL).
4. Also, I don’t see why irregular sampling would be a concern for, say, TARNet or SLOTS, since one can simply construct a positional embedding using the actual time-stamp.

---

> ### Author Rebuttal · Authors · 2025-07-29
>
> We express our gratitude to the reviewers for their careful reading and critical assessment of our work. We are greatly encouraged by the overall positive reception of our work, reflected in the reviewers' consensus across most reviewing criteria. We are glad to hear they found our approach of addressing irregularly sampled and non-stationary time series 'innovative' (nw2T), 'novel' (KH4k, 8jfK) and 'effective' (nw2T, 7K1F). Moreover, we are pleased that the reviewers acknowledged our efforts to evaluate our approach through 'comprehensive' (nw2T), 'thorough' (nw2T), and 'extensive' (7K1F, 8jfK) experiments and ablation studies. We are satisfied that these experiments demonstrate the effectiveness (nw2T, 7K1F) and strong performance (KH4k) of our approach compared to state-of-the-art baseline models. We will address your comments below and incorporate your feedback accordingly.
>
> **Q4)** _*Space utilization and offload less essential work (1.1, 1.2).*_
>
> Due to space constraints, we have chosen to offload specific information regarding the DAReM procedure. We explicitly provide the formal definition of each hyperparameter used to create the regional masks in the main document, on page 4, line 153. We consider it essential to refer to the related work, especially for readers who do not possess a similar background. Therefore, we have chosen to offload certain mechanics to the supplementary section, which interested readers can access, while maintaining the entirety of the related work. It is noteworthy that offloading or reducing the related work section can often lead to criticism from reviewers concerning missing related works; thus, a balanced trade-off is necessary. However, upon your recommendation, we are more than willing to accommodate your suggestion in the camera-ready version.
>
> **Q5)** _*The font size of the tables is too small (1.3).*_
>
> In the paper, we exclusively use the standard font size from the LaTex template. To retain as much information as possible and maintain transparency when selecting baseline models, we aimed to include as many baselines as feasible to avoid scrutiny for not including enough baselines. To ensure the tables fit within the page limit requirements, we enforced the tables to be text-width size using the command \adjustbox{max width=1.0\textwidth}{...}. In the camera-ready version, upon your recommendation, we would be happy to remove some baselines from the main document and include them in a comprehensive table in the supplementary information. As stated in our response to **Q2)**, we are also able to include additional experiments for Table 3, along with an additional task, which would necessitate reformatting Table 3 anyway.
>
> **Q6)** _*In the ablation table (Table 6), the variations in the scores are too tiny. I wonder if this is expected or informative.*_
>
> Indeed, the results for DKT vary only minimally. This is somewhat expected, considering the underlying data constellation of the DKT dataset, as explained in **Q2)** and Section D.2. When examining the other two datasets, we observe a deviation of about 1.3 to 2.7 percentage points, which is significant, especially considering that the experiments are conducted using 5 different seeds and data splits. Additionally, given that we are only testing different configurations while maintaining the overall modeling architecture, we do not expect to see large variations. We believe that the table provides reasonable evidence to conclude that: 1. semi-supervised CL is beneficial compared to self-supervised or supervised-only approaches by themselves and 2. masking regions is advantageous over masking individual elements, with the region parameter needing to be fine-tuned for optimal performance, as stated in the paper. This evidence supports our architectural choices, hence the ablations are also informative.
>
> **Q7)** _*Mainly, I don’t see a connection between their initial motivation of tackling irregularly sampled time series and the main novelty of their method (DAReM + CL).*_
>
> In the introduction, we clearly state the initial motivation for modeling specifically for non-stationary and irregularly sampled time series based on observations from our industrial partner. In the related work section, at the end of each paragraph, we describe how our proposed approach connects to potential challenges and how our modeling approach differs from others. This is followed by a distinct statement on page 4, line 145, explaining our rationale and how our approach sets itself apart. To summarize, by introducing artificially masked regions in sequences, we emphasize the sensation of irregular sampling (and non-stationarity). The rationale is that this emphasis will focus the learning on these irregularities when the transformer model forms the embedding representations during training in the latent space. Via our contrastive formulation, we then aim to align even disparate embeddings in the latent space to make the latent space, and hence the model, more robust to these irregularities. It is relatively common practice to mask sequences to create irregularly sampled data, as is done to achieve the datasets in Table 2 for example. To demonstrate this, we conduct an ablation study where we perform no masking and random masking (see Section 3.1, Table 5 and Table 27), showing that this approach outperforms the no masking and random masking approaches.
>
> **Q8)** _*Also, I don’t see why irregular sampling would be a concern for, say, TARNet or SLOTS, since one can simply construct a positional embedding using the actual time-stamp.*_
>
> To the best of our understanding, TARNet does not construct positional embeddings based on actual timestamps; at least, this is not achieved in their code implementation. We assume you might mean TimesURL? In the context of irregularly sampled time series, transformer models may face challenges due to their inherent reliance on regular positional encoding (which assumes equally spaced time intervals). Modeling approaches can indeed mitigate the issue of irregular sampling by using positional embeddings based on actual timestamps. However, this would require access to the actual timestamps, which in many datasets, including the dataset from our industrial sponsor, is not the case. Hence, we do not know/have real access to the exact timestamps and cannot account for the temporally contextualized informed embeddings.
>
> If our responses have adequately addressed your questions and concerns, we would appreciate your consideration in updating your recommendation score. Otherwise, we welcome further discussion.

---

> > ### Author Response · Authors · 2025-08-05
> >
> > Dear Reviewer,
> >
> > We appreciate your critical review of our work and your honest assessment. The primary weakness you identified was your concern about the connection between tackling irregularly sampled time series and our approach. This concern was not raised by any other reviewer. We have attempted to clarify this issue in our rebuttal.
> >
> > We have also addressed the additional, more minor concerns you mentioned. We hope that our rebuttal sufficiently addresses these points and that we can reach a consensus that may lead to updating your initial rating. We welcome any opportunities for further clarification through discussion, if necessary.
> >
> > Thank you once again for the time and effort to review our work.

---

### Official Review · Reviewer_nw2T · 2025-07-03

**Clarity:** 3
**Significance:** 3
**Originality:** 2
**Rating:** 4
**Confidence:** 3

**Summary:**

This paper proposes STaRFormer, a Transformer-based framework designed to handle irregularly sampled and non-stationary sequential data effectively. The key contributions include a dynamic attention-based regional masking (DAReM) mechanism and a semi-supervised contrastive learning (CL) method, explicitly coupling representation learning with downstream tasks. Experiments conducted across 19 diverse datasets, including irregularly sampled and real-world non-stationary datasets, demonstrate that STaRFormer achieves competitive  performance relative to current state-of-the-art (SOTA) methods.

**Questions:**

Is there a reason for putting the supplementary material in the main paper? I think the authors didn't follow the submission form properly.

**Ethical Concerns:**

["NO or VERY MINOR ethics concerns only"]

**Final Justification:**

Most concerns from the initial review have been addressed in the rebuttal. The authors clarified the computational efficiency limitation, noting potential future improvements (e.g., flash attention), and explained modest gains on certain datasets with additional robustness analysis and broader benchmark results. The performance improvements, especially on the full UEA benchmark and regression tasks, further support the method’s effectiveness. Formatting issues are minor and easily addressed. Overall, the rebuttal reinforces the initial assessment, and I maintain my weak accept recommendation.

**Limitations:**

The manuscript adequately addressed the limitations.

**Paper Formatting Concerns:**

The authors did not follow the format of NeurIPS and included supplementary material in the main paper.

**Quality:**

3

**Strengths And Weaknesses:**

Strengths

1. Novel Dynamic Masking Approach (DAReM): The proposed attention-based regional masking strategy is innovative and addresses the critical challenge of handling irregular sampling and non-stationarity effectively by adaptively perturbing key regions of the sequence based on learned attention scores.

2. Effective Semi-Supervised Contrastive Learning: The proposed contrastive learning paradigm combines supervised (class-wise) and self-supervised (batch-wise) signals, explicitly linking representation learning with downstream tasks, contributing to robust representations and improved performance.

3. Comprehensive Experimental Validation: The extensive evaluation across multiple datasets (regular, irregularly sampled, and non-stationary time series) provides strong empirical support for STaRFormer's versatility and effectiveness in diverse real-world scenarios.

4. Thorough Ablation Studies: The paper provides extensive ablations and clearly demonstrates the incremental contributions of each proposed component (DAReM, semi-supervised CL), supporting the methodological design choices.

Weaknesses

1. Limited Computational Efficiency: The proposed dynamic masking strategy, relying on full attention mechanisms, inherently incurs computational complexity of 𝑂(𝑁^2), potentially hindering scalability for very long sequences, as acknowledged by the authors. Although this complexity affects only training, it can limit practical usability on larger datasets.

2. Marginal Improvements on Certain Datasets: Despite consistent outperformance across most datasets, performance gains are modest in some cases, which may raise concerns regarding whether the additional complexity introduced by DAReM and contrastive learning fully justifies the marginal performance gains.

---

> ### Author Rebuttal · Authors · 2025-07-29
>
> We express our gratitude to the reviewers for their careful reading and critical assessment of our work. We are greatly encouraged by the overall positive reception of our work, reflected in the reviewers' consensus across most reviewing criteria. We are glad to hear they found our approach of addressing irregularly sampled and non-stationary time series 'innovative' (nw2T), 'novel' (KH4k, 8jfK) and 'effective' (nw2T, 7K1F). Moreover, we are pleased that the reviewers acknowledged our efforts to evaluate our approach through 'comprehensive' (nw2T), 'thorough' (nw2T), and 'extensive' (7K1F, 8jfK) experiments and ablation studies. We are satisfied that these experiments demonstrate the effectiveness (nw2T, 7K1F) and strong performance (KH4k) of our approach compared to state-of-the-art baseline models. We will address your comments below and incorporate your feedback accordingly.
>
> **Q1)** _*Limited Computational Efficiency: The proposed dynamic masking strategy, relying on full attention mechanisms, inherently incurs computational complexity of $𝑂(𝑁^2)$, potentially hindering scalability for very long sequences, as acknowledged by the authors. Although this complexity affects only training, it can limit practical usability on larger datasets.*_
>
> As you noted, we briefly acknowledge this limitation in the main paper. The computational complexity associated with transformers is a recognized concern, which is the reason we aim to explore methodologies that are more computationally efficient, such as flash attention in future work. Our initial objective was to establish a performance-effective methodology before potentially enhancing computational efficiency.
>
> **Q2)** _*Marginal Improvements on Certain Datasets: Despite consistent outperformance across most datasets, performance gains are modest in some cases, which may raise concerns regarding whether the additional complexity introduced by DAReM and contrastive learning fully justifies the marginal performance gains.*_
>
> In the DKT dataset, the initial motivation (task), we aim to address this concern through an additional robustness analysis and further elaborate on the challenges associated with the ground truth labeling in Appendix Section D.2. Regarding the numerous other datasets, the model performances are generally robust, achieving accuracy rates up to 99% in certain instances. Consequently, only marginal improvements are feasible in many situations. During the review phase, we computed the entire UEA benchmark, as presented in Table 3, comprising 30 datasets, along with an additional extrinsic regression task involving 19 datasets. On the complete UEA benchmark, STaRFormer (0.795) surpasses the next best models, TARNet (0.755) and TimesURL (0.752), by approximately 4 percentage points. In the regression task, see the tseregression (dot org) website, we also achieve state-of-the-art performance in terms of average relative mean difference, with STaRFormer (-0.254) outperforming the next best model, Rocket (-0.245), by approximately 1 percentage point. We are ready to incorporate these findings upon your suggestion to substantiate performance scores and address any further potential evaluation concerns.
>
> **Q3)** _*The authors did not follow the format of NeurIPS and included supplementary material in the main paper.*_
>
> As stated in the criteria for paper formatting instructions on the NeurIPS website in the Call for Papers section: "The authors may optionally choose to include some or all of the technical appendices in the same PDF above. But those included parts cannot be changed after the full submission deadline." We understood this as an option to attach the supplementary material to the main PDF. Upon your recommendation, we are happy to attach it as an additional file.
>
> If our responses have adequately addressed your questions and concerns, we would appreciate your consideration in updating your recommendation score. Otherwise, we welcome further discussion.

---

> > ### Author Response · Authors · 2025-08-05
> >
> > Dear Reviewer,
> >
> > We sincerely appreciate the time and effort you have dedicated to reviewing our paper, as well as your overall positive feedback. We have addressed the two weaknesses you identified, along with the additional question regarding the formatting, in our rebuttal.
> >
> > If our responses have satisfactorily addressed all your concerns, we would be grateful if you could take this into account in your final evaluation. Should there be any areas where our responses have not been sufficient, we welcome any opportunities for further clarification.
> >
> > Thank you once again for your valuable input.

---

> > ### Comment · Reviewer_nw2T · 2025-08-06
> >
> > I appreciate the authors for providing the rebuttal. Most of the concerns raised in the initial review stage are addressed in the rebuttal. Hence I keep my original recommendation of weak accept.

---

> ### Author Response · Authors · 2025-08-07
>
> Thank you for reviewing our rebuttal. We are delighted that we could address your concerns. We take your recommendations seriously and aim to enhance our work based on them. Since our rebuttal addressed most of your concerns but could not fully convince you, we would be grateful to receive more detailed insights to further enhance our work for a stronger recommendation. We would greatly appreciate your guidance.

---

### Note · Authors · 2025-08-12

Our paper presents STaRFormer, a time series modeling framework that explicitly couples representation learning with a downstream task, addressing challenges such as non-stationarity and irregular sampling. It features a Dynamic Attention-based Regional Masking method and employs a semi-supervised contrastive learning approach to enhance latent-space representation learning for downstream tasks. Our extensive evaluations and ablation studies on 19 datasets, spanning classification and anomaly detection, demonstrate its SOTA performance.

Most reviewers agreed on the quality of our work, describing it as innovative, novel, and effective, with comprehensive and thorough experiments. We believe all concerns raised during the review phase were adequately addressed in the rebuttal and discussion stages.

**Reviewer nw2T** responded to our rebuttal in three short sentences, acknowledging that we had addressed their concerns but maintaining their original rating. Unfortunately, no explanation was given for not updating the score, leaving us unable to clarify our position.

**Reviewer KH4k** was the only reviewer to assign a lower rating. They acknowledged our rebuttal just hours before the deadline and did
not communicate during the discussion phase, despite our earlier attempts to initiate dialogue.

**Reviewer 7K1F** engaged in a constructive exchange early in the discussion. We provided detailed clarifications supported by specific evidence and references for the follow-up questions. Although we did not receive any further feedback, we believe our responses have thoroughly resolved the concerns raised in the review.

**Reviewer BjfK** actively engaged from the outset and responded to all our comments. However, their primary concerns centered on limitations we had already acknowledged in the paper and identified as an area for immediate future work. While we appreciate their engagement, we find it slightly disappointing that this single point served as the sole reason for not upgrading the rating.

_**Kind request to the AC/Senior AC**_:

Please consider whether the rebuttal responses have been fully accounted for in post-rebuttal assessments. If appropriate, we would welcome an invitation for reviewers to update their scores or an AC-led synthesis that weighs the technical contribution alongside our clearly stated limitations.

Finally, we want to thank the reviewers for their time and guidance, and the AC/Senior AC for their effort and considerations.

---

### Decision · Program_Chairs · 2025-09-17

**Decision:**

Accept (poster)

**Comment:**

The authors propose a Transformer-based framework designed to handle irregularly sampled and non-stationary sequential data effectively. The key contributions include a dynamic attention-based regional masking (DAReM) mechanism and a semi-supervised contrastive learning (CL) method, explicitly coupling representation learning with downstream tasks. Experiments conducted across 19 diverse datasets, including irregularly sampled and real-world non-stationary datasets, demonstrate that STaRFormer achieves competitive performance relative to current state-of-the-art (SOTA) methods. During rebuttal, most concerns are addressed, including clarifying the computational efficiency limitation, the motivation and details of the proposed components, and providing additional robustness analysis/broader benchmark results. The final ratings are three borderline acceptances and one borderline rejection. However, the borderline rejection reviewer's final concern is "the repeated mention of an industrial partner". The AC checked that the double-blind requirement has not been breached. Hence, it is not clear what the remaining key concern is. As a result, the AC recommends acceptance.